# 3d $\mathcal{N} = 4$ gauge theories on an elliptic curve

**Mathew Bullimore[1⋆] and Daniel Zhang[2†]**

**1** Department of Mathematical Sciences, Durham University,
Durham, DH1 3LE, United Kingdom
**2** Department of Applied Mathematics and Theoretical Physics, University of Cambridge,
Cambridge, CB3 0WA, United Kingdom

⋆ mathew.r.bullimore@durham.ac.uk , † d.zhang@damtp.cam.ac.uk

## Abstract

This paper studies 3$d$ $\mathcal{N} = 4$ supersymmetric gauge theories on an elliptic curve, with the aim to provide a physical realisation of recent constructions in equivariant elliptic cohomology of symplectic resolutions. We first study the Berry connection for supersymmetric ground states in the presence of mass parameters and flat connections for flavour symmetries, which results in a natural construction of the equivariant elliptic cohomology variety of the Higgs branch. We then investigate supersymmetric boundary conditions and show from an analysis of boundary 't Hooft anomalies that their boundary amplitudes represent equivariant elliptic cohomology classes. We analyse two distinguished classes of boundary conditions known as exceptional Dirichlet and enriched Neumann, which are exchanged under mirror symmetry. We show that the boundary amplitudes of the latter reproduce elliptic stable envelopes introduced by Aganagic-Okounkov, and relate boundary amplitudes of the mirror symmetry interface to the mother function in equivariant elliptic cohomology. Finally, we consider correlation functions of Janus interfaces for varying mass parameters, recovering the chamber R-matrices of elliptic integrable systems.

 Check for updates

# 1 Introduction

This paper studies 3d $\mathcal{N} = 4$ supersymmetric gauge theories on $E_\tau \times \mathbb{R}$, where $E_\tau$ is a complex elliptic curve with complex structure parameter $\tau$, and Ramond-Ramond boundary conditions are imposed. The aim is to give a precise physical construction of work on the equivariant elliptic cohomology of conical symplectic resolutions, elliptic stable envelopes, and elliptic $R$-matrices [1–8].

A supersymmetric gauge theory on $E_\tau \times \mathbb{R}$ can be regarded as an infinite-dimensional supersymmetric quantum mechanics on $\mathbb{R}$. An important question is to determine the space of supersymmetric ground states, which can be understood as the cohomology of a supercharge

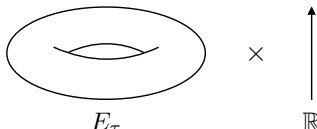

Figure 1: The set-up of this paper is supersymmetric gauge theories on $E_\tau \times \mathbb{R}$.

provided the system remains gapped. This has a straightforward answer in the presence of generic mass deformations. Suppose we can turn on mass deformations such that the gauge theory has only isolated massive and topologically trivial vacua, indexed by $\alpha$. Then there are corresponding supersymmetric ground states $|\alpha\rangle$ on $E_\tau \times \mathbb{R}$ labelled in the same manner.

The richness of this problem arises in determining supersymmetric ground states for non-generic mass deformations and more broadly how they depend on background fields on $E_\tau \times \mathbb{R}$ associated to flavour symmetries.

Let us consider a supersymmetric gauge theory with an abelian flavour symmetry $T$ acting on elementary matter supermultiplets.[1] This flavour symmetry acts on the Higgs branch moduli space of the supersymmetric gauge theory, which we denote by $X$, and the isolated massive vacua are the fixed points $X^T = \{\alpha\}$. The computation of supersymmetric ground states on $E_\tau \times \mathbb{R}$ is then compatible with the following parameters associated to the flavour symmetry:

- Real mass parameters valued in $\mathfrak{t} = \text{Lie}(T)$.

- A background flat connection on $E_\tau$ for the flavour symmetry $T$, parametrised by the $\text{rk}\,T$-dimensional complex torus

$$E_T := \mathfrak{t} \otimes_{\mathbb{R}} E_\tau . \tag{1}$$

The total parameter space is therefore

$$\mathfrak{t} \times E_T \cong (\mathbb{R} \times E_\tau)^{\text{rk}(T)} \tag{2}$$

and these parameters can be regarded as expectation values of scalar fields in a background vector multiplet for the symmetry $T$ in the supersymmetric quantum mechanics.

The dependence of supersymmetric ground states on these parameters is controlled by a Berry connection. A consequence of supersymmetry is that the Berry connection is enhanced to a solution of generalised Bogomolny equations [9–12]. The asymptotic behaviour of the solution is controlled by the effective supersymmetric Chern-Simons couplings in the vacua and there are 't Hooft monopole singularities at loci where the supersymmetric quantum mechanics fails to be gapped.

We will not need the full structure of the supersymmetric Berry connection here. Instead, we use a consequence of the generalised Bogomolny equations that there is a complex flat Berry connection in the real directions $\mathfrak{t}$ and a holomorphic Berry connection in the complex directions $E_T$, which commute with each other. This induces the structure of a holomorphic vector bundle $\mathcal{E}$ on each complex torus $\{m\} \times E_T$, which has a piecewise constant dependence on the mass parameters $m \in \mathfrak{t}$.

The piece-wise constant dependence is controlled by a hyperplane arrangement in the space of mass parameters $\mathfrak{t}$ constructed from hyperplanes where the gauge theory is no longer completely massive. To describe the hyperplanes geometrically, let us denote by $T_m \subset T$ the 1-parameter subgroup of the flavour symmetry generated by a mass parameter $m \in \mathfrak{t}$. We then have

---

[1]Topological symmetries transforming monopole operators are incorporated in the main text.

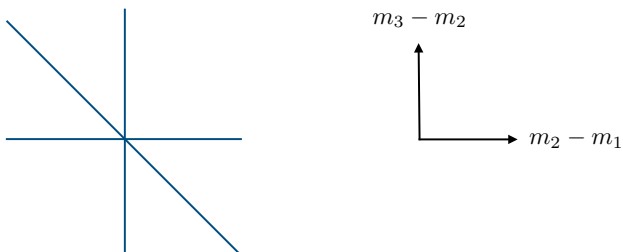

Figure 2: An example of a hyperplane arrangement in $\mathfrak{t} \cong \mathbb{R}^2$ arising in supersymmetric QED with Higgs branch $X = T^*\mathbb{CP}^2$ and flavour symmetry $T \cong U(1)^2$ generated by mass three parameters $(m_1, m_2, m_3)$ obeying the constraint $m_1 + m_2 + m_3 = 0$. There are six faces of maximal dimension where the fixed locus $X^{T_m}$ consists of three isolated points $\alpha = 1, 2, 3$, six faces of dimension one where it is a union of a point $\alpha = 1$ and a moduli space $T^*\mathbb{CP}^1$ connecting $\alpha = 2, 3$ or permutations thereof, and the origin where it is the whole $T^*\mathbb{CP}^2$.

- For a generic mass parameter, $X^{T_m} = \{\alpha\}$.

- For a non-generic mass parameter, $X^{T_m} \neq \{\alpha\}$ and an extended moduli space opens up. This happens along hyperplanes through the origin.

In general, the fixed locus $X^{T_m}$ and the structure of the holomorphic vector bundle $\mathcal{E}$ on $\{m\} \times E_T$ depends on a face of the hyperplane arrangement. As example of a hyperplane arrangement for supersymmetric QED is illustrated in figure 2.

For a generic mass parameter, lying in a face of maximal dimension or chamber of the hyperplane arrangement, $X^{T_m} = \{\alpha\}$ and the holomorphic vector bundle admits a holomorphic filtration

$$0 \subset \mathcal{E}_{\alpha_1} \subset \mathcal{E}_{\alpha_2} \subset \cdots \subset \mathcal{E}_{\alpha_N} = \mathcal{E}, \tag{3}$$

giving a complete flag on each fibre. The factors of automorphy of the holomorphic line bundles $\mathcal{L}_{\alpha_i} \cong \mathcal{E}_{\alpha_i}/\mathcal{E}_{\alpha_{i-1}}$ are fixed by the effective supersymmetric Chern-Simons couplings in the massive vacuum $\alpha$. The collection of line bundles $\{\mathcal{L}_\alpha\}$, or equivalently the associated graded $\mathcal{G}(\mathcal{E}) = \bigoplus_\alpha \mathcal{L}_\alpha$, can also be regarded as a section of a holomorphic line bundle on the union of identical copies $E_T^{(\alpha)} \cong E_T$ associated to each of the vacua $\alpha$,

$$\mathrm{Ell}_T(\{\alpha\}) = \bigsqcup_\alpha E_T^{(\alpha)}. \tag{4}$$

This is the $T$-equivariant elliptic cohomology variety of the fixed point set $X^T = \{\alpha\}$.

As the mass parameters are specialised to lie on faces of the hyperplane arrangement of lower dimension this structure becomes more intricate. We argue that on a general face of the hyperplane arrangement, $\mathcal{E}$ is encoded in a holomorphic line bundle on the equivariant elliptic cohomology variety of the fixed locus $T_m \subset T$,

$$\mathrm{Ell}_T(X^{T_m}) \longrightarrow E_T. \tag{5}$$

This is an $N$-sheeted cover of the space of $T$-flat connections $E_T$, which is obtained by making certain identifications on the sheets of $\mathrm{Ell}_T(\{\alpha\})$. Here $N$ is the number of massive vacua $\alpha$. At the origin of the hyperplane arrangement, $m = 0$, this is the equivariant elliptic cohomology variety $\mathrm{Ell}_T(X)$ of the entire Higgs branch $X$. This is why equivariant elliptic cohomology arises in this problem.

With the structure of the supersymmetric Berry connection in hand, we study boundary conditions compatible with the flavour symmetry $T$ and the supercharge whose cohomology

computes supersymmetric ground states. The fundamental objects of study are boundary amplitudes $\langle B|\alpha\rangle$, defined as the path integral on $E_\tau \times \mathbb{R}_+$ with the boundary condition $B$ at $x^3 = 0$ and a supersymmetric ground state $|\alpha\rangle$ at $x^3 \to \infty$, as illustrated in figure 3.

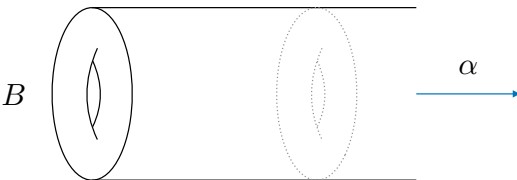

Figure 3: A boundary amplitude is computed from a partition function on $E_\tau \times \mathbb{R}_+$ with the boundary condition $B$ at $x^3 = 0$ and a massive vacuum $\alpha$ at $x^3 \to \infty$.

The boundary amplitudes can be regarded as elliptic genera of effective two-dimensional theories obtained by reduction on $\mathbb{R}_+$. From an analysis of boundary mixed 't Hooft anomalies and effective supersymmetric Chern-Simons couplings, we can determine how boundary amplitudes transform under global background gauge transformations on $E_\tau$ for the flavour symmetry $T$. This identifies boundary amplitudes as sections of holomorphic line bundles on $E_T$.

Furthermore, suppose a boundary condition $B$ is compatible with a mass parameter $m$ on some face of the hyperplane arrangement. Then we show that the collection of boundary amplitudes defined using supersymmetric ground states on that face of the hyperplane arrangement transform in such a way that they glue to a section of a holomorphic line bundle on $\mathrm{Ell}_T(X^{T_m})$. This provides a recipe to construct equivariant elliptic cohomology classes from suitable boundary conditions. We illustrate this by computing the boundary amplitudes of Neumann boundary conditions representing the elliptic cohomology classes of holomorphic Lagrangian sub-manifolds in $X$.

We then consider two distinguished collections of UV boundary conditions whose elements are in 1-1 correspondence with vacua $\alpha$. In abelian gauge theories, they have an explicit construction as follows:

1. **Exceptional Dirichlet**

   Exceptional Dirichlet boundary conditions $D_\alpha$ mimic the presence of a vacuum $\alpha$ at infinity in the presence of a mass parameter $m$ and are supported on the attracting set of the vacuum $\alpha$ under gradient flow for the moment map of the $T_m$-action on $X$. This allows boundary amplitudes to be computed as interval partition functions on $E_\tau \times [0, \ell]$ with the boundary condition $B$ at $x^3 = 0$ and the exceptional Dirichlet boundary condition $D_\alpha$ at $x^3 = \ell$, opening up the possibility of using results from supersymmetric localisation.

2. **Enriched Neumann**

   Enriched Neumann boundary conditions $N_\alpha$ involve Neumann boundary conditions for the gauge fields and couplings to $\mathbb{C}^*$-valued chiral multiplets via boundary superpotentials and twisted superpotentials. They are supported on unions of attracting sets corresponding to the stable envelopes introduced in [13]. We demonstrate that their boundary wavefunctions and amplitudes reproduce the construction of elliptic stable envelopes [5].

We make the observation that these two distinguished classes of boundary conditions are exchanged under 3d mirror symmetry [14]. In fact, we derive the form of enriched Neumann boundary conditions $N_\alpha$ by colliding exceptional Dirichlet boundary conditions $D_\alpha$ with the

mirror symmetry interface introduced in [15]. In the process, we identify correlation functions of this mirror symmetry interface with the mother function in equivariant elliptic cohomology [8].

To make the connection with elliptic stable envelopes more precise, we consider supersymmetric Janus interfaces for the real mass parameters $m$. These are interfaces representing a position dependent profile $m(x^3)$ connecting different faces of the hyperplane arrangement. We mainly focus on interfaces interpolating between chambers of the hyperplane arrangement or between the origin and a chamber, as illustrated in figure 4. A crucial observation is that the computation of boundary amplitudes and overlaps are independent of the profile in the intermediate region. This allows flexibility in computing the overlaps of boundary conditions compatible with mass parameters on different faces on the hyperplane arrangement.

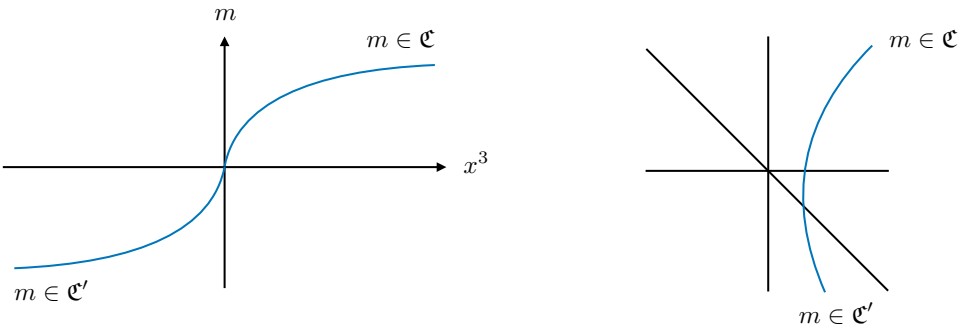

Figure 4: The left figure illustrates schematically a Janus interpolation function $m(x^3)$ between chambers $\mathfrak{C}$, $\mathfrak{C}'$. The right shows the projection onto the mass parameter plane in supersymmetric QED implementing between chambers $\mathfrak{C} = \{m_1 < m_2 < m_3\}$ and $\mathfrak{C}' = \{m_3 < m_1 < m_2\}$.

An important example is the correlation function of the Janus interface interpolating between a supersymmetric ground state $\langle \alpha |$ at $m = 0$ and an enriched Neumann boundary $|N_\beta\rangle$ condition for a generic mass parameter $m$ in some chamber of the hyperplane arrangement. The matrix of such boundary amplitudes represents a collection of $T$-equivariant elliptic cohomology classes on $X$ labelled by $\beta$. We show that they reproduce precisely the elliptic stable envelopes [5].

Finally, we consider the correlation functions of Janus interfaces between enriched Neumann boundary conditions defined for mass generic mass parameters in two different chambers of the hyperplane arrangement. We argue that independence of the profile function implies that they obey the same algebraic relations as chamber R-matrices of elliptic quantum integrable systems and check this correspondence explicitly in examples. We further exploit the independence of the profile to reproduce the decomposition of such R-matrices into elliptic stable envelopes.

The work in this paper has applications to half-indices or partition function of supersymmetric boundary conditions on three-manifolds $M_3$ with boundary $\partial M_3 = E_\tau$ [16–19]. In combination with our previous work [20], this can be used to shed light on the role of elliptic stable envelopes in implementing mirror symmetry of vertex functions [5, 73, 74]. We will return to this topic in a future publication [21].

**Note added:** in the process of completing this project we became aware of related ongoing work of Mykola Dedushenko and Nikita Nekrasov [22], and we are grateful to them for agreeing to coordinate the release.

## 2 Preliminaries

We introduce here background and assumptions on 3d $\mathcal{N} = 4$ gauge theories that we will need to connect with equivariant elliptic cohomology. This will serve to set up notation and introduce some constructions that are important for this paper but not commonly covered in the literature, such as the role of effective supersymmetric Chern-Simons couplings and domain walls between isolated massive vacua.

### 2.1 Data and Symmetries

We provide a brief summary of 3d $\mathcal{N} = 4$ supersymmetric gauge theories, referring the reader to [15, 23] for more details.

We consider theories specified by a compact connected group $G$ and a linear quaternionic representation of the form $Q = T^*R$ where $R$ is a unitary representation. The R-symmetry is $SU(2)_C \times SU(2)_H$, with vector multiplet scalars $\varphi^{\dot{A}\dot{B}}$ transforming in the adjoint of $SU(2)_C$ and hypermultiplet scalars $X^A$ in the fundamental of $SU(2)_H$. The flavour symmetry is of the form $G_C \times G_H$, where

- The Coulomb branch symmetry $G_C$ has an abelian subgroup given by the topological symmetry

$$T_C = \mathrm{Hom}(\pi_1(G), U(1)) \cong Z(^L G), \tag{6}$$

  which we assume is the maximal torus.

- The Higgs branch flavour symmetry

$$G_H = N_{U(R)}(G)/G \tag{7}$$

  is the normaliser of the unitary representation $G \subset U(R)$ modulo $G$. The hypermultiplets roughly transform in a quaternionic representation of $G \times G_H$.

We can introduce FI parameters $\zeta^{AB}$ and mass parameters $m^{\dot{A}\dot{B}}$ by turning on expectation values for background twisted vector multiplets and vector multiplets. In most of this article we turn on only generic real parameters

$$\zeta := \zeta^{+-} \qquad m := m^{\dot{+}\dot{-}}, \tag{8}$$

leaving unbroken maximal tori $U(1)_C \times U(1)_H$ and $T_C \times T_H$ of the R-symmetry and flavour symmetry respectively. We normalise the unbroken R-symmetries to have integer weights.

We can break to 3d $\mathcal{N} = 2$ supersymmetry by introducing a real mass parameter $\epsilon$ for the following combination of R-symmetries,

$$T_t := U(1)_H - U(1)_C, \tag{9}$$

which becomes a distinguished flavour symmetry from a 3d $\mathcal{N} = 2$ perspective. The vector multiplet scalars decompose into a real scalar $\sigma = \varphi^{\dot{1}\dot{2}}$ transforming in a vector multiplet and a complex scalar $\varphi = \varphi^{\dot{1}\dot{1}}$ transforming in an adjoint chiral multiplet. Similarly, the hypermultiplet scalars decompose into a pair of scalars $X = X^1$, $Y = \bar{X}^2$ transforming in chiral multiplets in unitary representations $R, R^*$. Finally, there is a superpotential

$$W = \varphi \cdot \mu_{\mathbb{C}}, \tag{10}$$

where $\mu_{\mathbb{C}} : T^*R \to \mathfrak{g}^* \otimes_{\mathbb{R}} \mathbb{C}$ is the complex moment map for the $G$ action on the hypermultiplet representation $T^*R$.

### 2.1.1 Comment on Notation

Our notation is somewhat of a compromise between standard notation in supersymmetric gauge theory and the mathematics literature on equivariant elliptic cohomology. We therefore take this opportunity to summarise some notation used in this paper.

- We define $T := T_H \times T_t$ as the maximal torus of the 3d $\mathcal{N} = 2$ flavour symmetry acting on elementary chiral multiplets. We denote the corresponding Lie algebra by $\mathfrak{t}$ and use a shorthand notation $x = (m, \epsilon) \in \mathfrak{t}$ to denote collectively the associated real mass parameters.

- We define $T_f := T_C \times T_H \times T_t$ as the total 3d $\mathcal{N} = 2$ flavour symmetry acting on elementary chiral multiplets and monopole operators. We denote the corresponding Lie algebra by $\mathfrak{t}_f$ and use a shorthand notation $x_f = (\zeta, m, \epsilon) \in \mathfrak{t}_f$ to denote collectively the associated real parameters.

In comparing with the mathematics literature, the complexification of $T_H$ is often denoted by $A$ and the complexification of $T_t$ by $\hbar$.

### 2.1.2 Example

A running example is supersymmetric QED, with $G = U(1)$ and $Q = T^* \mathbb{C}^N$ where $\mathbb{C}^N$ denotes $N$ copies of the charge $+1$ representation. The flavour symmetries are $G_C = U(1)$ and $G_H = PSU(N)$. Correspondingly, we introduce an FI parameter $\zeta \in \mathbb{R}$, mass parameters $(m_1, \ldots, m_N) \in \mathbb{R}^{N-1}$ with $\sum_\alpha m_\alpha = 0$.

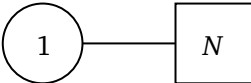

Figure 5: Supersymmetric QED with $N$ flavours represented as a quiver.

The hypermultiplets decompose into chiral multiplets with complex scalar components $(X_\alpha, Y_\alpha)$, which transform with charge $(+1, -1)$ under the gauge symmetry, in the anti-fundamental and fundamental representation of $G_H = PSU(N)$ and with charge $(+1, +1)$ under $T_t$. The chiral multiplets $\varphi, X_\alpha, Y_\alpha$ have total real mass $-2\epsilon, \sigma - m_\alpha + \epsilon, -\sigma + m_\alpha + \epsilon$ respectively. This is represented as a quiver in figure 5.

## 2.2 Supersymmetric Vacua

We assume the gauge theory flows to an interacting superconformal fixed point (without a decoupled free sector) and, upon introducing generic real FI and mass parameters, has isolated, massive, topologically trivial vacua. We label the isolated vacua by indices, $\alpha, \beta, \cdots$ and denote the number of them by $N$.

The isolated supersymmetric vacua can be regarded as solutions of the classical vacuum equations, which are summarised as the simultaneous critical points of the complex 3d $\mathcal{N} = 2$ superpotential $W$ and the real superpotential

$$
\begin{aligned}
h &= \sigma \cdot (\mu_{\mathbb{R}} + [\varphi, \varphi^\dagger] - \zeta) + m \cdot \mu_{H,\mathbb{R}} + \epsilon \cdot \mu_{t,\mathbb{R}} \\
&= \sigma \cdot (\mu_{\mathbb{R}} + [\varphi, \varphi^\dagger] - \zeta) + x \cdot \mu_{T,\mathbb{R}},
\end{aligned}
\tag{11}
$$

where $\mu_{\mathbb{R}}, \mu_{H,\mathbb{R}}, \mu_{t,\mathbb{R}}$ are the real moment maps for the $G$, $T_H$, $T_t$ action on $T^*R$. In the second line, we have used the shorthand notation $x = (m, \epsilon)$ and $\mu_{T,\mathbb{R}} = (\mu_{H,\mathbb{R}}, \mu_{t,\mathbb{R}})$ for the mass parameters and moment maps for $T = T_H \times T_t$. Our assumptions require that for generic mass

and FI parameters there are isolated critical points where the gauge symmetry is completely broken.

The supersymmetric vacua admit the following combinatorial description. Each supersymmetric vacuum $\alpha$ is specified by a set of $r := \text{rank}(G)$ distinct weights

$$\alpha : \quad \{\varrho_1, \ldots, \varrho_r\} \tag{12}$$

appearing in the $G \times T_H \times T_t$ weight decomposition of the hypermultiplet representation $T^*R$. These weights are the charges of the hypermultiplet fields that do not vanish in the vacuum. If we decompose into components

$$\varrho_i = (\rho_i, \rho_{H,i}, \rho_{t,i}), \tag{13}$$

the vector multiplet scalar $\sigma$ is fixed by the equations

$$\rho_i \cdot \sigma + \rho_{H,i} \cdot m + \rho_{t,i} \cdot \epsilon = 0, \tag{14}$$

for all $i = 1, \ldots, r$.

To define a supersymmetric vacuum, the gauge components must satisfy

1. $\{\rho_1, \ldots, \rho_r\}$ span $\mathfrak{h}^*$,

2. $\zeta \in \text{Cone}^+(\rho_1, \ldots, \rho_r) \subset \mathfrak{h}^*$,

where $\mathfrak{h} \subset \mathfrak{g}$ is the Cartan subalgebra of $G$. Here we regard the FI parameter $\zeta$ as an element of $\mathfrak{h}^*$ through the inclusion $\mathfrak{t}_C = Z(\mathfrak{h}^*) \subset \mathfrak{h}^*$. These conditions resemble JK residue prescriptions appearing in the computation of supersymmetric observables and can be understood by realising supersymmetric vacua as fixed points on the Higgs branch, which is discussed below.

Finally, due to the hypermultiplet field expectation values, the Higgs branch flavour and R-symmetries preserved in a massive vacuum $\alpha$ are shifted compared to the UV gauge theory. When needed, we distinguish the unbroken symmetries in a massive vacuum $\alpha$ by a superscript $U(1)_H^{(\alpha)}$, $T_H^{(\alpha)}$, or from the perspective of 3d $\mathcal{N} = 2$ flavour symmetries $T^{(\alpha)} = T_H^{(\alpha)} \times T_t^{(\alpha)}$. However, we mostly drop the superscript to lighten the notation with the understanding that these symmetries are shifted as appropriate for the supersymmetric massive vacuum.

### 2.2.1 Example

In supersymmetric QED, the complex and real superpotentials are

$$\begin{aligned}
W &= \varphi \sum_{\alpha=1}^N X_\alpha Y_\alpha, \\
h &= \sum_{\alpha=1}^N (\sigma - m_\alpha + \epsilon)|X_\alpha|^2 + \sum_{\alpha=1}^N (-\sigma + m_\alpha + \epsilon)|Y_\alpha|^2 - \zeta\sigma.
\end{aligned} \tag{15}$$

There are $N$ isolated critical points for generic mass and FI parameters. The critical points $\alpha = 1, \ldots, N$ corresponds to non-vanishing expectation values $|X_\alpha|^2 = \zeta$ and $\sigma = m_\alpha - \epsilon$ when $\zeta > 0$, and $|Y_\alpha|^2 = -\zeta$ and $\sigma = m_\alpha + \epsilon$ when $\zeta < 0$.

### 2.3 Chern-Simons Couplings

An important characteristic of vacua $\alpha$ are the effective supersymmetric Chern-Simons terms for flavour symmetries and R-symmetries, which are obtained from a 1-loop computation by integrating out massive degrees of freedom [24–28].

We keep track of supersymmetric Chern-Simons terms for $\mathcal{N} = 2$ flavour symmetries $T_f$, which are encapsulated in pairings

$$K_\alpha : \Gamma_f \times \Gamma_f \to \mathbb{Z}, \tag{16}$$

where $\Gamma_f \subset \mathfrak{t}_f$ denotes the co-character lattice of $T_f$. As mentioned above, here $T_f = T_C \times T$ denotes the symmetries preserved in the massive vacuum $\alpha$, which may be shifted compared to the UV gauge theory definition.

The supersymmetric Chern-Simons terms $K_\alpha$ are piece-wise constant in the parameters $x_f$ and may jump across loci where an extended moduli space of supersymmetric vacua opens up. This structure is controlled by a vector central charge function

$$C_\alpha := K_\alpha(x_f, x_f). \tag{17}$$

The central charge function determines the tension of 2d $\mathcal{N} = (0,2)$ domain walls, which correspond to solutions of the gradient flow equations for the real superpotential $h$ in (11). A domain wall interpolating between vacua $\alpha$ and $\beta$ has tension proportional to $|C_\alpha - C_\beta|$. This cuts out loci

$$\{C_\alpha - C_\beta = 0\} \subset \mathfrak{t}_f, \tag{18}$$

where the tension of a domain wall connecting $\alpha$ and $\beta$ vanishes and a compact moduli space opens up. There are additional loci where a non-compact moduli space attached to a single vacuum $\alpha$ opens up. We refer to the connected components of the complement of these loci as chambers. The couplings $K_\alpha$ are constant within chambers.

For a 3d $\mathcal{N} = 4$ gauge theory broken to $\mathcal{N} = 2$ by the mass parameter $\epsilon$, the potential supersymmetric Chern-Simons terms for flavour symmetries are restricted to the following:

- A mixed $T_H$-$T_C$ supersymmetric Chern-Simons term $\kappa_\alpha : \Gamma_H \times \Gamma_C \to \mathbb{Z}$.

- A mixed $U(1)_H$-$T_C$ coupling, which becomes a mixed flavour $T_t$-$T_C$ supersymmetric Chern-Simons term $\kappa_\alpha^C : \Gamma_t \times \Gamma_C \to \mathbb{Z}$.

- A mixed $T_H$-$U(1)_C$ coupling, which becomes a mixed flavour $T_H$-$T_t$ supersymmetric Chern-Simons term $\kappa_\alpha^H : \Gamma_H \times \Gamma_t \to \mathbb{Z}$.

- A mixed $U(1)_H$-$U(1)_C$ coupling, which becomes a flavour $T_t$-$T_t$ supersymmetric Chern-Simons term $\widetilde{\kappa}_\alpha : \Gamma_t \times \Gamma_t \to \mathbb{Z}$.

where $\Gamma_C, \Gamma_H, \Gamma_t$ denote the co-character lattices of $T_C, T_H, T_t$ respectively. The vector central charge function decomposes

$$C_\alpha = K_\alpha(x_f, x_f) = \kappa_\alpha(m, \zeta) + \kappa_\alpha^C(\epsilon, \zeta) + \kappa_\alpha^H(m, \epsilon) + \widetilde{\kappa}(\epsilon, \epsilon) \tag{19}$$

and we write collectively $K_\alpha = (\kappa_\alpha, \kappa_\alpha^C, \kappa_\alpha^H, \widetilde{\kappa}_\alpha)$. We note that the terms linear in the FI parameter arising from Chern-Simons couplings involving $T_C$ can be obtained by evaluating the real superpotential $h$ at a supersymmetric vacuum,

$$h|_\alpha = \kappa_\alpha(m, \zeta) + \kappa_\alpha^C(\epsilon, \zeta). \tag{20}$$

However, in a supersymmetric gauge theory all the coefficients are generated at 1-loop by integrating out massive fluctuations around a supersymmetric vacuum $\alpha$.

### 2.3.1 $\mathcal{N} = 4$ Limit

In the limit $\epsilon \to 0$ where $\mathcal{N} = 4$ supersymmetry is restored, the vector central charge function simplifies dramatically to

$$C_\alpha = \kappa_\alpha(m, \zeta) \tag{21}$$

and depends only on the $\mathcal{N} = 4$ supersymmetric Chern-Simons coupling $\kappa_\alpha : \Gamma_H \times \Gamma_C \to \mathbb{Z}$ between a vector multiplet and a twisted vector multiplet. In this limit, loci in the space of mass and FI parameters where supersymmetric vacua fail to be isolated are all of the form

$$\{C_\alpha = C_\beta\} \subset \mathfrak{t}_H \times \mathfrak{t}_C . \tag{22}$$

Their projections onto the two factors are linear hyperplanes through the origin, forming hyperplane arrangements in the spaces of mass and FI parameters $\mathfrak{t}_H$, $\mathfrak{t}_C$. We refer to the connected components of the complement of the hyperplanes, or equivalently the faces of the hyperplane arrangement of maximal dimension, as chambers $\mathfrak{C}_H$, $\mathfrak{C}_C$.

These hyperplane arrangements are a fundamental structure throughout this paper. Various quantities depend in a piece-wise constant way on the mass and FI parameters, such that they depend only on a choice of face of the hyperplane arrangement. In particular, $\kappa_\alpha$ is independent of the chambers, $\kappa_\alpha^C$ depends on a chamber $\mathfrak{C}_C$, $\kappa_\alpha^H$ depends on a chamber $\mathfrak{C}_H$, and $\widetilde{\kappa}_\alpha$ depends on both.

In this paper, we will ultimately set $\epsilon = 0$. However, in section 3 we will introduce another expectation value for the vector multiplet for the $\mathcal{N} = 2$ flavour symmetry $T_t$ and our computations will therefore be sensitive to all of the effective supersymmetric Chern-Simons couplings, $\kappa_\alpha$, $\kappa_\alpha^C$, $\kappa_\alpha^H$, $\widetilde{\kappa}_\alpha$.

### 2.3.2 Example

Let us determine the effective supersymmetric Chern-Simons terms in supersymmetric QED. We introduce fundamental weights $e_1, \ldots, e_N$ for $T_H$ and a fundamental weight $e_t$ for $T_t$. Then

$$\kappa_\alpha = -e_\alpha \otimes e_C , \tag{23}$$

which is independent of the mass and FI parameters.

When $\epsilon = 0$ the vector central charge is $C_\alpha = -m_\alpha \zeta$ and the loci $\{(m_\beta - m_\alpha)\zeta = 0\}$ split the space of FI parameters into two chambers depending on the sign of $\zeta$, and mass parameters into $N!$ chambers labelled by a permutation of $m_1 > m_2 > \cdots > m_N$. Our default chambers are

$$\begin{aligned}
\mathfrak{C}_C &= \{\zeta > 0\} , \\
\mathfrak{C}_H &= \{m_1 > m_2 > \cdots > m_N\} .
\end{aligned} \tag{24}$$

This hyperplane arrangement is illustrated for $N = 3$ in figure 6.

The remaining supersymmetric Chern-Simons couplings in the default chambers are

$$\begin{aligned}
\kappa_\alpha^C &= e_t \otimes e_C , \\
\kappa_\alpha^H &= \left( \sum_{\beta < \alpha} (e_\alpha - e_\beta) + \sum_{\beta > \alpha} (e_\beta - e_\alpha) \right) \otimes e_t , \\
\widetilde{\kappa}_\alpha &= (N - 2\alpha + 1) e_t \otimes e_t .
\end{aligned} \tag{25}$$

These couplings are unchanged when $\epsilon \neq 0$, provided $|\epsilon| \ll |m_\alpha - m_\beta|$. It is straightforward to check that $h|_\alpha = -\zeta(m_\alpha - \epsilon)$ when $\zeta > 0$, which reproduces the supersymmetric Chern-Simons couplings $\kappa_\alpha$, $\kappa_\alpha^C$ above.

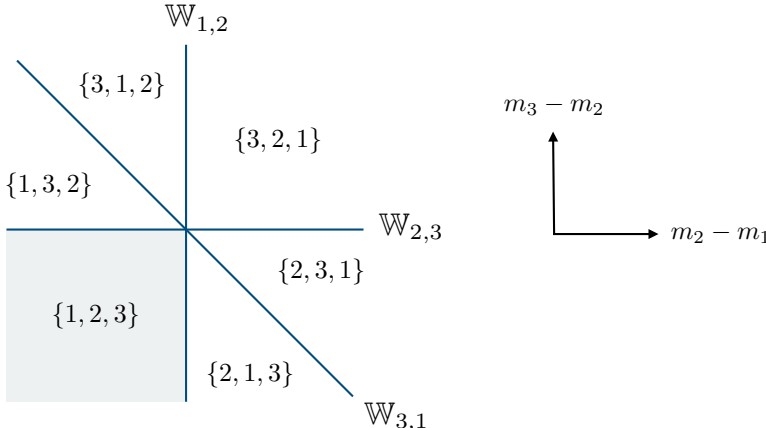

Figure 6: The hyperplane arrangement in $\mathfrak{t}_H \cong \mathbb{R}^2$ for supersymmetric QED with $N = 3$ with horizontal coordinate $m_2 - m_1$ and vertical coordinate $m_3 - m_2$. The chambers are labelled by permutations of $\{1, 2, 3\}$. The default chamber $m_1 > m_2 > m_3$ is shaded.

## 2.4 Higgs Branch Geometry

To connect with equivariant elliptic cohomology, we view a supersymmetric gauge theory through the lens of its Higgs branch. This means we first set $m = 0$, keeping $\zeta$ generic, such that the theory flows to a smooth sigma model onto a Higgs branch. We can then turn the mass parameter back on as a deformation of the sigma model.

The Higgs branch is the hyper-Kähler quotient

$$X = \mu_{\mathbb{C}}^{-1}(0) \cap \mu_{\mathbb{R}}^{-1}(\zeta)/G, \tag{26}$$

where $\mu_{\mathbb{C}}, \mu_{\mathbb{R}}$ are the real and complex moment maps for the $G$-action on the quaternionic representation $Q$. Under our assumptions, for a generic $\zeta$ this is a smooth hyper-Kähler manifold. We view $X$ as a Kähler manifold with a holomorphic symplectic form. The FI parameter $\zeta$ is a Kähler parameter and $X$ depends on the chamber $\mathfrak{C}_C$. The holomorphic symplectic structure is independent of $\zeta$ within each chamber. We typically fix a chamber and omit this from the notation.

The flavour symmetries are realised as follows:

- The topological symmetry is $T_C = \text{Pic}(X) \otimes_{\mathbb{Z}} U(1)$ and has co-character lattice $\Gamma_C = \text{Pic}(X)$. The chamber $\mathfrak{C}_C \subset \text{Pic}(X) \otimes_{\mathbb{Z}} \mathbb{R}$ containing $\zeta$ is the ample cone of $X$.

- The flavour symmetry $T_H$ is a maximally commuting set of Hamiltonian isometries of $X$ leaving the holomorphic symplectic form invariant.

- The flavour symmetry $T_t$ is a Hamiltonian isometry of $X$ transforming the holomorphic symplectic form with weight $+2$.

Under our assumptions, the Hamiltonian isometries $T_H$ have isolated fixed points that are identified with the massive supersymmetric vacua $\{\alpha\}$.

Let us now re-introduce the mass parameters $m, \epsilon$ in the sigma model description. A mass parameter $x = (m, \epsilon)$ is a deformation of the sigma model by the real moment maps $h_m, h_\epsilon : X \to \mathbb{R}$ for the corresponding Hamiltonian isometries of $X$,

$$\begin{aligned} h_m &= m \cdot \mu_{H,\mathbb{R}}, \\ h_\epsilon &= \epsilon \cdot \mu_{t,\mathbb{R}}, \end{aligned} \tag{27}$$

where $\mu_{H,\mathbb{R}}$, $\mu_{t,\mathbb{R}}$ now denote the moment maps for the $T = T_H \times T_t$ action on $X$. Provided $m, \epsilon$ are generic, the critical points are the massive vacua,

$$\text{Crit}(h_m) = \text{Crit}(h_m + h_\epsilon) = \{\alpha\}. \tag{28}$$

### 2.4.1 Algebraic Description

For many computations, it is convenient to introduce an algebraic description of the Higgs branch as a holomorphic symplectic quotient

$$X = \mu_{\mathbb{C}}^{-1}(0)^s / G_{\mathbb{C}}, \tag{29}$$

where $G_{\mathbb{C}}$ is the complexification of the gauge group and the superscript denotes a stability condition depending on the chamber $\mathfrak{C}_C$ containing $\zeta$.

From this perspective, $T_H$, $T_t$ combine with gradient flow for the associated moment maps to give an action of the complexification of $T_H$, $T_t$ on $X$ by complex isometries transforming the holomorphic symplectic form with weight $0$, $+2$ respectively. The vacua $\alpha$ are again the fixed points of these actions.

The algebraic description provides a convenient way to enumerate the collections of weights $\Phi_\alpha$ of the tangent spaces $T_\alpha X$, which will play an important part in this paper. First, the tangent bundle is constructed as the cohomology of the complex

$$0 \to \mathfrak{g}_{\mathbb{C}} \longrightarrow T^*R \longrightarrow \mathfrak{g}_{\mathbb{C}}^* \to 0, \tag{30}$$

restricted to the stable locus, where the first map is an infinitesimal complex gauge transformation and the second is the differential of the complex moment map.

The terms in this complex transform as representations of $G \times T_H \times T_t$. We introduce formal grading parameters $w = (s, v, t)$ such that a weight $\varrho = (\rho, \rho_H, \rho_t)$ is represented by a Laurent monomial $w^\varrho = s^\rho v^{\rho_H} t^{\rho_t}$. Recall from section 2.2 that fixed points $\alpha$ are in 1-1 correspondence with collections of weights $(\varrho_1, \ldots, \varrho_r)$ appearing in the weight decomposition of $T^*R$. Since the gauge components span $\mathfrak{h}^*$, we may solve the $r$ equations

$$w^{\varrho_i} = s^{\rho_i} v^{\rho_{H,i}} t^{\rho_{t,i}} = 1 \tag{31}$$

uniquely for the $r$ components of $s$. We denote the solution associated to a supersymmetric vacuum $\alpha$ by $s_\alpha$. The character of $T_\alpha X$ is obtained from that of the tangent complex by the substitution $s = s_\alpha$,

$$\begin{aligned}
\text{Ch}\, T_\alpha X &= \text{Ch}\, Q - \text{Ch}\, \mathfrak{g}_{\mathbb{C}} - t^{-2} \text{Ch}\, \mathfrak{g}_{\mathbb{C}}^* \Big|_{s=s_\alpha} \\
&= \sum_{\lambda \in \Phi_\alpha} v^{\lambda_H} t^{\lambda_t},
\end{aligned} \tag{32}$$

from which the weights $\lambda = (\lambda_H, \lambda_t) \in \Phi_\alpha$ can be determined.

### 2.4.2 Reproducing Chern-Simons Couplings

From a sigma model perspective, the supersymmetric Chern-Simons terms $\kappa_\alpha$, $\kappa_\alpha^C$ arise from classical contributions to the vector central charge given by the values of the moment maps at critical points,

$$\begin{aligned}
\kappa_\alpha(m, \zeta) &= h_m(\alpha), \\
\kappa_\alpha^C(\epsilon, \zeta) &= h_\epsilon(\alpha).
\end{aligned} \tag{33}$$

The remaining $\kappa_\alpha^H$, $\widetilde{\kappa}_\alpha$ arise from a 1-loop contribution from integrating out massive fluctuations around a critical point. To describe these contributions cleanly, we introduce some notation for tangent weights.

As above, let $\Phi_\alpha$ denote the collection of weights in the $T$ weight decomposition of $T_\alpha X$. We assume here that elements of $\Phi_\alpha$ are pair-wise linearly independent for all critical points $\alpha$, which is the GKM condition for the Hamiltonian $T$-action on $X$ [29,30]. This assumption is satisfied in the supersymmetric QED example and in more general abelian theories where $X$ is hyper-toric [31]. It is also satisfied for supersymmetric quiver gauge theories where $X$ is the cotangent bundle of a partial flag variety.

Introducing mass parameters $x = (m, \epsilon)$, there is a decomposition

$$\Phi_\alpha = \Phi_\alpha^+ \cup \Phi_\alpha^-, \tag{34}$$

where

$$\begin{aligned}
\Phi_\alpha^+ &= \{\, \lambda \in \Phi_\alpha \,|\, \lambda \cdot x > 0 \,\}, \\
\Phi_\alpha^- &= \{\, \lambda \in \Phi_\alpha \,|\, \lambda \cdot x < 0 \,\},
\end{aligned} \tag{35}$$

denote the collections of positive and negative weights. This decomposition depends only on the chamber $\mathfrak{C}$ containing $x$. We denote the corresponding decomposition of the tangent space by

$$T_\alpha X = N_\alpha^+ \oplus N_\alpha^-. \tag{36}$$

The form of this decomposition is constrained by the transformation properties of the holomorphic symplectic form. As the holomorphic symplectic form transforms with weight $+2$ under $T_t$, if $\lambda \in \Phi_\alpha^+$ then $\lambda^* \in \Phi_\alpha^-$, where we define $\lambda^* = -2e_t - \lambda$ and $e_t$ is the fundamental weight of $T_t$.

The remaining supersymmetric Chern-Simons terms $\kappa_\alpha^H$, $\widetilde{\kappa}_\alpha$ are obtained by integrating out the massive fluctuations corresponding to tangent directions at the critical point $\alpha$. The sign of the correction is correlated with the sign of the mass of a fluctuation, with the result

$$\begin{aligned}
\kappa_\alpha^H + \widetilde{\kappa}_\alpha &= \frac{1}{4} \left( \sum_{\lambda \in \Phi_\alpha^+} \lambda \otimes \lambda - \sum_{\lambda \in \Phi_\alpha^-} \lambda \otimes \lambda \right) \\
&= \frac{1}{2} \left( \sum_{\lambda \in \Phi_\alpha^-} \lambda \otimes e_t - \sum_{\lambda \in \Phi_\alpha^+} e_t \otimes \lambda \right) \\
&= \frac{1}{2} (\lambda_\alpha^- - \lambda_\alpha^+) \otimes e_t,
\end{aligned} \tag{37}$$

where

$$\lambda_\alpha^\pm = \sum_{\alpha \in \Phi_\alpha^\pm} \lambda \tag{38}$$

and we regard the sum $\kappa_\alpha^H + \widetilde{\kappa}_\alpha$ as a pairing $\Gamma \times \Gamma \to \mathbb{Z}$. In the second line, we re-arranged the sum using the transformation properties of the holomorphic symplectic form. Despite the factor of half, this defines an integer pairing due to contributions of pairs of weights $\lambda$, $\lambda^*$ with opposite sign. The result depends only on the chamber $\mathfrak{C}$ containing the mass parameters $x = (m, \epsilon)$.

### 2.4.3 Example

In supersymmetric QED, the Higgs branch is the holomorphic symplectic quotient obtained by imposing the real and complex moment map equations,

$$\sum_{\alpha=1}^N (|X_\alpha|^2 - |Y_\alpha|^2) = \zeta \qquad \sum_{\alpha=1}^N X_\alpha Y_\alpha = 0, \tag{39}$$

and dividing by $G = U(1)$ gauge transformations. The holomorphic symplectic form on $X$ is induced from $\Omega = \sum_\alpha dX_\alpha \wedge dY_\alpha$ under the quotient. In both chambers $\mathfrak{C}_C = \{\pm\zeta > 0\}$, we have

$$X = T^*\mathbb{CP}^{N-1}, \tag{40}$$

which is a symplectic resolution of the minimal nilpotent orbit in $\mathfrak{sl}(N,\mathbb{C})$. The two chambers are distinguished by the identification of the ample cone $\mathfrak{C}_C = \text{Pic}(X) \otimes_\mathbb{Z} \mathbb{R}_\pm$.

The flavour symmetry $T_H$ acts by Hamiltonian isometries $(X_\alpha, Y_\alpha) \rightarrow (e^{-i\theta_\alpha}X_\alpha, e^{+i\theta_\alpha}Y_\alpha)$, while $T_t$ transforms $(X_\alpha, Y_\alpha) \rightarrow (e^{i\theta}X_\alpha, e^{i\theta}Y_\alpha)$. There are $N$ fixed points $\{\alpha\}$ corresponding to the image of $X_\alpha = \sqrt{\zeta}$ when $\zeta > 0$ and $Y_\alpha = \sqrt{-\zeta}$ when $\zeta < 0$ under the quotient. They are the coordinate lines in the base $\mathbb{CP}^{N-1}$.

The mass parameters $(m_1, \ldots, m_N)$, $\epsilon$ act by Hamiltonian isometries induced by the transformations of the coordinates $X_\alpha, Y_\alpha$, with moment maps

$$
\begin{aligned}
h_m &= -\sum_{\alpha=1}^N m_\alpha(|X_\alpha|^2 - |Y_\alpha|^2), \\
h_\epsilon &= \epsilon \sum_{\alpha=1}^N (|X_\alpha|^2 + |Y_\alpha|^2).
\end{aligned}
\tag{41}
$$

The supersymmetric Chern-Simons couplings $\kappa_\alpha$, $\kappa_\alpha^C$ are recovered from the values of these moment maps at critical loci,

$$
\begin{aligned}
\kappa_\alpha(m, \zeta) &= h_m(\alpha) = -m_\alpha\zeta, \\
\kappa_\alpha^C(\epsilon, \zeta) &= h_\epsilon(\alpha) = \epsilon|\zeta|.
\end{aligned}
\tag{42}
$$

To describe the remaining supersymmetric Chern-Simons levels, let us fix the default chamber $\mathfrak{C}_C = \{\zeta > 0\}$ and determine the weight spaces $\Phi_\alpha$. We have

$$\text{Ch}\, Q - \text{Ch}\, \mathfrak{g}_\mathbb{C} - t^{-2}\text{Ch}\, \mathfrak{g}_\mathbb{C}^* = t^{-1}s^{-1}\sum_{\beta=1}^N v_\beta + t^{-1}s\sum_{\beta=1}^N v_\beta^{-1} - t^{-2} - 1. \tag{43}$$

The expectation value of $X_\alpha$ determines $s^{-1}v_\alpha t^{-1} = 1$, and thus

$$\text{Ch}\, T_\alpha X = \prod_{\beta \neq \alpha} \frac{v_\beta}{v_\alpha} + t^{-2}\prod_{\beta \neq \alpha} \frac{v_\alpha}{v_\beta}. \tag{44}$$

The tangent weights at a vacuum $\alpha$ are therefore

$$\Phi_\alpha = \{e_\beta - e_\alpha, \beta \neq \alpha\} \cup \{-2e_t - e_\beta + e_\alpha, \beta \neq \alpha\}, \tag{45}$$

coinciding with the tangent weights of $X = \mathbb{CP}^{N-1}$ at coordinate hyperplanes in the base.

Suppose $\epsilon = 0$ with the mass parameters in the default chamber $\mathfrak{C}_H = \{m_1 > m_2 > \cdots > m_N\}$, or turning on a small mass parameter $\epsilon$, the corresponding chamber $\mathfrak{C}$ where $|\epsilon| < |m_\alpha - m_\beta|$. Then the decomposition into positive and negative weights is

$$
\begin{aligned}
\Phi_\alpha^+ &= \{e_\beta - e_\alpha, \beta < \alpha\} \cup \{-2e_t - e_\beta + e_\alpha, \beta > \alpha\}, \\
\Phi_\alpha^- &= \{e_\beta - e_\alpha, \beta > \alpha\} \cup \{-2e_t - e_\beta + e_\alpha, \beta < \alpha\}.
\end{aligned}
\tag{46}
$$

We have

$$
\begin{aligned}
\lambda_\alpha^+ &= \sum_{\beta < \alpha}(e_\beta - e_\alpha) + \sum_{\beta > \alpha}(-2e_t - e_\beta + e_\alpha), \\
\lambda_\alpha^- &= \sum_{\beta > \alpha}(e_\beta - e_\alpha) + \sum_{\beta < \alpha}(-2e_t - e_\beta + e_\alpha),
\end{aligned}
\tag{47}
$$

and

$$\frac{1}{2}(\lambda_\alpha^- - \lambda_\alpha^+) = \sum_{\beta < \alpha}(e_\alpha - e_\beta) + \sum_{\beta > \alpha}(e_\beta - e_\alpha) + (N - 2\alpha + 1)e_t. \tag{48}$$

We therefore recover the remaining supersymmetric Chern-Simons couplings,

$$\kappa_\alpha^H = \left(\sum_{\beta < \alpha}(e_\alpha - e_\beta) + \sum_{\beta > \alpha}(e_\beta - e_\alpha)\right) \otimes e_t, \tag{49}$$
$$\widetilde{\kappa}_\alpha = (N - 2\alpha + 1)e_t \otimes e_t.$$

The supersymmetric Chern-Simons couplings in other chambers for the mass parameters $(m_1, \ldots, m_N)$, $\epsilon$ can be computed similarly.

## 2.5 Hyperplane Arrangements

We now re-visit the hyperplane arrangement in the space $\mathfrak{t}_H$ of mass parameters from the perspective of a massive sigma model on $X$.

First, a domain wall preserving $\mathcal{N} = (0, 2)$ supersymmetry in the deformed sigma model is a solution of the gradient flow equations on $X$ for the moment map $h_x = h_m + h_\epsilon$ connecting critical points $\alpha$, $\beta$. The tension of domain walls is again controlled by the vector central charge via the formula $|C_\alpha - C_\beta|$. From the perspective of a massive sigma model, this receives a classical contribution from $\kappa_\alpha$, $\kappa_\alpha^C$ and a 1-loop correction from $\kappa_\alpha^H$, $\widetilde{\kappa}_\alpha$, as detailed above.

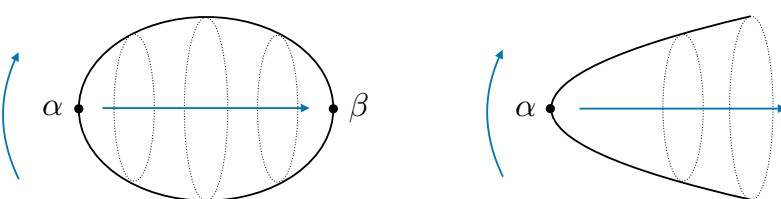

Figure 7: Families of gradient flows attached to supersymmetric vacua.

Solutions of the gradient flow equation on $X$ are not isolated. Acting with $T$ generates an $S^1$ family of gradient flows between vacua $\alpha$, $\beta$ that sweep out a compact curve $\mathbb{CP}^1 \subset X$. Furthermore, there are non-compact families of gradient flows extending out from a supersymmetric vacuum $\alpha$ to infinity. These possibilities are illustrated in figure 7.

These families of gradient flows generate the 1-skeleton of $X$, which can be represented by the GKM graph [29,30]. This is a representation of the supersymmetric vacua and families of domain walls connecting them, which consists of the following elements:

- Vertices labelled by supersymmetric vacua $\{\alpha\}$.

- Internal edges $\alpha \to \beta$ representing curves $\Sigma_\lambda \cong \mathbb{CP}^1$ labelled by a tangent weight $\lambda \in \Phi_\alpha^+ \cap (-\Phi_\beta^-)$.

- External edges $\alpha \leftarrow \infty$ and $\alpha \to \infty$ representing curves $\Sigma_\lambda \cong \mathbb{C}$ labelled by a tangent weight $\lambda \in (-\Phi_\alpha^-)$ and $\Phi_\alpha^+$ respectively.

The arrows represent directions of positive gradient flow for fixed parameters $x = (m, \epsilon)$. The GKM assumption ensures there is at most one internal edge connecting any pair of vertices. Supersymmetric QED with $N = 3$ is illustrated in figure 8.

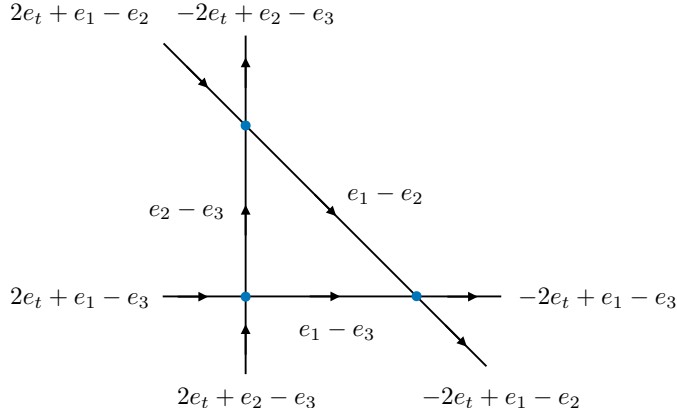

Figure 8: The GKM diagram for the Higgs branch $X = T^* \mathbb{CP}^{N_1}$ of supersymmetric QED with $N = 3$ in the default chamber $\mathfrak{C}_H = \{m_1 > m_2 > m_3\}$.

Now consider the hyperplane arrangement in the space $\mathfrak{t} \cong \mathfrak{t}_H \oplus \mathbb{R}$ of mass parameters $x = (m, \epsilon)$. The hyperplanes where the critical locus of the moment map $h_x$ is larger than the set of isolated critical points $\{\alpha\}$ take the form

$$\mathbb{W}_\lambda = \{\lambda \cdot x = 0\}, \tag{50}$$

where $\lambda$ is a tangent weight in the GKM graph. When such a hyperplane is crossed, the arrow on the corresponding edge flips orientation. We distinguish two types of hyperplane:

- If $\lambda$ labels an internal edge $\alpha \to \beta$, the hyperplane $\mathbb{W}_\lambda$ corresponds to a locus where a compact Higgs branch $\Sigma_\lambda \cong \mathbb{CP}^1$ opens up.

- If $\lambda$ labels an external edge connecting $\alpha$, the hyperplane $\mathbb{W}_\lambda$ corresponds to a locus where a non-compact Higgs branch $\Sigma_\lambda \cong \mathbb{C}$ opens up.

We are primarily interested in the limit $\epsilon \to 0$. Then gradient flows for the moment map $h_m$ correspond to domain walls preserving 2d $\mathcal{N} = (2, 2)$ supersymmetry whose tension receives a classical contribution only from $C_\alpha = h_m(\alpha)$. Furthermore, the hyperplane arrangement degenerates as follows. Recall that an incoming edge with weight $\lambda \in -\Phi_\alpha^-$ is always paired with an outgoing edge with weight $\lambda^* \in \Phi_\alpha^+$. In the limit $\epsilon \to 0$,

$$\mathbb{W}_\lambda = \mathbb{W}_{\lambda^*} = \{\lambda_H \cdot m = 0\} \tag{51}$$

and there is no distinction between internal and external hyperplanes. The hyperplanes are all of the form $\mathbb{W}_\lambda = \{C_\alpha = C_\beta\}$ and reproduce the hyperplane arrangement in the space $\mathfrak{t}_H$ of mass parameters discussed previously.

# 3 3d $\mathcal{N} = 4$ Theories on an Elliptic Curve

In this section we consider elementary aspects of 3d $\mathcal{N} = 4$ supersymmetric gauge theories on $\mathbb{R} \times E_\tau$, where $E_\tau$ is a complex torus with complex structure modulus $\tau$. We focus on the computation of supersymmetric ground states and their supersymmetric Berry connection over the space of mass parameters and flat connections on $E_\tau$ for flavour symmetries. We show that this leads naturally to algebraic constructions appearing in equivariant elliptic cohomology.

### 3.1 Supersymmetry

Let us first decompose the 3d $\mathcal{N} = 4$ supersymmetry algebra on $\mathbb{R} \times \mathbb{R}^2$ as a 1d $\mathcal{N} = (4,4)$ supersymmetric quantum mechanics on $\mathbb{R}$. The supersymmetry algebra can be written

$$
\begin{aligned}
\{Q_+^{A\dot{A}}, Q_+^{B\dot{B}}\} &= \epsilon^{AB}\epsilon^{\dot{A}\dot{B}} P \,, \\
\{Q_+^{A\dot{A}}, Q_-^{B\dot{B}}\} &= \epsilon^{AB}\epsilon^{\dot{A}\dot{B}} H + \epsilon^{AB}\widetilde{Z}^{\dot{A}\dot{B}} + \epsilon^{\dot{A}\dot{B}} Z^{AB} + C^{AB,\dot{A}\dot{B}} \,, \\
\{Q_-^{A\dot{A}}, Q_-^{B\dot{B}}\} &= \epsilon^{AB}\epsilon^{\dot{A}\dot{B}} \bar{P} \,,
\end{aligned}
\tag{52}
$$

where $H := P_{+-}$ is the Hamiltonian and $P := P_{++}$ and $\bar{P} := P_{--}$ become central charges from the perspective of supersymmetric quantum mechanics. The remaining scalar central charges are associated to global symmetries, $Z^{AB} = \zeta^{AB} \cdot J_C$ and $\widetilde{Z}^{\dot{A}\dot{B}} = m^{\dot{A}\dot{B}} \cdot J_H$, while $C^{AB,\dot{A}\dot{B}}$ arises from a vector central charge associated to domain walls and is bi-linear in the mass and FI parameters.

As in section 2, we set the complex parameters to zero, $m^{++} = 0$ and $t^{++} = 0$, and define $m := m^{+-}$ and $\zeta = \zeta^{+-}$. We then consider a 3d $\mathcal{N} = 2$ subalgebra commuting with $T_t := U(1)_H - U(1)_C$, which becomes a 1d $\mathcal{N} = (2,2)$ subalgebra in supersymmetric quantum mechanics. The supercharges are $Q_+^{++}$, $Q_-^{++}$, $Q_+^{--}$, $Q_-^{--}$ and the non-vanishing commutators are

$$
\begin{aligned}
\{Q_+^{++}, Q_+^{--}\} &= P \,, \\
\{Q_+^{++}, Q_-^{--}\} &= H + Z + C \,, \\
\{Q_+^{--}, Q_-^{++}\} &= H - Z + C \,, \\
\{Q_-^{++}, Q_-^{--}\} &= \bar{P} \,,
\end{aligned}
\tag{53}
$$

where we define $Z := Z^{+-} + \widetilde{Z}^{+-}$ and $C = C^{+-,+-}$. The Higgs and Coulomb flavour symmetries are now on the same footing with $Z = m \cdot J_H + \zeta \cdot J_C$. In the presence of a domain wall in the $x^{1,2}$-plane interpolating between massive vacua $\alpha$, $\beta$, $C = C_\alpha - C_\beta$ where $C_\alpha = \kappa_\alpha(m, \zeta)$ is the central charge function (21).

As the combination $T_t$ is now a flavour symmetry, it is possible to turn on a real mass parameter $\epsilon$. This deforms the central charges further to $Z = m \cdot J_H + \zeta \cdot J_C + \epsilon \cdot J_t$ and $C_\alpha$ has a more general form (19). In shorthand notation, $Z = x_f \cdot J_f$ and $C_\alpha = K_\alpha(x_f, x_f)$.

### 3.2 Reduction on Elliptic Curve

Let us now place such a theory on $E_\tau \times \mathbb{R}$ where $E_\tau$ is the elliptic curve with complex structure modulus $\tau = \tau_1 + i\tau_2$ and area $\tau_2 > 0$. This is implemented by forming the complex combination $x^1 + ix^2$ and making the identifications $(x^1, x^2) \sim (x^1 + 1, x^2)$ and $(x^1, x^2) \sim (x^1 + \tau^1, x^2 + \tau^2)$.

It is also convenient to introduce real coordinates $(s, t)$ and identify $s \sim s + 1$ and $t \sim t + 1$, such that $x^1 = s + \tau_1 t$ and $x^2 = \tau_2 t$. This gives a continuous isomorphism of groups $E_\tau \to S^1 \times S^1$ induced by the transformation $x^1 + ix^2 \mapsto (e^{2\pi i s}, e^{2\pi i t})$. These coordinates are illustrated in figure 9.

We impose R-R boundary conditions, which preserves the full supersymmetry. We can also now introduce background flat connections $A_f = (A_C, A_H, A_t)$ on $E_\tau$ for all 3d $\mathcal{N} = 2$ flavour symmetries. This background preserves the same supersymmetry algebra (53), but now

$$
\begin{aligned}
P &= +\frac{i}{\tau_2}(\partial_t - \tau \partial_s) + \frac{i}{\tau_2}\left(z_f \cdot J_f\right) \,, \\
\bar{P} &= -\frac{i}{\tau_2}(\partial_t - \bar{\tau} \partial_s) - \frac{i}{\tau_2}\left(\bar{z}_f \cdot J_f\right) \,,
\end{aligned}
\tag{54}
$$

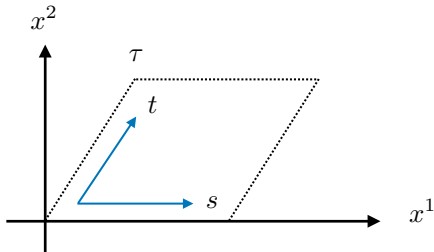

Figure 9: Coordinates on the elliptic curve $E_\tau$.

where

$$z_f := \oint A_{f,t} dt - \tau \oint A_{f,s} ds \tag{55}$$

and we again combine $z_f = (z_C, z_H, z_t)$ in shorthand notation. From the perspective of $\mathcal{N} = (2,2)$ supersymmetric quantum mechanics, the real parameters $x_f$ and complex flat connections $z_f$ combine as expectation values for the triplet of scalar fields in a background vector multiplet for the 3d $\mathcal{N} = 2$ flavour symmetry $T_f$.

It is often convenient to restrict attention to a sector with fixed KK momentum and flavour charge, whereupon $Z = x_f \cdot \gamma_f$ and

$$
\begin{aligned}
P &= +\frac{i}{\tau_2}(n - \tau m) + \frac{i}{\tau_2}\left(z_f \cdot \gamma_f\right), \\
\bar{P} &= -\frac{i}{\tau_2}(n - \bar{\tau} m) - \frac{i}{\tau_2}\left(\bar{z}_f \cdot \gamma_f\right),
\end{aligned}
\tag{56}
$$

where $(m,n) \in \mathbb{Z}^2$ are KK momenta conjugate to the coordinates $(s,t)$ and $\gamma_f \in \Gamma_f^\vee$ is a weight of the flavour symmetry $T_f$.

A global background gauge transformation $z_f \mapsto z_f + (\nu_f - \tau \mu_f)$ is specified by a pair of co-characters $\mu_f, \nu_f \in \Gamma_f$ associated to the cycles with coordinates $s, t$. In the sector with flavour weight $\gamma_f$, this can be absorbed by shifting the KK momenta $(m,n) \to (m - \gamma_f \cdot \mu_f, n - \gamma_f \cdot \nu_f)$.

### 3.2.1 Infinite-Dimensional Model

It is often useful to invoke an infinite-dimensional model for the effective $\mathcal{N} = (2,2)$ supersymmetric quantum mechanics.

Let us fix a generic FI parameter $\zeta$, such that in the absence of mass parameters the theory flows to a smooth sigma model onto the Higgs branch $X$ in flat space. Passing to $E_\tau \times \mathbb{R}$ and setting the background connection $z_C = 0$, the system is described by an $\mathcal{N} = (2,2)$ quantum mechanics whose target is the space of smooth maps

$$\mathcal{X} = \mathrm{Map}(E_\tau \to X), \tag{57}$$

which is an infinite-dimensional Kähler manifold. The kinetic terms involving derivatives along $E_\tau$ are obtained by coupling to a background vector multiplet for the $S^1 \times S^1$ symmetry of $\mathcal{X}$ induced by translations of the coordinates $(s,t)$.

The mass parameters $x = (m, \epsilon)$ and background connections $z = (z_H, z_t)$ are introduced by coupling to a background vector multiplet for the induced action of $T = T_H \times T_t$ on $\mathcal{X}$ and turning on expectation values for the scalar fields. The background flat connection $z_C$ is expected to induce a flat connection on the target $\mathcal{X}$ that deforms the action of the supercharges.

## 3.3 Supersymmetric Ground States

Let us now consider states of the supersymmetric quantum mechanics annihilated by all four of the supercharges generating the 1d $\mathcal{N} = (2,2)$ supersymmetry algebra (53). We refer to such states as supersymmetric ground states. Supersymmetric ground states are necessarily annihilated by $H + C$, $Z$, $P$, $\bar{P}$.

As usual in supersymmetric quantum mechanics, if the spectrum is gapped, it is convenient to introduce a cohomological description of supersymmetric ground states. For this purpose, we consider the supercharge

$$Q := Q_+^{++} + Q_+^{--}, \tag{58}$$

which satisfies $Q^2 = 2P$ and commutes with the central charges $Z$, $P$. We then restrict attention to states in the supersymmetric quantum mechanics annihilated by $Z$, $P$, which requires

$$
\begin{aligned}
x_f \cdot \gamma_f &= 0, \\
(n - m\tau) + z_f \cdot \gamma_f &= 0,
\end{aligned}
\tag{59}
$$

where $(m, n)$ are the KK momenta conjugate to the coordinates $(s, t)$ and $\gamma_f \in \Gamma_f^\vee$ is a weight of $T_f$. The supercharge $Q$ becomes a differential and its cohomology is an alternative description of supersymmetric ground states.

We can provide a heuristic picture of supersymmetric ground states using the infinite-dimensional quantum mechanics with target space $\mathcal{X} = \mathrm{Map}(E_\tau, X)$. For simplicity we set the flat connection for the topological symmetry to zero, $z_C = 0$. The supercharge $Q$ is then a twisted equivariant deformation of the de Rham differential on $\mathcal{X}$,

$$Q = e^{-h_x} \left( d + \iota_{V_z} \right) e^{h_x}, \tag{60}$$

where

- $h_x$ is the moment map for the Hamiltonian isometry generated by the mass parameters $m$, $\epsilon$. Here we abuse notation and write $h_x = h_m + h_\epsilon$ for the moment map on both $X$ and $\mathcal{X} = \mathrm{Map}(E_\tau, X)$.

- $V_z$ is a combination of the vector field $\partial_t - \tau \partial_s$ generating the $S^1 \times S^1$ group action on $\mathcal{X}$ induced by translations of the coordinates $(s, t)$ and the vector field generating the $S^1 \subset T$ action on $\mathcal{X}$ with parameters $z = (z_H, z_t)$.

This type of supercharge was already encountered in [32] and has recently been further studied in quiver supersymmetric quantum mechanics [33, 34]. It arises whenever an $\mathcal{N} = (2,2)$ supersymmetric quantum mechanics is coupled to background vector multiplets.

The supersymmetric ground states can be analysed by applying standard arguments in supersymmetric quantum mechanics to this infinite-dimensional model. First, we can scale the superpotential $h_x$ in order to localise supersymmetric ground states around $\mathrm{Crit}(h_x) \subset \mathcal{X}$. The supersymmetric ground states can then be obtained from the cohomology of the equivariant differential $d + \iota_{V_z}$ on $\mathrm{Crit}(h_x)$, which is the equivariant cohomology of $\mathrm{Crit}(h_x)$ localised at the equivariant parameter $z$.

Introducing a background connection $z_C$ would further deform the supercharge $Q$ by the addition of a background flat connection on $\mathcal{X}$. This does not materially change the outcome for supersymmetric ground states provided the FI parameter $\zeta$ is generic and therefore we set it to zero for simplicity.

This computation of supersymmetric ground states is clearly sensitive to the mass parameters $x \in \mathfrak{t}$ and will jump along the loci $W_\lambda \subset \mathfrak{t}$ introduced in section 2. We consider various cases in turn before presenting the general construction.

### 3.3.1 Generic Mass Parameters

First suppose the mass parameters lie in the complement of all of the loci $\mathbb{W}_\lambda = \{\lambda \cdot x = 0\}$. Then condition that supersymmetric ground states are annihilated by the central charge $Z = x_f \cdot J_f$ implies that they are uncharged under flavour symmetries, or $\gamma_f = 0$. In turn, $P = 0$ implies they have zero KK momentum, $n, m = 0$.

From the perspective of supersymmetric quantum mechanics to $\mathcal{X}$, the critical points of $h_x$ are constant maps $E_\tau \to \alpha$. In the limit that the coefficient in front of the superpotential is sent to infinity, there are normalisable perturbative ground states given by Gaussian wavefunctions localised at constant maps $E_\tau \to \alpha$, which may be chosen to be orthonormal. Since $h_x$ is the moment map for a Hamiltonian isometry of a Kähler manifold $\mathcal{X}$ and the spectrum is gapped, there are no instanton corrections and this is an exact description of supersymmetric ground states.

We can introduce another description of the supersymmetric ground states as follows. Let us fix a generic mass parameters in some chamber $\mathfrak{C} \subset \mathfrak{t}$. We then define

- $|\alpha\rangle_{\mathfrak{C}}$ is the supersymmetric ground state obtained from the path integral on $E_\tau \times \mathbb{R}_+$ with the supersymmetric vacuum $\alpha$ at infinity $x^3 \to +\infty$,

- $_{\mathfrak{C}}\langle\alpha|$ is the supersymmetric ground state obtained from the path integral on $E_\tau \times \mathbb{R}_-$ with the supersymmetric vacuum $\alpha$ at infinity $x^3 \to -\infty$.

These states are orthonormal,

$$_{\mathfrak{C}}\langle\alpha|\beta\rangle_{\mathfrak{C}} = \delta_{\alpha,\beta}, \tag{61}$$

which is interpreted as the partition function on $E_\tau \times \mathbb{R}$ with vacuum $\alpha$ at $x^3 \to -\infty$ and vacuum $\beta \to +\infty$. Note that the normalisation is independent of the potential background connection $z$. This basis of supersymmetric ground states depends on the chamber $\mathfrak{C}$ for the real mass parameters $x = (m, \epsilon)$.

### 3.3.2 Mass Parameters on Walls

Now consider supersymmetric ground states when the real mass parameters lie on a wall $\mathbb{W}_\lambda = \{\lambda \cdot x = 0\}$ for some weight $\lambda \in \Phi_\alpha$ of the tangent space at the vacuum $\alpha$.

Then, we claim that provided $\lambda \cdot z \notin \mathbb{Z} + \tau\mathbb{Z}$, there are again $N$ supersymmetric ground states of zero KK momentum and flavour charge, but whose properties now depend on whether $\lambda$ corresponds to an internal or external edge in the GKM diagram, as discussed in section 2. Thus there is a doubly-periodic array of distinguished points in the space of background flat connections $z$

$$\lambda \cdot z \in \mathbb{Z} + \tau\mathbb{Z}, \tag{62}$$

where supersymmetric ground states may carry KK momenta and flavour weight, and the nature of these points again depends on whether $\lambda$ is an internal or external edge.

We now prove this using our infinite-dimensional description of the effective $\mathcal{N} = (2, 2)$ quantum mechanics, considering external and internal edges in turn.

#### External Edge

If $\lambda \in \Phi_\alpha$ is an external edge, the critical locus of the superpotential is

$$\text{Crit}(h_x) = \{\gamma \neq \alpha\} \cup \text{Map}(E_\tau, \Sigma_\lambda), \tag{63}$$

where $\Sigma_\lambda \cong \mathbb{C}$. There are $N - 1$ ground states localised around constant maps $E_\tau \to \gamma$ with $\gamma \neq \alpha$, as in the discussion of generic mass parameters. However, the ground state associated to $\alpha$ is different. We must now consider the cohomology of the remaining differential

$Q = d + \iota_{V_z}$, which is the equivariant cohomology of the critical locus $\text{Map}(E_\tau, \Sigma_\lambda)$, localised at the background flat connection $z$.

Provided the background flat connection is not a distinguished point, $\lambda \cdot z \notin \mathbb{Z} + \tau\mathbb{Z}$, the vector field $V_z$ has a single fixed point corresponding to the constant map $E_\tau \to \alpha$, with associated supersymmetric ground state $|\alpha\rangle$. There is now some ambiguity in the normalisation as unitarity is lost in describing supersymmetric ground states cohomologically [35]. A natural choice is to define $|\alpha\rangle$ as the Poincaré dual of the equivariant fundamental class of the constant map $\{E_\tau \to \alpha\}$ inside the critical locus $\text{Map}(E_\tau, \Sigma_\lambda)$. This is normalised such that

$$
\begin{aligned}
\langle \alpha | \alpha \rangle &= \prod_{n,m \in \mathbb{Z}} (n + m\tau + \lambda \cdot z) \\
&= i\frac{\vartheta_1(\lambda \cdot z, \tau)}{\eta(\tau)},
\end{aligned}
\tag{64}
$$

where the first line is the equivariant Euler class of the normal bundle to the constant map $E_\tau \to \alpha$ inside $\text{Map}(E_\tau, \Sigma_\lambda)$. In the second line, we use zeta-function regularisation to define this in terms of the Jacobi theta function $\vartheta_1(z, \tau)$ and Dedekind eta function $\eta(\tau)$.

When the background flat connection satisfies $\lambda \cdot z \in \mathbb{Z} + \tau\mathbb{Z}$, the fixed locus of $V$ is non-compact, the supersymmetric quantum mechanics is not gapped and the cohomological description of supersymmetric ground states breaks down.

**Internal Edge**

If $\lambda \in \Phi_\alpha \cap (-\Phi_\beta)$ labels an internal edge connecting $\alpha$ and $\beta$, the critical locus of the real superpotential is

$$
\text{Crit}(h) = \{\gamma \neq \alpha, \beta\} \cup \text{Map}(E_\tau, \Sigma_\lambda),
\tag{65}
$$

where now $\Sigma_\lambda \cong \mathbb{CP}^1$. There are now $N - 2$ supersymmetric ground states corresponding to constant maps $E_\tau \to \gamma$ with $\gamma \neq \alpha, \beta$. However, for the supersymmetric ground states associated to $\alpha, \beta$, we must again consider the cohomology of the remaining differential $d + \iota_V$, which is the equivariant cohomology of the component $\text{Map}(E_\tau, \Sigma_\lambda)$ localised at the background flat connection $z$.

This component of the critical locus is compact, so the supersymmetric quantum mechanics is gapped and the cohomological construction of supersymmetric ground states is valid for any background flat connection $z$. Nevertheless, there are interesting phenomena at the loci where the fixed locus of $V_z$ does not consist of isolated points.

Provided $\lambda \cdot z \notin \mathbb{Z} + \tau\mathbb{Z}$, the vector field $V_z$ has isolated fixed points on $\text{Map}(E_\tau, \Sigma_\lambda)$ corresponding to the constant map $E_\tau \to \alpha$ and $E_\tau \to \beta$. There are therefore two supersymmetric ground states, which normalised such that

$$
\begin{aligned}
\langle \alpha | \alpha \rangle &= \prod_{n,m \in \mathbb{Z}} (n + m\tau + \lambda \cdot z) = i\frac{\vartheta_1(\lambda \cdot z, \tau)}{\eta(\tau)}, \\
\langle \beta | \beta \rangle &= \prod_{n,m \in \mathbb{Z}} (n + m\tau - \lambda \cdot z) = i\frac{\vartheta_1(-\lambda \cdot z, \tau)}{\eta(\tau)}, \\
\langle \alpha | \beta \rangle &= \langle \beta | \alpha \rangle = 0.
\end{aligned}
\tag{66}
$$

When $\lambda \cdot z \in \mathbb{Z} + \tau\mathbb{Z}$ the fixed locus of $V_z$ is non-isolated and the above supersymmetric ground states are not linearly independent. To find a linearly independent basis of supersymmetric ground states that extends across this locus one can, for example, pass to the linear combinations

$$
\begin{aligned}
|1\rangle &= -i\frac{\eta(\tau)}{\vartheta_1(\lambda \cdot z, \tau)}(|\alpha\rangle - |\beta\rangle), \\
|2\rangle &= \frac{1}{2}(|\alpha\rangle + |\beta\rangle),
\end{aligned}
\tag{67}
$$

which mix the contributions from the supersymmetric vacua $\alpha, \beta$.

### 3.3.3 Vanishing Mass Parameters

We may continue this process to construct supersymmetric ground states on the intersection series of loci $\mathbb{W}_\lambda, \mathbb{W}_{\lambda'}, \ldots$. Instead, we skip to the endpoint and consider the case of vanishing mass parameters $x = 0$, or equivalently the intersection of all hyperplanes $\mathbb{W}_\lambda$ with $\lambda$ running over all edges of the GKM diagram.

The real superpotential now vanishes and we must consider the equivariant cohomology of the whole $\mathcal{X} = \mathrm{Map}(E_\tau, X)$, localised at the background flat connection $z$. Provided the background flat connection is generic, now meaning $\lambda \cdot z \notin \mathbb{Z} + \tau\mathbb{Z}$ for all tangent weights, the vector field $V_z$ has only isolated fixed points corresponding to constant maps $E_\tau \to \alpha$. Following the discussion above, there are then $N$ supersymmetric ground states $|\alpha\rangle$, normalised such that

$$\langle\alpha|\beta\rangle = \prod_{\lambda \in \Phi_\alpha} i\frac{\vartheta_1(\lambda \cdot z, \tau)}{\eta(\tau)}\delta_{\alpha\beta}\,. \tag{68}$$

They are the equivariant fundamental classes of the constant maps $\{E_\tau \to \alpha\}$ inside $\mathcal{X}$. We will see below that these supersymmetric ground states play the role of the fixed point basis in $T$-equivariant elliptic cohomology of $X$.

If $\lambda \cdot z \in \mathbb{Z} + \tau\mathbb{Z}$, this construction breaks down. If $\lambda$ is an external edge of the GKM diagram, the supersymmetric quantum mechanics is not gapped. If $\lambda \in \Phi_\alpha^+ \cap (-\Phi_\beta^-)$ is an internal edge, we must take linear combinations corresponding to de Rham cohomology classes on $\mathrm{Map}(E_\tau, \Sigma_\lambda)$. These loci can of course further overlap leading to more intricate structures.

Finally, let us consider the relationship between the supersymmetric ground states $|\alpha\rangle_{\mathfrak{C}}$ for generic mass parameters in some chamber $\mathfrak{C}$ and the supersymmetric ground states $|\alpha\rangle$ at the origin. We have seen that in the limit $x \to 0$, the supersymmetric ground states $|\alpha\rangle_{\mathfrak{C}}$ are no longer appropriate. However, we claim that

$$
\begin{aligned}
&|\alpha\rangle_{\mathfrak{C}} \prod_{\lambda \in \Phi_\alpha^-} i\frac{\vartheta_1(\lambda \cdot z)}{\eta(\tau)} \to |\alpha\rangle\,, \\
&\prod_{\lambda \in \Phi_\alpha^+} i\frac{\vartheta_a(\lambda \cdot z)}{\eta(\tau)}\,{}_{\mathfrak{C}}\langle\alpha| \to \langle\alpha|\,,
\end{aligned} \tag{69}
$$

with the understanding that this holds for computations preserving the supercharge $Q$ used in the cohomological construction of supersymmetric ground states. This is compatible with the normalisations set out above.

We will discuss this relation further using boundary conditions and supersymmetric localisation in section 5. In that context, computations involving supersymmetric ground states are independent of the real mass parameters, so it is convenient to write this as an equality

$$
\begin{aligned}
&|\alpha\rangle_{\mathfrak{C}} \prod_{\lambda \in \Phi_\alpha^-} i\frac{\vartheta_1(\lambda \cdot z)}{\eta(\tau)} = |\alpha\rangle\,, \\
&\prod_{\lambda \in \Phi_\alpha^+} i\frac{\vartheta_a(\lambda \cdot z)}{\eta(\tau)}\,{}_{\mathfrak{C}}\langle\alpha| = \langle\alpha|\,.
\end{aligned} \tag{70}
$$

However, as we discuss further in section 7 it is more accurate to say that the sets of supersymmetric ground states are related by the action of a Janus interface interpolating between vanishing mass parameters and mass parameters in a chamber $\mathfrak{C}$.

### 3.3.4 Example

In supersymmetric QED, there are supersymmetric vacua $\alpha = 1,\ldots,N$, mass parameters $m_1,\ldots,m_N, \epsilon$ and background flat connections $z_1,\ldots,z_N, z_t$. Let us choose the default chambers $\mathfrak{C}_C = \{\zeta > 0\}$ and $\mathfrak{C} = \{m_1 > \cdots > m_N, |\epsilon| < |m_\alpha - m_\beta|\}$. Then

$$|\alpha\rangle_{\mathfrak{C}} \prod_{\beta > \alpha} i\frac{\vartheta_1(z_\beta - z_\alpha)}{\eta(\tau)} \prod_{\beta < \alpha} i\frac{\vartheta_1(-2z_t - z_\beta + z_\alpha)}{\eta(\tau)} = |\alpha\rangle,$$
$$_{\mathfrak{C}}\langle\alpha| \prod_{\beta < \alpha} i\frac{\vartheta_1(z_\beta - z_\alpha)}{\eta(\tau)} \prod_{\beta > \alpha} i\frac{\vartheta_1(-2z_t - z_\beta + z_\alpha)}{\eta(\tau)} = \langle\alpha|, \tag{71}$$

and

$$\langle\alpha|\beta\rangle = \delta_{\alpha\beta} \prod_{\beta \neq \alpha} i\frac{\vartheta_1(z_\beta - z_\alpha)}{\eta(\tau)} i\frac{\vartheta_1(-2z_t - z_\beta + z_\alpha)}{\eta(\tau)}. \tag{72}$$

### 3.4 Supersymmetric Berry Connection

A more systematic approach to supersymmetric ground states and their dependence on the background parameters is via the supersymmetric Berry connection. The form of the Berry connection is dictated by the fact that the mass parameters $x_f = (\zeta, m, \epsilon)$ and background connections $z_f = (z_C, z_H, z_t)$ transform as the real and complex scalar components of 1d $\mathcal{N} = (2, 2)$ vector multiplets [9–11].

Let us denote the number of supersymmetric ground states by $N$. Then there is a Berry connection on the rank-$N$ vector bundle of supersymmetric ground states over $\mathfrak{t}_f \times E_{T_f}$. Here $\mathfrak{t}_f$ parametrises the real parameters $x_f$ and

$$E_{T_f} := \Gamma_f \otimes_{\mathbb{Z}} E_\tau \tag{73}$$

is the complex torus parametrising background flat connections $z_f$ modulo gauge transformations $z_f \to z_f + (\nu_f + \tau\mu_f)$.

The Berry connection is enhanced to a solution of the generalised $U(N)$ Bogomolny equations on $\mathfrak{t}_f \times E_{T_f}$, which is perhaps best described as a rank-$N$ hyper-holomorphic connection on $\mathfrak{t}_f \times \mathfrak{t}_f \times E_{T_f}$ that is invariant under translations in the additional $\mathfrak{t}_f$ direction.

Concretely, this involves a pair $(A, \Phi)$ consisting of

- a connection $A$ on a principal $U(N)$ bundle $P$ on $\mathfrak{t}_f \times E_{T_f}$,

- a $\mathfrak{t}_f^\vee$-valued section $\Phi$ of $\mathrm{Ad}(P)$, which arises from the components of the hyper-holomorphic connection in the additional directions.

The asymptotic behaviour in the non-compact $\mathfrak{t}_f$-directions is that of a generalised doubly-periodic abelian monopole whose charges are controlled by the effective supersymmetric Chern-Simons couplings [11]. In an asymptotic region $|x_f| \to \infty$ in some chamber $\mathfrak{C}_f$, $P$ splits as a direct sum of principal $U(1)$ bundles $P_\alpha$ and the solution is abelian $(A_\alpha, \Phi_\alpha)$ with leading growth

$$\Phi_\alpha \to \frac{2\pi}{\tau_2} K_\alpha(x_f) + \cdots, \tag{74}$$

where we regard the effective $\mathcal{N} = 2$ supersymmetric Chern-Simons couplings in the chamber $\mathfrak{C}_f$ as a linear map $K_\alpha : \Gamma_f \to \Gamma_f^\vee$. In particular, contracting with the real parameters shows that

$$x_f \cdot \Phi_\alpha \to C_\alpha + \ldots, \tag{75}$$

where $C_\alpha = K_\alpha(x_f, x_f)$ is the vector central charge function in section 2.3. This type of boundary condition when $\mathrm{rk}\, T_f = 1$ was introduced in the construction of doubly-periodic monopole solutions in [36–38].

Let us fix parameters $(\zeta, z_C)$ and focus on the supersymmetric Berry connection for the remaining parameters $(x, z) \in \mathfrak{t} \times E_T$. The analysis of supersymmetric ground states in the previous section indicates the supersymmetric Berry connection will have important features at real co-dimension three loci $\mathbb{S}_\lambda \subset \mathfrak{t} \times E_T$ labelled by tangent weights $\lambda \in \Phi_\alpha$ and defined by

$$\begin{aligned} \lambda \cdot m &= 0, \\ \lambda \cdot z &\in \mathbb{Z} + \tau\mathbb{Z}. \end{aligned} \qquad (76)$$

Based on the considerations of the previous section and the explicit form of the supersymmetric Berry connections for supersymmetric quantum mechanics with targets $\Sigma_\lambda \cong \mathbb{CP}^1$ and $\Sigma_\lambda \cong \mathbb{C}$, we expect the following behaviour:

- If $\lambda \in \Phi_\alpha$ corresponds to an external edge of the GKM diagram, there is a singular 't Hooft monopole configuration centred on $\mathbb{S}_\lambda$, where the spectrum of the supersymmetric quantum mechanics fails to be gapped.

- If $\lambda \in \Phi_\alpha \cap (-\Phi_\beta)$ corresponds to an internal edge of the GKM diagram, there is a smooth $SU(2)$ monopole configuration centred on $\mathbb{S}_\lambda$, which mixes the supersymmetric ground states associated to $\alpha, \beta$.

These loci can intersect in higher co-dimension leading to more intricate configurations, for example smooth $SU(k)$ monopole configurations with $1 < k \leq N$.

We will not attempt a full analysis of the Berry connection and its connection to doubly-periodic monopoles here. Instead, we will focus on a particular algebraic construction that makes direct contact with equivariant elliptic cohomology.

## 3.5 Spectral Data

Let us now consider the supersymmetric Berry connection using the picture of supersymmetric ground states as elements of the cohomology of the supercharge $Q$. We consider the real and complex parameters in turn:

- The supercharge depends on the real parameters $x_f$ such that

$$\partial_{x_f} Q = -[\Phi, Q], \qquad (77)$$

where $\Phi \in \mathfrak{t}_f^\vee$ are hermitian operators independent of $x_f$. They play the role analogous to a moment map for the symmetry $T_f$ in the supersymmetric quantum mechanics. This descends to a complexified flat connection $\mathcal{D}_{x_f} = D_{x_f} + \Phi$ for supersymmetric ground states along $\mathfrak{t}_f$.

- The supercharge $Q$ depends holomorphically on the background connection $z_f$, so the anti-holomorphic derivative commutes with $Q$ and descends to a holomorphic Berry connection $\mathcal{D}_{\bar{z}_f}$ on supersymmetric ground states along $E_{T_f}$.

In the language of [39], this is consistent with the effective quantum mechanics being of BAA-type. The Berry connections commute,

$$[\mathcal{D}_{\bar{z}_f}, \mathcal{D}_{x_f}] = 0, \qquad (78)$$

which form part of the generalised Bogomolny equations. Thus the Berry connection $\mathcal{D}_{\bar{z}_f}$ determines the structure of a rank $N$ holomorphic vector bundle $\mathcal{E}$ on each slice $\{x_f\} \times E_{T_f}$, that varies in a covariantly constant way with the mass parameters $x_f$.

The asymptotic boundary conditions imply that for parameters $x_f$ in a given chamber $\mathfrak{C}_f \subset \mathfrak{t}_f$ in the space of mass and FI parameters the holomorphic vector bundle admits a holomorphic filtration

$$0 \subset \mathcal{E}_{\alpha_1} \subset \mathcal{E}_{\alpha_2} \subset \cdots \subset \mathcal{E}_{\alpha_N} = \mathcal{E}, \tag{79}$$

where $\mathcal{E}_{\alpha_i}$ is a rank $i$ holomorphic subbundle labelled by a vacuum $\alpha_i$, generated by holomorphic sections of $\mathcal{E}$ with a decay rate fixed by $C_{\alpha_i}$. This follows from [41] for the doubly periodic monopoles we consider, following classic analogous results for monopoles in $\mathbb{R}^3$ [42–44].

One can take the associated graded bundle:

$$\mathcal{G}(\mathcal{E}) = \bigoplus_\alpha \mathcal{L}_\alpha, \tag{80}$$

which splits by construction as a sum of holomorphic line bundles $\mathcal{L}_{\alpha_i} \cong \mathcal{E}_{\alpha_i}/\mathcal{E}_{\alpha_{i-1}}$. A section of $\mathcal{L}_\alpha$ transforms with factor of automorphy

$$s_\alpha(z_f + \nu_f + \tau \mu_f) = e^{-i\theta_\alpha(z_f,\mu_f)} s_\alpha(z_f), \tag{81}$$

where

$$\theta_\alpha(z_f,\mu_f) = 2\pi \left( K_\alpha(z_f,\mu_f) + K_\alpha(\mu_f,z_f) + \tau K_\alpha(\mu_f,\mu_f) \right). \tag{82}$$

The supersymmetric ground states $|\alpha\rangle_\mathfrak{C}$ introduced above will transform as sections of the holomorphic line bundles $\mathcal{L}_\alpha$. Note that the factor of automorphy depends on the chamber $\mathfrak{C}_f \subset \mathfrak{t}_f$ through the Chern-Simons couplings $K_\alpha$.

There are a number of algebraic approaches to the generalised Bogomolny equations obeyed by the supersymmetric Berry connection. For example, the scattering method would study the scattering problem for $\mathcal{D}_{x_f}$ and the associated spectral data. This would generalise the classical scattering methods [40, 42, 45] and correspond to the $z$-spectral data for doubly periodic monopoles [37].

### 3.5.1 Elliptic Cohomology Variety

Here we present an alternative spectral construction that makes direct contact with equivariant elliptic cohomology. Let us again fix parameters $(\zeta, z_C)$ and focus on the supersymmetric Berry connection for the remaining parameters $(x, z) \in \mathfrak{t} \times E_T$. We denote the chamber containing the mass parameters $x$ by $\mathfrak{C} \subset \mathfrak{t}$.

In place of the scattering problem for $\mathcal{D}_x$ across the origin of the mass parameter space, recall that our analysis of supersymmetric ground states using the infinite-dimensional supersymmetric quantum mechanics model showed that

$$|\alpha\rangle_\mathfrak{C} \prod_{\lambda \in \Phi_\alpha^-} i \frac{\vartheta_1(\lambda \cdot z)}{\eta(\tau)} \to |\alpha\rangle \tag{83}$$

as $x \to 0$ in each chamber $\mathfrak{C}$, where $|\alpha\rangle$ denotes the common set of supersymmetric ground states at the origin of $0 \in \mathfrak{t}$ of the space of mass parameters.

Let us first check that this is consistent with the supersymmetric Berry connection. Recall that the supersymmetric ground states $|\alpha\rangle_\mathfrak{C}$ transform as sections of the holomorphic line bundles $\mathcal{L}_\alpha$ whose factors of automorphy are fixed by the Chern-Simon levels $K_\alpha$. The supersymmetric ground states $|\alpha\rangle$ will then transform as sections of holomorphic line bundles $\mathcal{L}'_\alpha$ whose factors of automorphy are shifted by the additional Jacobi theta functions

$$\prod_{\lambda \in \Phi_\alpha^-} i \frac{\vartheta_1(\lambda \cdot z)}{\eta(\tau)}. \tag{84}$$

This is equivalent to shifting the supersymmetric Chern-Simons couplings as follows,

$$
\begin{aligned}
K'_\alpha &= K_\alpha + \frac{1}{2} \sum_{\lambda \in \Phi_\alpha^-} \lambda \otimes \lambda \\
&= \kappa_\alpha + \kappa_\alpha^C + \frac{1}{4} \sum_{\lambda \in \Phi_\alpha} \lambda \otimes \lambda,
\end{aligned}
\tag{85}
$$

where in the second line we have used (37) for the supersymmetric Chern-Simons levels $\kappa_\alpha^H$, $\widetilde{\kappa}_\alpha$. Since $\kappa_\alpha$, $\kappa_\alpha^C$ are independent of the chamber $\mathfrak{C}$ for the mass parameters, the factors of automorphy of $|\alpha\rangle$ have the same property and therefore (83) is consistent with the supersymmetric Berry connection.

With this observation in hand, the supersymmetric ground states at the origin $0 \in \mathfrak{t}$ of the mass parameter space have a remarkable property:

- The holomorphic line bundles $\mathcal{L}'_\alpha$, $\mathcal{L}'_\beta$ are isomorphic on restriction to the locus $\lambda \cdot z \in \mathbb{Z} + \tau\mathbb{Z}$ where the weight $\lambda \in \Phi_\alpha \cap (-\Phi_\beta)$ labels an internal edge of the GKM diagram of $X$.

We provide a detailed argument for this result in appendix A. Concretely, the factors of automorphy defined by the shifted Chern-Simons couplings $K'_\alpha$, $K'_\beta$ are equivalent (in a way made precise in the appendix) on restriction to the locus $\lambda \cdot z \in \mathbb{Z} + \tau\mathbb{Z}$ in the space of background flat connections $E_{T_f}$.

This means the collection of holomorphic line bundles $\mathcal{L}'_\alpha$ on the space of background flat connections $E_{T_f}$ is equivalent to a single holomorphic line bundle on an $N$-sheeted cover

$$
E_T(X) := \left( \bigsqcup_\alpha E_{T_f}^{(\alpha)} \right) / \Delta,
\tag{86}
$$

where:

- $E_{T_f}^{(\alpha)} \cong \Gamma_f \otimes_{\mathbb{Z}} E_\tau$ are $N$ copies of the torus of background flat connections for the full flavour symmetry $T_f = T_C \times T$ associated to the supersymmetric vacua $\alpha$.

- $\Delta$ identifies the copies $E_{T_f}^{(\alpha)}$ and $E_{T_f}^{(\beta)}$ at points $\lambda \cdot z \in \mathbb{Z} + \tau\mathbb{Z}$ where $\lambda \in \Phi_\alpha \cap (-\Phi_\beta)$ labels an internal edge of the GKM diagram.

This is the extended $T$-equivariant elliptic cohomology variety[2] of $X$ [1–4]. Note that copies of the space of flat connections are only identified along the components parametrising flat connections for the non-topological flavour symmetry $T \subset T_f$. It is therefore sometimes convenient to consider the non-extended equivariant elliptic cohomology variety by removing the factors of $E_{T_C}$, which is denoted by $\mathrm{Ell}_T(X)$.

More generally, on a generic face of the hyperplane arrangement in the space of mass parameters, the holomorphic line bundles associated to supersymmetric ground states combine to a section of a line bundle on the equivariant elliptic cohomology variety

$$
E_T(X^{T_m}),
\tag{87}
$$

where the $X^{T_m}$ is the fixed locus of the symmetry $T_m \subset T$ generated by the mass parameters $m$. In particular, if $m$ lies in a chamber of the hyperplane arrangement and $X^{T_m} = \{\alpha\}$, then $E_T(\{\alpha\})$ consists of $N$ independent copies of $E_T$ without identifications.

---

[2]Note that in general, the elliptic cohomology of a variety $X$ is a scheme. However, we assume the GKM property which implies that $E_T(X)$ is in fact a variety [8].

We could regard the collection of varieties $E_T(X^{T_m})$ as $m \in \mathfrak{t}$ varies over the space of mass parameters together with the holomorphic line bundles generated by supersymmetric ground states as as a kind of spectral data for the supersymmetric Berry connection on $\mathfrak{t} \times E_T$. It would be interesting to pin down its relation to the usual spectral data associated to the generalised Bogomolny equations, for example using the scattering method.

### 3.5.2 Gauge Theory Picture

There is an another description of the elliptic cohomology variety from the perspective of supersymmetric gauge theory, without passing to a sigma model on $X$.

We first imagine the un-gauged theory with target space $T^*R$ and regard $G$ as an additional flavour symmetry with real mass parameter $\sigma$ and background flat connection specified by $u$. In this case, the parameter space of background flat connections is

$$E_{T_f} \times E_G, \tag{88}$$

where

$$E_G = (E_\tau \otimes_{\mathbb{R}} \mathfrak{h})/W \tag{89}$$

parametrises background flat connections for $G$. The coordinates on the latter are Weyl-invariant functions of the coordinates $u$. For generic mass parameters and flat connections there is a single supersymmetric ground state corresponding to the fixed point at the origin of $T^*R$. The elliptic cohomology variety as constructed above is $E_{T_f} \times E_G$.

If we now fix a generic FI parameter $\zeta$ and gauge the symmetry $G$, recall that there are $N$ supersymmetric vacua $\alpha$ in flat space labelled by sets of weights $\{\varrho_1, \dots, \varrho_r\}$ of $G \times T$ satisfying conditions in section 2.2. This fixes the components of the real vector multiplet scalar in a supersymmetric vacuum $\alpha$ via the equations

$$\rho_a \cdot \sigma + \rho_{H,a} \cdot x = 0, \tag{90}$$

for $a = 1, \dots r$. Similarly, in the effective supersymmetric quantum mechanics on $\mathbb{R} \times E_\tau$ this fixes the gauge holonomy in a supersymmetric ground state, up to gauge transformations, via

$$\rho_a \cdot u + \rho_{H,a} \cdot z_H + \rho_{t,a} \cdot z_t \in \mathbb{Z} + \tau \mathbb{Z}. \tag{91}$$

The set of $N$ solutions modulo gauge transformations, $u_\alpha(z)$, generate an $N$-sheeted cover of $E_T$ as the background flat connections $z$ is varied. Trivially including the flat connection for the topological symmetry, this becomes an $N$-sheeted cover of $E_{T_f}$. This gives a construction of the extended equivariant elliptic cohomology variety

$$c : E_T(X) \hookrightarrow E_{T_f} \times E_G. \tag{92}$$

In this construction, the coordinates $u$ are identified with the elliptic Chern roots. This perspective on the elliptic cohomology variety will be useful in our discussion of Dirichlet boundary conditions in section 4.

### 3.5.3 Example

Let us consider supersymmetric QED with $N$ flavours in the default chamber $\mathfrak{C}_C = \{\zeta > 0\}$. There are background flat connections $z_f = (z_C, z_1, \dots, z_N, z_t)$ for the flavour symmetries $T_f = T_C \times T_H \times T_t$. They are subject to $\sum_\alpha z_\alpha \in \mathbb{Z} + \tau \mathbb{Z}$.

The supersymmetric ground states $|\alpha\rangle$ transform under background gauge transformations with factors of automorphy determined by the shifted levels

$$
\begin{aligned}
K'_\alpha =\ & -e_\alpha \otimes e_C + e_t \otimes e_C \\
& + \frac{1}{4} \sum_{\gamma \neq \alpha} (e_\gamma - e_\alpha) \otimes (e_\gamma - e_\alpha) \\
& + \frac{1}{4} \sum_{\gamma \neq \alpha} (-2e_t - e_\gamma + e_\alpha)(-2e_t - e_\gamma + e_\alpha).
\end{aligned}
\tag{93}
$$

If we restrict to background flat connections with $z_\alpha - z_\beta \in \mathbb{Z} + \tau \mathbb{Z}$, it is straightforward to check that $\theta_\alpha - \theta_\beta \in \mathbb{Z}$ and therefore the factors of automorphy of the supersymmetric ground states $|\alpha\rangle$ and $|\beta\rangle$ coincide.

The spectral curve $E_T(X)$ is therefore constructed from $N$ identical copies $E_{T_f}^{(\alpha)}$ of the space of background flat connections parametrised by $z_f = (z_C, z_1, \ldots, z_N, z_t)$. The copies $E_{T_f}^{(\alpha)}, E_{T_f}^{(\beta)}$ are identified along the loci $z_\alpha - z_\beta \in \mathbb{Z} + \tau \mathbb{Z}$ for all distinct pairs, say $\alpha < \beta$.

# 4 Boundary Amplitudes

We now consider boundary conditions preserving at least $\mathcal{N} = (0, 2)$ supersymmetry. Building on the description of boundary conditions in Rozansky-Witten theory [46–48], the study of $\mathcal{N} = (2, 2)$ boundary conditions in supersymmetric gauge theories was initiated in [15]. Various aspects of such boundary conditions have been further studied in [20, 35, 49, 50, 52]. $\mathcal{N} = (0, 2)$ boundary conditions were studied in [16–18, 53].

To a boundary condition $B$ preserving the flavour symmetry $T_f$, we will associate a state $|B\rangle$ in the supersymmetric quantum mechanics on $\mathbb{R} \times E_\tau$ studied in section 3. We will study the boundary amplitudes $\langle B|\alpha\rangle$ formed from the overlap with supersymmetric ground states and how they transform under large background gauge transformations on $E_\tau$ according to the boundary 't Hooft anomalies of $B$. We show that if a boundary condition is compatible with real mass parameters $m$, the collection of boundary amplitudes assemble into a section of a holomorphic line bundle on the elliptic cohomology variety $\mathrm{Ell}_T(X^{T_m})$, focussing on the cases where the mass parameters are zero or generic. In this way, we associate equivariant elliptic cohomology classes to boundary conditions.

## 4.1 Assumptions

We consider boundary conditions preserving at least 2d $\mathcal{N} = (0, 2)$ supersymmetry in the $x^{1,2}$-plane, generated by $Q_+^{++}$, $Q_+^{--}$. In many cases, they will preserve a larger $\mathcal{N} = (2, 2)$ supersymmetry generated by $Q_+^{++}, Q_+^{--}, Q_-^{+-}, Q_-^{-+}$. All such boundary conditions preserve the combination $Q = Q_+^{++} + Q_+^{--}$ and are compatible with cohomological construction of supersymmetric ground states introduced in section 3.

For $\mathcal{N} = (2, 2)$ boundary conditions, we require that they preserve a boundary vector and axial R-symmetry $U(1)_V \times U(1)_A$ and at least a boundary flavour symmetry $T_H \times T_C$. For $\mathcal{N} = (0, 2)$ boundary conditions, we require a boundary R-symmetry and at least a boundary flavour symmetry $T_f = T_C \times T_H \times T_L$. In the case of an $\mathcal{N} = (2, 2)$ boundary condition,

$$
T_L := U(1)_V - U(1)_A,
\tag{94}
$$

which is twice the left-moving R-symmetry.

The boundary vector and axial R-symmetry may be the same as the bulk R-symmetry $U(1)_H \times U(1)_C$ and hence $T_L = T_t$. However, the bulk R-symmetry may be spontaneously broken at the boundary, but a linear combination of the bulk R-symmetries and flavour symmetries is preserved and becomes $U(1)_V \times U(1)_A$. When this happens, we will draw attention to this distinction, but will abuse notation and still denote $T_f = T_C \times T_H \times T_L$.

We will also encounter boundary conditions where a mixed gauge-flavour anomaly breaks some subgroup of the above symmetries, and deal with this subtlety as it arises in the article. We will see this problem is immaterial when passing to the boundary amplitudes associated to the boundary condition.

A boundary condition preserving $T_H \times T_C$ may or may not be compatible with turning on the associated real mass and FI parameters. In this section, we exclusively set $\epsilon = 0$ and denote the chambers of the hyperplane arrangements for the remaining FI and mass parameters $\zeta, m$ by $\mathfrak{C}_C, \mathfrak{C}_H$. If a boundary condition is compatible with FI and mass parameters on a face of the hyperplane arrangement, it will be compatible with all such parameters on that face. We then say the boundary condition is compatible with that face. We mostly consider boundary conditions compatible with mass and FI parameters in given chambers of the hyperplane arrangements.

### 4.1.1 Higgs Branch Image

An important characteristic of a boundary condition is the Higgs branch image, which is a rough description of the boundary condition in the regime where the bulk gauge theory flows to a sigma model on $X$. A $\mathcal{N} = (0, 2)$ boundary condition satisfying the conditions above has support on a Kähler sub-manifold in $X$ invariant under $T$. For a $\mathcal{N} = (2, 2)$ boundary condition, the additional supersymmetry ensures the support is a holomorphic Lagrangian in $X$.

The compatibility with FI and mass parameters can be neatly understood from this perspective. First, compatibility with FI parameters in a fixed chamber $\mathfrak{C}_C$ is necessary for the boundary condition to preserve supersymmetry and define a reasonable boundary condition in a regime where the bulk gauge theory flows to a sigma model to $X$. Second, compatibility with mass parameters in a chamber $\mathfrak{C}_H$ requires that

- a right boundary condition on $x^3 \leq 0$ has support $S \subset \bigcup_\alpha X_\alpha^-$,

- a left boundary condition on $x^3 \geq 0$ has support $S \subset \bigcup_\alpha X_\alpha^+$.

where $X_\alpha^\pm$ denotes the attracting and repelling sets of the critical point $\alpha$ generated by a positive gradient flow for the moment map $h_m : X \to \mathbb{R}$ for all mass parameters $m \in \mathfrak{C}_H$.

The origin of the latter characterisation is that the BPS equations for the supercharges $Q_+^{++}$, $Q_+^{--}$ generating the $\mathcal{N} = (0, 2)$ supersymmetry algebra are inverse gradient flow for the moment map $h_m$ in the $x^3$-direction [15]. With our conventions, the moment map $h_m$ decreases as $x^3 \to \infty$, and increases as $x^3 \to -\infty$.

### 4.1.2 Anomalies

Boundary conditions are subject to mixed 't Hooft anomalies for the R-symmetries $U(1)_V$, $U(1)_A$ and flavour symmetries $T_C$, $T_H$, which are of paramount important in the presence of background connections on $E_\tau$. They may also suffer from gauge anomalies.

We keep track of boundary 't Hooft anomalies using an anomaly polynomial [16]. The anomaly polynomial of an $\mathcal{N} = (0, 2)$ boundary condition is bilinear in the curvatures $\mathbf{f}_V$, $\mathbf{f}_A$, $\mathbf{f}_C$, $\mathbf{f}_H$, associated to $U(1)_V$, $U(1)_A$, $T_C$, $T_H$. If the boundary condition preserves a boundary gauge symmetry, it may also depend on an associated curvature $\mathbf{f}$.

The computation of boundary amplitudes on $E_\tau$ will yield elliptic genera that involve a background for the left-moving $R$ symmetry $T_L$. They will therefore only detect boundary

anomalies of $T_L$, rather than those of the vector and axial R-symmetries separately. With this in mind, we only turn on a field strength $\mathbf{f}_L$, which may be implemented in the anomaly polynomial by substituting $\mathbf{f}_V \rightsquigarrow \mathbf{f}_L$ and $\mathbf{f}_A \rightsquigarrow -\mathbf{f}_L$.

We therefore consider boundary anomalies for $T_C \times T_H \times T_L$. We will later encounter Neumann boundary conditions where the gauge anomaly does not vanish, and also Dirichlet boundary conditions where the gauge symmetry becomes a boundary flavour symmetry, which we treat as they arise. Putting aside these cases for now, the boundary anomaly polynomial of an $\mathcal{N} = (0, 2)$ boundary condition takes the form

$$\mathcal{P} = K(\mathbf{f}_f, \mathbf{f}_f), \tag{95}$$

where we have introduced a shorthand notation $\mathbf{f}_f = (\mathbf{f}_C, \mathbf{f}_H, \mathbf{f}_L)$ and $K : \Gamma_f \times \Gamma_f \to \mathbb{Z}$ is a pairing on the co-character lattice of the boundary flavour symmetry $T_f = T_C \times T_H \times T_L$. For an $\mathcal{N} = (2, 2)$ boundary condition, the anomaly polynomial specialises to

$$\mathcal{P} = k(\mathbf{f}_H, \mathbf{f}_C) + k^C(\mathbf{f}_L, \mathbf{f}_C) + k^H(\mathbf{f}_H, \mathbf{f}_L) + \widetilde{k}(\mathbf{f}_L, \mathbf{f}_L), \tag{96}$$

where the coefficients $k, k^C, k^H, \widetilde{k}$ are pairings with the same structure as the supersymmetric Chern-Simons terms $\kappa_\alpha, \kappa_\alpha^C, \kappa_\alpha^H, \widetilde{\kappa}_\alpha$ in section 2.

For convenience, we define $\mathcal{P}_\alpha$ to be the polynomial above with pairings set to the corresponding supersymmetric Chern-Simons terms, encoding the contribution to the boundary anomaly from anomaly inflow from a massive vacuum $\alpha$ at $x^3 \to +\infty$. Thus $-\mathcal{P}_\alpha$ encodes the anomaly inflow from the vacuum at $x^3 \to -\infty$.

## 4.2 Boundary Amplitudes

A boundary condition preserving at least 2d $\mathcal{N} = (0, 2)$ supersymmetry in the $x^{1,2}$-plane shares a common pair of supercharges $Q_+^{++}, Q_+^{--}$ with the 1d $\mathcal{N} = (2, 2)$ subalgebra along $x^3$ annihilating supersymmetric ground states on $E_\tau$. In particular, the boundary condition preserves the combination $Q = Q_+^{++} + Q_+^{--}$, whose cohomology we use to compute supersymmetric ground states.

To a right or left $\mathcal{N} = (0, 2)$ boundary condition $B$, we can therefore associate boundary state $|B\rangle$ or $\langle B|$ respectively in the effective supersymmetric quantum mechanics on $\mathbb{R} \times E_\tau$. The overlaps of boundary states with supersymmetric ground states associated to vacua $\alpha$ are known as boundary amplitudes. Boundary amplitudes can be represented as a path integral on $E_\tau \times \mathbb{R}_{\leq 0}$ or $E_\tau \times \mathbb{R}_{\geq 0}$ with the boundary condition at $x^3 = 0$ and and the vacuum $\alpha$ at $x^3 \to -\infty$ or $x^3 \to +\infty$. This is illustrated in figure 10.

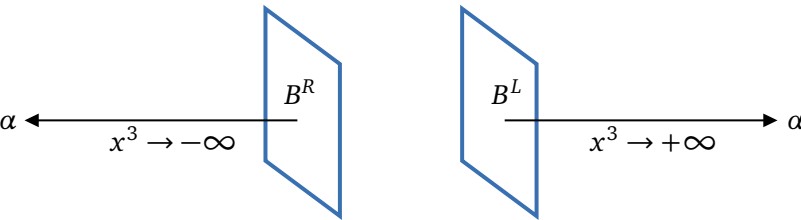

Figure 10: Boundary amplitudes for left and right boundary conditions

The presence of the vacuum $\alpha$ at infinity breaks the gauge symmetry of the theory on $E_\tau \times \mathbb{R}_{\geq 0}$ or $E_\tau \times \mathbb{R}_{\leq 0}$. One is led to consider boundary 't Hooft anomalies and anomaly inflow from the remaining $T_f = T_C \times T_H \times T_L$ flavour symmetry.[3] This may be computed by making

---

[3]Note that strictly speaking there is the usual subtlety in that the $T_H$ and $U(1)_V$ appearing here are actually the unbroken symmetries $T_H^{(\alpha)}$ and $U(1)_H^{(\alpha)}$ discussed in section 2.2.

the analogous substitutions to (14) in the boundary anomaly polynomial for the boundary condition (if the boundary condition supports a boundary gauge symmetry), replacing $(\sigma, m, \epsilon)$ with $(\mathbf{f}, \mathbf{f}_H, \mathbf{f}_t)$, and then adding the anomaly polynomial $\mathcal{P}_\alpha$ encoding anomaly inflow from $\alpha$.

The boundary amplitudes can be regarded as the elliptic genera on $E_\tau$ of effective 2d $\mathcal{N} = (0, 2)$ or $\mathcal{N} = (2, 2)$ theories obtained by reduction on a half-line. The mixed anomalies of the effective theory are simply the sum of the boundary mixed 't Hooft anomalies and the anomaly inflow from the supersymmetric Chern-Simons terms associated to the isolated massive vacua. As they are elliptic genera, the amplitudes therefore transform as sections of holomorphic line bundles on the torus $E_{T_f}$ of background flat connections $z_f = (z_C, z_H, z_t)$, whose quasi-periodicities are fixed by this sum [54].

In section 5, we will compute boundary amplitudes using supersymmetric localisation on $E_\tau \times I$ where $I$ is a finite interval, replacing the vacuum at infinity by a distinguished class of boundary conditions at finite distance that generate states in the same $Q$-cohomology class. The anomaly inflow from supersymmetric Chern-Simons terms is reproduced by the boundary 't Hooft anomalies of these boundary conditions.

The boundary amplitudes transform under large gauge transformations as follows,

$$f_\alpha(z_f + \nu_f + \tau \mu_f) = (-1)^{\ell_\alpha(\mu_f + \nu_f)} e^{-i\theta_\alpha(\mu_f, z_f)} f_\alpha(z_f), \tag{97}$$

where

$$\theta_\alpha(\mu_f, z_f) = 2\pi \left( k_\alpha(\mu_f, z_f) + k_\alpha(z_f, \mu_f) + \tau k_\alpha(\mu_f, \mu_f) \right) \tag{98}$$

and $k_\alpha$ is the total boundary mixed 't Hooft anomaly from the boundary condition and anomaly inflow from the vacuum $\alpha$.

The contribution $\ell_\alpha : \Gamma_f \to \mathbb{Z}$ is known as the linear anomaly [54]. We expect that with a careful identification of the $\mathbb{Z}_2$ fermion number in the elliptic genus with an R-symmetry whose background flat connection implements R-R boundary conditions, the linear anomaly may be considered as a mixed anomaly between flavour and R-symmetry, and placed on the same footing as $\theta_\alpha$. In this work, we follow the conventions of [55, 56] for $\mathcal{N} = (0, 2)$ boundary conditions and give a concrete geometric description of the linear anomaly for the amplitudes we consider.

To write down expressions for boundary amplitudes, it is convenient to view the elliptic curve as $E_\tau \cong \mathbb{C}^\times / q^{\mathbb{Z}}$ with $q = e^{2\pi i \tau}$ and write the background flat connections as[4]

$$\xi = e^{2\pi i z_C}, \qquad x = e^{2\pi i z_H}, \qquad t = e^{2\pi i z_t}, \tag{99}$$

or collectively $a_f = e^{2\pi i z_f}$ and $a = e^{2\pi i z}$, where recall $z_f = (z_C, z_H, z_t)$ and $z = (z_H, z_t)$. In this notation, the quasi-periodicities of boundary amplitudes becomes

$$\begin{aligned} f_\alpha(e^{2\pi i \nu_f} a_f) &= (-1)^{\ell_\alpha(\nu_f)} f_\alpha(a_f), \\ f_\alpha(q^{\mu_f} a_f) &= (-1)^{\ell_\alpha(\mu_f)} q^{-k_\alpha(\mu_f, \mu_f)} a_f^{-2k_\alpha(\mu_f)} f_\alpha(a_f). \end{aligned} \tag{100}$$

It is useful to define the normalised theta function,

$$\vartheta(a) := i \frac{\vartheta_1(z; \tau)}{\eta(\tau)}, \tag{101}$$

where $\vartheta_1(z; \tau)$ is the Jacobi theta function and $\eta(\tau)$ is the Dedekind eta function. It transforms under large background gauge transformations as $\vartheta(qx) = -x^{-1} q^{-\frac{1}{2}} \vartheta(x)$. This combination appears naturally in the computation of the elliptic genera of 2d $\mathcal{N} = (0, 2)$ supersymmetric

---

[4]In the remainder of this article, if the bulk $T_t$ symmetry is re-defined to a boundary $T_L$, we will still use $z_t$ and $t$ to denote the (exponentiated) background holonomies for $T_L$, with this understanding implicit.

gauge theories [55, 56] and, up to a factor of $q^{\frac{1}{12}}$ to ensure modularity, it is the same combination used in reference [5].

As in the discussion of supersymmetric ground states in 3.3, our construction of boundary amplitudes will depend on the real mass parameters $m$. We consider the cases of generic mass parameters and zero mass parameters in turn.

### 4.2.1 Generic Masses

First consider generic mass parameters in some chamber, $m \in \mathfrak{C}_H$. Recall from section 3.3 that the supersymmetric ground states $|\alpha\rangle_{\mathfrak{C}}$ and $_{\mathfrak{C}}\langle\alpha|$ are defined by placing a massive supersymmetric vacuum at $x^3 \to +\infty$ and $x^3 \to -\infty$.

For a boundary condition $B$ that is compatible with mass parameters in the chamber $\mathfrak{C}_H$, the boundary amplitudes are then defined as follows.

- The boundary amplitude $_{\mathfrak{C}}\langle\alpha|B\rangle$ is defined by the path integral on $E_\tau \times \mathbb{R}_{\leq 0}$ with a right boundary condition $B$ at $\tau = 0$ and the massive vacuum $\alpha$ at $\tau \to -\infty$.

- The boundary amplitude $\langle B|\alpha\rangle_{\mathfrak{C}}$ is defined by the path integral on $E_\tau \times \mathbb{R}_{\geq 0}$ with a left boundary condition $B$ at $\tau = 0$ and the massive vacuum $\alpha$ at $\tau \to +\infty$.

The boundary amplitudes transform as in (100) with

$$k_\alpha^\pm := k_B \pm K_\alpha, \tag{102}$$

where $k_B$ is the mixed 't Hooft anomaly of the boundary condition $B$. If $B$ suffers from a gauge anomaly, by $k_B$ we mean the anomalies in the unbroken flavour symmetries in the vacuum $\alpha$ as discussed at the beginning of section 4.2. The $\pm$ sign is for the vacuum at $\pm\infty$.

The contribution to the linear anomaly is more tricky to pin down. In section 5, we will formulate boundary amplitudes as elliptic genera of effective 2d $\mathcal{N} = (0,2)$ theories obtained by reduction on an interval. If these only involve standard $\mathcal{N} = (2,2)$ multiplets, the linear anomaly is determined by the difference between the sum of the $T_t$ weights of chiral multiplets and Fermi multiplets that contribute to the boundary amplitude. Since the linear anomaly is only defined mod 2, we have the relation

$$\ell_\alpha^\pm \otimes e_t = -\widetilde{k}_\alpha^\pm. \tag{103}$$

We will also see that equation (103) is true for the more exotic periodic boundary matter considered in section 6.

### 4.2.2 Vanishing Mass Parameters

We now consider boundary amplitudes obtained from the overlaps with supersymmetric ground states $|\alpha\rangle$ and $\langle\alpha|$ appropriate for vanishing mass parameters.

Such boundary amplitudes may be computed even if the boundary condition $B$ is incompatible with turning on mass parameters. However, if the boundary condition $B$ is compatible with mass parameters in the chamber $\mathfrak{C}_H$, using the relationship between supersymmetric ground states (70) we have

$$\begin{aligned}
\langle\alpha|B\rangle &= {}_{\mathfrak{C}}\langle\alpha|B\rangle \times \prod_{\lambda \in \Phi_\alpha^+} \vartheta(a^\lambda), \\
\langle B|\alpha\rangle &= \langle B|\alpha\rangle_{\mathfrak{C}} \times \prod_{\lambda \in \Phi_\alpha^-} \vartheta(a^\lambda).
\end{aligned} \tag{104}$$

They transform as in equation (100) with

$$k_\alpha^\pm := k_B \pm (\kappa_\alpha + \kappa_\alpha^C) + \frac{1}{4} \sum_{\lambda \in \Phi_\alpha} \lambda \otimes \lambda \,, \tag{105}$$

using the form of $\kappa_\alpha^H + \tilde{\kappa}$ in equation (37) and the additional contribution from the normalisation in (104). As in our discussion of supersymmetric ground states, an important compatibility condition is that this is independent of the chamber $\mathfrak{C}_H$. The normalisation also modifies the linear anomaly of the boundary amplitudes to

$$\ell_\alpha^\pm = -\widetilde{k}_B + \sum_{\lambda \in \Phi_\alpha^\mp} \lambda_H \pm \frac{1}{2} \sum_{\lambda \in \Phi_\alpha} \lambda_t \,, \tag{106}$$

which is again independent of the chamber $\mathfrak{C}_H$ due to the symplectic pairing of weights. Additionally, both $\ell_\alpha^\pm$ give equivalent factors of automorphy due to the symplectic pairing and the fact that the linear anomaly is sensitive only to parity.

The factors of automorphy of the boundary amplitudes are now independent of the chamber for the mass parameters. Additionally, on the loci

$$\lambda \cdot z \in \mathbb{Z} + \tau \mathbb{Z} \,, \tag{107}$$

where $\lambda \in \Phi_\alpha \cap (-\Phi_\beta)$ is any internal edge of the GKM diagram of $X$, $k_\alpha$ and $k_\beta$ define isomorphic line bundles (see appendix A). Following the discussion in section 3, this implies that the boundary amplitudes transform as a section of a holomorphic line bundle on the spectral curve $E_T(X)$.[5] The boundary amplitudes $\langle \alpha | B \rangle$ of a given boundary condition $B$ therefore represent a class in the $T$-equivariant elliptic cohomology of $X$.

### 4.2.3 Lagrangian Branes

Suppose we have a left / right $\mathcal{N} = (2,2)$ boundary condition $B$ that flows to a Lagrangian boundary condition $L \subset X$ in the sigma model to $X$.

First, suppose that the boundary condition is compatible with the mass parameters in some chamber $\mathfrak{C}_H$. This means concretely that $L \subset \bigcup_\alpha X_\alpha^\pm$ for a left / right boundary condition. Then we propose that the boundary amplitudes with mass parameters turned on are given by

$$\begin{aligned}
{}_\mathfrak{C}\langle \alpha | B \rangle &= \prod_{\lambda \in \Phi_\alpha^+(L)} \frac{\vartheta(a^{\lambda^*})}{\vartheta(a^\lambda)} = \prod_{\lambda \in \Phi_\alpha^+(L)} \frac{\vartheta(t^{-2-\lambda_t} v^{-\lambda_H})}{\vartheta(t^{\lambda_t} v^{\lambda_H})} \,, \\
\langle B | \alpha \rangle_\mathfrak{C} &= \prod_{\lambda \in \Phi_\alpha^-(L)} \frac{\vartheta(a^{\lambda^*})}{\vartheta(a^\lambda)} = \prod_{\lambda \in \Phi_\alpha^-(L)} \frac{\vartheta(t^{-2-\lambda_t} v^{-\lambda_H})}{\vartheta(t^{\lambda_t} v^{\lambda_H})} \,,
\end{aligned} \tag{108}$$

where $\Phi_\alpha^\pm(L) \subset \Phi_\alpha^\pm$ denotes the weights of the tangent space $T_\alpha L \subset T_\alpha X^\pm$. This is the elliptic genus of the $\mathcal{N} = (2,2)$ chiral multiplets parametrising the fluctuations in $T_\alpha L$. This will be derived using supersymmetric localisation in section 5, by introducing boundary conditions that generate boundary states in the same $Q$ cohomology classes as the supersymmetric ground states $|\alpha\rangle_\mathfrak{C}$.

When the mass parameters are set to zero, we instead consider the overlaps with the supersymmetric ground states $|\alpha\rangle$. The right boundary amplitude may be computed from the

---

[5]Note that in section 3 we ignored the contribution of the linear anomaly for the sake of brevity. However it is easy to check that the factors of automorphy arising from the linear anomaly coincide on the loci $\lambda \cdot z \in \mathbb{Z} + \tau \mathbb{Z}$ in the sense of appendix A.

above result as follows,

$$
\begin{aligned}
\langle B|\alpha\rangle &= \langle B|\alpha\rangle_{\mathfrak{C}} \prod_{\lambda\in\Phi_\alpha^-} \vartheta(a^\lambda) \\
&= \prod_{\lambda\in\Phi_\alpha^-(L)} \vartheta(a^{\lambda^*}) \prod_{\lambda\in\Phi_\alpha^-(L)^\perp} \vartheta(a^\lambda) \\
&= \prod_{\lambda\in\Phi_\alpha^+(L)^\perp} \vartheta(a^\lambda) \prod_{\lambda\in\Phi_\alpha^-(L)^\perp} \vartheta(a^\lambda) \\
&= \prod_{\lambda\in\Phi_\alpha(L)^\perp} \vartheta(a^\lambda),
\end{aligned}
\tag{109}
$$

where $\Phi_\alpha^\pm(L)^\perp$ denotes the complement of $\Phi_\alpha^\pm(L) \subset \Phi_\alpha^\pm$. A similar computation yields the same result for $\langle B|\alpha\rangle$. Note again the consistency check that there is no dependence on a chamber for the mass parameters.

This boundary amplitude corresponds to the elliptic genus of $\mathcal{N} = (0,2)$ Fermi multiplets parametrising the normal directions to $T_\alpha L \subset T_\alpha X$ with weights $\Phi_\alpha(L)^\perp$. The set of boundary amplitudes $\langle\alpha|B\rangle$ represent a section of a holomorphic line bundle on $E_T(X)$, which is the elliptic cohomology class of $L \subset X$.

### 4.2.4 Example

Let us consider an example from supersymmetric QED. We fix the default chambers and consider the boundary condition $N$, defined by a $\mathcal{N} = (2,2)$ Neumann boundary condition for the vector multiplet:

$$
F_{3\mu} = 0, \qquad \sigma = 0, \qquad D_3\sigma = 0,
\tag{110}
$$

together with the hypermultiplet boundary condition

$$
D_3 X_\beta = 0, \qquad Y_\beta = 0, \qquad \beta = 1,\ldots,N.
\tag{111}
$$

This flows to a compact Lagrangian brane supported on $L \cong \mathbb{CP}^{N-1} \subset X$ and is therefore compatible with any chamber for the mass parameters.

The Neumann boundary condition has a mixed 't Hooft anomaly between the boundary gauge symmetry and the bulk $U(1)_C$ R-symmetry and $T_C$ flavour symmetry, encoded in a contribution to the anomaly polynomial,

$$
\begin{aligned}
\mathbf{f}(-\mathbf{f}_C + N\mathbf{f}_{U(1)_C}) \quad &\text{for a right boundary condition,} \\
\mathbf{f}(+\mathbf{f}_C + N\mathbf{f}_{U(1)_C}) \quad &\text{for a left boundary condition.}
\end{aligned}
\tag{112}
$$

Here, $\mathbf{f}_{U(1)_C}$ denotes the field strength for the $U(1)_C$ R-symmetry at the boundary.

If we were to consider the boundary condition in isolation, we could define an unbroken boundary axial R-symmetry $U(1)_A$ generated by the current $J_A = J_{U(1)_C} \pm NJ_{T_C}$, which does not suffer a mixed gauge anomaly. This would implemented in the anomaly polynomial by setting $\mathbf{f}_{U(1)_C} = \mathbf{f}_A$ and $\mathbf{f}_C = \pm N\mathbf{f}_A$.

However, since we ultimately consider boundary amplitudes where the gauge symmetry is broken anyway, we instead consider the full boundary anomaly with $U(1)_V = U(1)_H$ and $U(1)_A = U(1)_C$. For example, for the left boundary condition we have

$$
\begin{aligned}
\mathcal{P}[N] &= \mathbf{f}\cdot\mathbf{f}_C + \mathbf{f}_V\cdot\mathbf{f}_A + \sum_{\beta=1}^{N}(\mathbf{f}-\mathbf{f}_H^\beta)\cdot\mathbf{f}_A \\
&= \mathbf{f}\cdot\mathbf{f}_C - \mathbf{f}_L\cdot\mathbf{f}_L - \sum_{\beta=1}^{N}(\mathbf{f}-\mathbf{f}_H^\beta)\cdot\mathbf{f}_L + \ldots,
\end{aligned}
\tag{113}
$$

where in the second line we only keep track of $\mathcal{N} = 2$ flavour symmetries. The first term arises from anomaly inflow, the second from gauginos surviving the Neumann boundary condition for the vector multiplet, and the remaining terms from fermions in the hypermultiplets.

Let us now consider the boundary amplitudes. The spaces of positive and negative weights in the default chamber are

$$
\begin{aligned}
\Phi_\alpha^-(L) &= \{e_\beta - e_\alpha, \beta > \alpha\}, \\
\Phi_\alpha^+(L) &= \{e_\beta - e_\alpha, \beta < \alpha\},
\end{aligned}
\tag{114}
$$

and therefore

$$
{}_{\mathfrak{c}}\langle\alpha|N\rangle = \prod_{\beta<\alpha} \frac{\vartheta(t^{-2}v_\alpha/v_\beta)}{\vartheta(v_\beta/v_\alpha)}, \qquad \langle N|\alpha\rangle_{\mathfrak{c}} = \prod_{\beta>\alpha} \frac{\vartheta(t^{-2}v_\alpha/v_\beta)}{\vartheta(v_\beta/v_\alpha)}.
\tag{115}
$$

These are the elliptic genera of the $\mathcal{N} = (2,2)$ chiral multiplets corresponding to the positive and negative weight fluctuations in $T_\alpha \mathbb{CP}^{N-1}$ respectively.

It is straightforward to check that these boundary amplitudes transform according to (102), where $k_B$ is obtained via substituting $\mathbf{f} = \mathbf{f}_H^\alpha - \mathbf{f}_L$ in the anomaly for $N$. For example, for the boundary amplitude $\langle N|\alpha\rangle_{\mathfrak{c}}$, the total boundary mixed 't Hooft anomaly is

$$
\begin{aligned}
\mathcal{P}_\alpha^+ &= \mathcal{P}[N]|_{\mathbf{f}=\mathbf{f}_H^\alpha-\mathbf{f}_L} + \mathcal{P}_\alpha \\
&= 2(N-\alpha)\mathbf{f}_L^2 + 2\sum_{\beta>\alpha}(\mathbf{f}_H^\beta - \mathbf{f}_H^\alpha)\mathbf{f}_L,
\end{aligned}
\tag{116}
$$

which reproduces the quasi-periodicity of the boundary amplitude. Note now $\mathbf{f}_L$ and $\mathbf{f}_H$ are field strengths for the symmetries $T_t^{(\alpha)}$ and $T_H^{(\alpha)}$ in the vacuum $\alpha$, as defined in section 2.2.

The boundary amplitudes at the origin of the mass parameter space are

$$
\langle\alpha|N\rangle = \langle N|\alpha\rangle = \prod_{\beta\neq\alpha} \vartheta(t^{-2}v_\alpha v_\beta^{-1}),
\tag{117}
$$

which is the elliptic genus of Fermi multiplets parametrising the cotangent directions at each fixed point $\alpha \subset \mathbb{CP}^{N-1}$. They represent the elliptic cohomology class of the compact Lagrangian submanifold $\mathbb{CP}^{N-1} \subset X$. By construction they transform according to quasi-periodicities (105).

### 4.3 Boundary Overlaps

The overlap $\langle B^l|B^r\rangle$ of boundary states can be defined as the partition function on $E_\tau \times [0,\ell]$ with R-R boundary conditions on $E_\tau$, with a left boundary condition $B^l$ at $x^3 = 0$ and right boundary condition $B^r$ at $x^3 = \ell$. The computation of such partition functions has been addressed using supersymmetric localisation in [57].

The overlap of boundary conditions is independent of the length $\ell$. This gives two ways to interpret the boundary overlap:

1. Sending $\ell \to 0$, it is the elliptic genus of the effective 2d $\mathcal{N} = (0,2)$ or $\mathcal{N} = (2,2)$ theory obtained by colliding the boundary conditions $B^l$, $B^r$.

2. Sending $\ell \to \infty$ and expanding in isolated massive vacua $\alpha$ in the intermediate region, it can be expressed in terms of the boundary amplitudes

$$
\begin{aligned}
\langle B^l|B^r\rangle &= \sum_\alpha \langle B^l|\alpha\rangle_{\mathfrak{c}}\ {}_{\mathfrak{c}}\langle\alpha|B^r\rangle \\
&= \sum_\alpha \langle B^l|\alpha\rangle\langle\alpha|B^r\rangle \prod_{\lambda\in\Phi_\alpha} \frac{1}{\vartheta(a^\lambda)}.
\end{aligned}
\tag{118}
$$

In the first line, we assume we can turn on mass parameters in a chamber $\mathfrak{C}$ compatible with both boundary conditions.

These interpretations are both compatible with the transformation properties under large background gauge transformations, namely that the boundary overlap transforms with a factor of automorphy fixed by the sum $k_l + k_r$ of boundary anomalies from $B^l$ and $B^r$.

This is because the only possible 't Hooft anomalies arise from boundary chiral fermions and anomaly inflow. In colliding the boundary conditions, the contributions from anomaly inflow to the left and the right cancel out. In the decomposition into boundary amplitudes, the factor of automorphy of each term in the first line are, using (102)

$$k_l + k_r = (k_l - K_\alpha) + (k_r + K_\alpha),\tag{119}$$

where $k_{l,r}$ are the boundary 't Hooft anomalies of $B^{l,r}$. The second decomposition also has the same factors of automorphy,

$$
\begin{aligned}
k_l + k_r = \Big(k_l - \kappa_\alpha - \kappa_\alpha^C + \frac{1}{4}\sum_{\lambda \in \Phi_\alpha} \lambda \otimes \lambda\Big) + \Big(k_r + \kappa_\alpha + \kappa_\alpha^C + \frac{1}{4}\sum_{\lambda \in \Phi_\alpha} \lambda \otimes \lambda\Big) \\
- \frac{1}{2}\sum_{\lambda \in \Phi_\alpha} \lambda \otimes \lambda,
\end{aligned}
\tag{120}
$$

and so all of these interpretations are compatible.

Let us finally mention an important subtlety. We have already encountered the fact that Neumann boundary conditions for the vector multiplet generically have mixed 't Hooft anomalies for the unbroken boundary gauge symmetry. This is not problematic for boundary amplitudes as the gauge symmetry is completely broken in a massive vacuum $\alpha$ anyway. However, for overlaps $\langle B^l | B^r \rangle$ between pairs of Neumann boundary conditions, a mixed 't Hooft anomaly between a boundary gauge symmetry and flavour symmetry $T_f$ will require a specialisation of the background flat connections $z_f$ for consistency. An example is presented below.

### 4.3.1 Example

Let us continue with the example of supersymmetric QED with $N$ flavours, and compute the overlap $\langle N | N \rangle$ of the Neumann boundary condition $N$ supported on $\mathbb{CP}^{N-1} \subset X$.

In the limit $\ell \to 0$, we recover a 2d $\mathcal{N} = (2,2)$ gauge theory with $G = U(1)$ and $N$ chiral multiplets of charge $+1$. The computation of the elliptic genus is subtle due to the mixed $G - T_L$ anomaly with coefficient $2N$. This is an example presented in [55]. The result is

$$
\begin{aligned}
\langle N | N \rangle &= -\oint_\Gamma \frac{ds}{2\pi i s} \frac{\eta(\tau)^2}{\vartheta(t^{-2})} \prod_{\alpha=1}^N \frac{\vartheta(t s^{-1} v_\beta)}{\vartheta(t s v_\beta^{-1})} \\
&= \sum_{\alpha=1}^N \prod_{\beta \neq \alpha} \frac{\vartheta(t^{-2} v_\alpha / v_\beta)}{\vartheta(v_\beta / v_\alpha)},
\end{aligned}
\tag{121}
$$

where the JK contour $\Gamma$ selects poles at $s = v_\alpha t^{-1}$, and single-valuedness of the integrand requires $t^{2N} = 1$ due to the mixed anomaly.

The latter is a consequence of considering the overlap between left and right Neumann boundary conditions $N$; one cannot simultaneously make both of the redefinitions below equation (112) to define boundary axial $R$-symmetries with no mixed gauge anomaly. The $G - U(1)_C$ anomaly of the $N$ chiral multiplets in the limit $\ell \to 0$ is equal, as expected, to the sum of the anomalies of the left and right $N$ boundary conditions given in (112).

The same result is reproduced by the decomposition into boundary amplitudes given in equations (115) and (117) obtained in the opposite limit $\ell \to \infty$, but global consistency requires $t^{2N} = 1$.

### 4.4 Boundary Wavefunctions

We will introduce another decomposition by cutting the path integral using an auxiliary set of Dirichlet supersymmetric boundary conditions. This type of construction has been used extensively in the literature on supersymmetric localisation [58–64] and analysed systematically in [50, 65].

There is significant freedom in the choice of auxiliary Dirichlet boundary conditions. Different choices have advantages and disadvantages, especially in how the auxiliary boundary conditions interact with mass parameters. We describe two choices of auxiliary boundary conditions that preserve $\mathcal{N} = (2, 2)$ and $\mathcal{N} = (0, 2)$ supersymmetry.

#### 4.4.1 $\mathcal{N} = (2, 2)$ Cutting

The first method uses boundary conditions that preserve 2d $\mathcal{N} = (2, 2)$ supersymmetry and involves Dirichlet boundary conditions for the vector multiplet [15]. Specifically, we consider the following Dirichlet boundary conditions $D_\varepsilon$:

- An $\mathcal{N} = (2, 2)$ Dirichlet boundary condition for the 3d $\mathcal{N} = 4$ vector multiplet, where the complex scalar vanishes at the boundary $\varphi = 0$, as does the parallel component of the field strength $F_{12} = 0$.

- A $\mathcal{N} = (2, 2)$ Neumann-Dirichlet boundary condition for the hypermultiplet. This depends on a polarisation or Lagrangian splitting $\varepsilon$ of the representation $Q = T^*R$.[6] Let us write $R = \mathbb{C}^N$ with polarisation denoted by a sign vector $\varepsilon \in \{\pm\}^N$ specifying

$$(X_{\varepsilon_\beta}, Y_{\varepsilon_\beta}) = \begin{cases} (X_\beta, Y_\beta) & \text{if } \varepsilon_\beta = + \\ (Y_\beta, -X_\beta) & \text{if } \varepsilon_\beta = - \end{cases} \quad \text{for } \beta = 1, \ldots, N. \tag{122}$$

The boundary condition specifies that

$$D_\perp X_{\varepsilon_\beta}| = 0, \qquad Y_{\varepsilon_\beta}| = 0 \tag{123}$$

and in particular $X_{\varepsilon_\beta}$ transform in $\mathcal{N} = (2, 2)$ chiral multiplets at the boundary.

In addition to the bulk flavour symmetry $T_f$, the boundary conditions support a boundary $G_\partial$ flavour symmetry, generated by global gauge transformations at the boundary. Placing the boundary condition on $E_\tau$, in addition to the background flavour connection with holonomy $a_f = e^{2\pi i z_f}$, we may also introduce a background flat connection for the boundary $G_\partial$ symmetry with holonomy $s = e^{2\pi i u}$. We therefore denote the boundary conditions by $D_\varepsilon(s)$.

We now cut the path integral as follows. We first impose the above Dirichlet boundary conditions on the left and right of the cut. This introduces a pair of boundary flavour symmetries $G_\partial$, $G'_\partial$. We then introduce $N$ boundary $\mathcal{N} = (2, 2)$ chiral multiplets $\Phi_\beta$ coupled to the bulk hypermultiplet fields to the boundary via a superpotential

$$W = \sum_{\beta=1}^N X_{\varepsilon_\beta} |\phi_\beta - \phi_\beta| X'_{\varepsilon_\beta}, \tag{124}$$

which involves the hypermultiplet fields with Neumann boundary conditions. The boundary superpotential identifies the boundary flavour symmetries $G_\partial$, $G'_\partial$ and imposes (omitting the subscript on the polarisation)

$$Y_\varepsilon| = \frac{\partial W}{\partial X_\varepsilon|} = \phi, \quad |Y'_\varepsilon = -\frac{\partial W}{\partial |X'_\varepsilon} = \phi, \quad 0 = \frac{\partial W}{\partial \phi} = X_\varepsilon| - |X'_\varepsilon, \tag{125}$$

---

[6]This is a decomposition of the representation $T^*R \cong L \oplus L^*$, where $L$ is a Lagrangian. We will use the two terminologies interchangeably.

thus identifying the hypermultiplet fields on each side. We then gauge the remaining diagonal $G_\partial$ boundary symmetry to $G$, by introducing a dynamical 2d $\mathcal{N} = (2,2)$ vector multiplet.

This leads to a decomposition of boundary overlaps

$$\left\langle B^l | B^r \middle| B^l | B^r \right\rangle = (-)^r \oint_{\mathrm{JK}} du \, \mathcal{Z}_\varepsilon(s) \left\langle B^l | D_\varepsilon(s) \middle| B^l | D_\varepsilon(s) \right\rangle \left\langle D_\varepsilon(s) | B^r \middle| D_\varepsilon(s) | B^r \right\rangle , \tag{126}$$

where $\mathcal{Z}_\varepsilon(s)$ is the elliptic genus of the boundary vector multiplet and chiral multiplets. We refer to $\langle D_\varepsilon(s) | B | D_\varepsilon(s) | B \rangle$ as the wavefunction of a right boundary condition $B$ and $\langle B | D_\varepsilon(s) | B | D_\varepsilon(s) \rangle$ as the wavefunction of a left boundary condition $B$. Note that the integrand is independent of the choice of polarisation, with the dependence on $\varepsilon$ in $\mathcal{Z}_\varepsilon(s)$ cancelled by the dependence of the wavefunctions.

The integral is over a real contour in the parameter space $E_G$ of flat connections and performed according to the JK residue prescription for elliptic genera [55, 56]. Provided there is no anomaly for the total boundary $G$ symmetry obtained by summing the contributions from $B^l, B^r$, the integrand is invariant under $s \to qs$ and there are a finite set of poles.

### 4.4.2 Example

For supersymmetric QED with $N$ hypermultiplets,

$$\mathcal{Z}_\varepsilon(s) = \frac{\eta(q)^2}{\vartheta(t^{-2})} \prod_{\beta=1}^{N} \frac{\vartheta(ts^{\varepsilon_\beta} v_\beta^{-\varepsilon_\beta})}{\vartheta(ts^{-\varepsilon_\beta} v_\beta^{\varepsilon_\beta})} . \tag{127}$$

Let us reconsider the normalisation of the Neumann boundary condition $N$ supported on the base $\mathbb{CP}^{N-1} \subset X$ from this perspective. For this boundary condition, it is convenient to choose the polarisation $\varepsilon = (+, \ldots, +)$. Then we also have

$$\langle N | D_\varepsilon(s) | N | D_\varepsilon(s) \rangle = \langle D_\varepsilon(s) | N | D_\varepsilon(s) | N \rangle = \prod_{\beta=1}^{N} \frac{\vartheta(ts^{-1} v_\beta)}{\vartheta(ts v_\beta^{-1})} , \tag{128}$$

which reproduces the elliptic genus of the $\mathcal{N} = (2,2)$ chiral multiplets arising from the hypermultiplet fields $X_\beta$ that have Neumann boundary conditions at both boundaries. Putting these components together we find

$$\begin{aligned}
\langle N | N \rangle &= -\oint_{\mathrm{JK}} du \, \frac{\eta(q)^2}{\vartheta(t^{-2})} \prod_{\beta=1}^{N} \frac{\vartheta(ts v_\beta^{-1})}{\vartheta(ts^{-1} v_\beta)} \frac{\vartheta(ts^{-1} v_\beta)}{\vartheta(ts v_\beta^{-1})} \frac{\vartheta(ts^{-1} v_\beta)}{\vartheta(ts v_\beta^{-1})} \\
&= -\oint_{\mathrm{JK}} du \, \frac{\eta(q)^2}{\vartheta(t^{-2})} \prod_{\beta=1}^{N} \frac{\vartheta(ts^{-1} v_\beta)}{\vartheta(ts v_\beta^{-1})} ,
\end{aligned} \tag{129}$$

which agrees with the previous computation (121).

### 4.4.3 $\mathcal{N} = (0,2)$ Cutting

The $\mathcal{N} = (2,2)$ cutting has some inconvenient features. First, it depends on a choice of polarisation $\varepsilon$. Second, there may not exist a polarisation that is compatible with all the supersymmetric massive vacua $\{\alpha\}$, meaning that some of the overlaps $\langle D_\varepsilon(s) | \alpha | D_\varepsilon(s) | \alpha \rangle$ break supersymmetry and vanish. Finally, the auxiliary boundary conditions $D_\varepsilon(s)$ may not be compatible with introducing mass parameters in the same chamber as a given boundary condition $B$.

To circumvent these difficulties, we consider an alternative set of auxiliary boundary conditions preserving only $\mathcal{N} = (0,2)$ supersymmetry. The additional flexibility will allow us to make more canonical choices that are compatible with all supersymmetric massive vacua and mass parameters in any chamber.

Let us first note that a 3d $\mathcal{N} = 4$ vector multiplet decomposes into a 3d $\mathcal{N} = 2$ vector multiplet and an adjoint chiral multiplet with scalar component $\varphi$. A 3d $\mathcal{N} = 4$ hypermultiplet decomposes into a pair of 3d $\mathcal{N} = 2$ chiral multiplets $X$ and $Y$. They decompose further under 2d $\mathcal{N} = (0,2)$ supersymmetry as follows:

- The 3d $\mathcal{N} = 2$ vector multiplet decomposes into a $\mathcal{N} = (0,2)$ chiral superfield $S$ containing $A_3 - i\sigma$ as its scalar component, and a $\mathcal{N} = (0,2)$ Fermi field strength multiplet $\Upsilon$, containing $F_{12}$.

- The 3d $\mathcal{N} = 2$ chiral multiplets $\varphi, X, Y$ decompose into $\mathcal{N} = (0,2)$ chiral multiplets $\Phi_\varphi$, $\Phi_X$, $\Phi_Y$, and $\mathcal{N} = (0,2)$ Fermi multiplets $\Psi_\varphi, \Psi_X, \Psi_Y$.

Alternatively, we could have first decomposed under $\mathcal{N} = (2,2)$ supersymmetry, before further decomposing under $\mathcal{N} = (0,2)$ supersymmetry. From this perspective, the above supermultiplets arise from a chiral multiplet $(S, \bar{\Psi}_\varphi)$, a twisted chiral field strength multiplet $(\Phi_\varphi, \Upsilon)$, and chiral multiplets $(\Phi_X, \Psi_Y)$ and $(\Phi_Y, -\Psi_X)$.

Let us now describe the auxiliary $\mathcal{N} = (0,2)$ boundary conditions. First, we always assign a Dirichlet boundary condition for the 3d $\mathcal{N} = 2$ vector multiplet. This supports a boundary $G_\partial$ flavour symmetry and allows us to introduce a background flat connection with holonomy $s = e^{2\pi i u}$. We then assign a Neumann boundary condition for the $\mathcal{N} = 2$ chiral multiplet containing $\varphi$,

$$\Psi_\varphi| = 0, \qquad D_3 \Phi_\varphi = 0. \tag{130}$$

This is in contrast with the $\mathcal{N} = (2,2)$ Dirichlet boundary condition for a 3d $\mathcal{N} = 4$ vector multiplet, which would assign a Dirichlet boundary condition to $\varphi$.

We then introduce two sets of auxiliary boundary conditions, with Neumann and Dirichlet boundary conditions for all the hypermultiplet scalar fields. In terms of $\mathcal{N} = (0,2)$ supermultiplets, they are:

$$\begin{aligned} D_C(s): \quad & \Psi_X| = \Psi_Y = 0|, \qquad D_3 \Phi_X| = D_3 \Phi_Y| = 0, \\ D_F(s): \quad & \Phi_X| = \Phi_Y| = 0, \qquad D_3 \Psi_X| = D_3 \Psi_Y| = 0. \end{aligned} \tag{131}$$

The subscripts therefore signify whether $\mathcal{N} = (0,2)$ chiral or Fermi multiplets obey Neumann boundary conditions. We note that the $D_C$ boundary condition is compatible with all supersymmetric vacua. We can then associate wavefunctions $\langle D_C(s)|B\rangle$ and $\langle D_F(s)|B\rangle$ to a right boundary condition $B$ and wavefunctions $\langle B|D_C(s)\rangle$ and $\langle B|D_F(s)\rangle$ to a left boundary condition.

There are now four ways to cut the path integral by introducing the boundary conditions $D_C, D_F$ on each side with appropriate superpotential couplings. Different choices will reflect different mathematical interpretations of the overlaps. We describe two of the four explicitly.

First, let us assign a left $D_F$ boundary condition on the right of the cut and a right $D_C$ on the left of the cut. They are then coupled by a boundary superpotential given in terms of boundary superfields as

$$\int d^2x \, d\theta^+ \, \Phi_X| \cdot |\Psi_{X'} + \Phi_Y| \cdot |\Psi_{Y'} + \Phi_\varphi| \cdot \Gamma_\varphi - \Gamma_\varphi \cdot |\Phi_{\varphi'}, \tag{132}$$

where $\Gamma_\varphi$ is an auxiliary boundary Fermi multiplet in the adjoint representation of $G$, whose $\mathcal{N} = 2$ flavour charges are fixed by invariance of the superpotential. This identifies the bound-

ary $G$ symmetries and imposes

$$\Psi_X| = |\Psi_{X'}, \quad \Phi_X| = |\Phi_{X'}, \quad \Psi_Y| = |\Psi_{Y'}, \quad \Phi_Y| = |\Phi_{Y'},$$
$$\Phi_\varphi| - |\Phi_{\varphi'} = 0, \quad \Psi_\varphi| = \Gamma_\varphi = |\Psi_{\varphi'},$$

(133)

which identifies $X| = |X'$, $Y| = |Y'$ and $\varphi| = |\varphi'$ and their super-partners across the interface. We then gauge the remaining diagonal $G$ boundary symmetry by introducing a dynamical $2d$ $\mathcal{N} = (0,2)$ vector multiplet.

This interface leads to the decomposition of overlaps into boundary amplitudes,

$$\langle B^l | B^r | B^l | B^r \rangle = \oint du, \mathcal{Z}_{V,\Gamma_\varphi}(s) \langle B^l | D_C(s) | B^l | D_C(s) \rangle \langle D_F(s) | B^r | D_F(s) | B^r \rangle,$$

(134)

where $\mathcal{Z}_{V,\Gamma_\varphi}(s)$ is the contribution of the dynamical $\mathcal{N} = (0,2)$ vector multiplet and Fermi multiplet $\Gamma_\varphi$ at the boundary together with a minus sign $(-)^r$ from the gauge integral,

$$\mathcal{Z}_{V,\Gamma_\varphi}(s) = (\eta(q)^2 \vartheta(t^2))^r \prod_{\alpha \in G} \vartheta(s^\alpha) \vartheta(t^2 s^\alpha).$$

(135)

The product is over roots $\alpha$ of $G$.

The second type of interface assigns the boundary conditions $D_C$ to both sides of the cut and introduces the boundary superpotential

$$\int d^2 x d\theta^+ \Phi_X| \cdot \Gamma_X - \Gamma_X \cdot |\Phi_{X'} + \Phi_Y| \cdot \Gamma_Y - \Gamma_Y \cdot |\Phi_{Y'} + \Phi_\varphi| \cdot \Gamma_\varphi - \Gamma_\varphi \cdot |\Phi_{\varphi'},$$

(136)

where $\Gamma_X$, $\Gamma_Y$, and $\Gamma_\varphi$ are boundary Fermi multiplets in the appropriate representations. The superpotential couplings identify

$$\Phi_X| - |\Phi_{X'} = 0, \quad \Phi_Y| - |\Phi_{Y'} = 0, \quad \Psi_X| = \Gamma_X = |\Psi_{X'}, \quad \Psi_Y| = \Gamma_Y = |\Psi_{Y'},$$
$$\Phi_\varphi| - |\Phi_{\varphi'} = 0, \quad \Psi_\varphi| = \Gamma_\varphi = |\Psi_{\varphi'},$$

(137)

which again identifies the fields across the interface. We then gauge the remaining diagonal $G$ symmetry by introducing an $\mathcal{N} = (0,2)$ vector multiplet. This interface allows overlaps to be constructed from wavefunctions,

$$\langle B^l | B^r | B^l | B^r \rangle = \oint_{\mathrm{JK}} du \, \mathcal{Z}_{V,\Gamma_\varphi} \mathcal{Z}_\Gamma(s) \langle B^l | D_C(s) | B^l | D_C(s) \rangle \langle D_C(s) | B^r | D_C(s) | B^r \rangle,$$

(138)

where $\mathcal{Z}_\Gamma(s)$ is the elliptic genus of the boundary Fermi multiplets $\Gamma_X$ and $\Gamma_Y$.

It is straightforward to check that this decomposition is equivalent to the first. The Fermi multiplets $\Gamma_X$, $\Gamma_Y$ implement a flip of the left boundary condition for the 3d $\mathcal{N} = 2$ chiral multiplets $X$, $Y$ from Dirichlet to Neumann. For the decompositions of overlaps, the ratio of the wavefunctions $\langle D_C(s)|B|D_C(s)|B \rangle$ and $\langle D_F(s)|B|D_F(s)|B \rangle$ is precisely the contribution $\mathcal{Z}_\Gamma(s)$ of the boundary Fermi multiplets.

The remaining two decompositions are constructed in a similar manner. In summary, the four possible decompositions of an overlap into wavefunctions are

$$\langle B^l | B^r \rangle = \oint du \, \mathcal{Z}_{V,\Gamma_\varphi}(s) \langle B^l | D_C(s) | B^l | D_C(s) \rangle \langle D_F(s) | B^r | D_F(s) | B^r \rangle$$

$$= \oint du \, \mathcal{Z}_{V,\Gamma_\varphi}(s) \langle B^l | D_F(s) | B^l | D_F(s) \rangle \langle D_C(s) | B^r | D_C(s) | B^r \rangle$$

$$= \oint du \, \mathcal{Z}_{V,\Gamma_\varphi} \mathcal{Z}_\Gamma(s) \langle B^l | D_C(s) | B^l | D_C(s) \rangle \langle D_C(s) | B^r | D_C(s) | B^r \rangle$$

$$= \oint du \, \mathcal{Z}_{V,\Gamma_\varphi} \mathcal{Z}_C(s) \langle B^l | D_F(s) | B^l | D_F(s) \rangle \langle D_F(s) | B^r | D_F(s) | B^r \rangle.$$

(139)

Here $Z_C(s)$ is the elliptic genus of auxiliary $\mathcal{N} = (0,2)$ chirals $C_X$ and $C_Y$ coupled to $\Psi_X$ and $\Psi_Y$ at the analogous interface to (136) with the roles of chirals and Fermis interchanged. It is easy to check from the charge assignments that $Z_C = Z_\Gamma^{-1}$.

If both $B^l$, $B^r$ prescribe Neumann boundary conditions for the vector multiplet, the integral is a JK residue prescription [55–57]. As before, if the effective 2d $\mathcal{N} = (0,2)$ theory has mixed 't Hooft anomalies involving the gauge symmetry, it is necessary to restrict the background flat connections to ensure the integrand is periodic and the contour integral is well-defined. In section 5, we will consider boundary conditions involving Dirichlet for the vector multiplet, which enforce a different pole prescription.

For the wavefunctions for the auxiliary boundary conditions themselves, by setting $B^l = D_F(s')$ in the top line of (139), consistency requires that

$$\langle D_F(s')|D_C(s)\rangle = \mathcal{Z}_{V,\Gamma_\varphi}(s)^{-1}\delta^{(r)}(u-u').\tag{140}$$

Here $\delta^{(r)}(u-u')$ should be considered as a pole prescription around a pole of rank $r$ at $u = u'$ of residue 1. The wavefunctions involving other combinations of $D_C$, $D_F$ are related by a normalisation by $Z_C$ or $Z_\Gamma$.

The wavefunction (140) is consistent with its path integral representation on $E_\tau \times [0,\ell]$. If $s \neq s'$ the system breaks supersymmetry and the path integral vanishes. If $s = s'$, sending $\ell \to 0$, the remaining fluctuating 2d $\mathcal{N} = (0,2)$ supermultiplets are the adjoint chiral $\Phi_\varphi$ charged under $T_t$, and an adjoint chiral $S$ neutral under $T_t$. The Cartan components of the latter naively gives a factor

$$(-\vartheta(1)^{-1})^r,\tag{141}$$

which is singular. However, noting that

$$2\pi i\,\mathrm{Res}_{u=0}\vartheta(q;e^{2\pi i u})^{-1} = \eta(q)^{-2},\tag{142}$$

we replace the contribution of the $S$ by

$$(-)^r\frac{\delta^{(r)}(u-u')}{\eta(q)^{2r}},\tag{143}$$

where the delta function is regarded as a pole prescription as above. If we combine this with the off-diagonal contribution of $S$ and the adjoint chiral $\varphi$, we reproduce (140).

### 4.4.4 Example

Let us take supersymmetric QED with $N$ hypermultiplets and again consider the overlap of the Neumann boundary condition supported on $\mathbb{CP}^{N-1} \subset X$. By taking the limit $\ell \to 0$, one has the following wavefunctions

$$\langle D_F(s)|N|D_F(s)|N\rangle = \langle N|D_F(s)|N|D_F(s)\rangle = \frac{-1}{\vartheta(t^2)}\prod_{\beta=1}^{N}\vartheta(t^{-1}sv_\beta^{-1}),$$

$$\langle D_C(s)|N|D_C(s)|N\rangle = \langle N|D_C(s)|N|D_C(s)\rangle = \frac{-1}{\vartheta(t^2)}\prod_{\beta=1}^{N}\frac{-1}{\vartheta(tsv_\beta^{-1})}.\tag{144}$$

In the first line, the remaining degrees of freedom after collapsing the interval are the chiral multiplet $\Phi_\varphi$ and the $N$ Fermi multiplets $\Psi_{Y_\beta}$. Similarly, in the second line, the remaining degrees of freedom are $\Phi_\varphi$ and the $N$ chiral multiplets $\Phi_{X_\beta}$.

The various contributions to the cutting formula are

$$\mathcal{Z}_{V,\Gamma_\varphi} = (\eta(q)^2\,\vartheta(t^2))^r\,,$$

$$\mathcal{Z}_\Gamma = \prod_{\beta=1}^{N}\vartheta(t^{-1}sv_\beta^{-1})\,\vartheta(t^{-1}s^{-1}v_\beta)\,,\qquad \mathcal{Z}_C = \prod_{\beta=1}^{N}\frac{1}{\vartheta(tsv_\beta^{-1})\,\vartheta(ts^{-1}v_\beta)}\,, \tag{145}$$

and using any of the four decompositions in equation (139), the normalisation of the Neumann boundary condition agrees with (121).

## 4.5 Wavefunctions to Amplitudes

We now explain how to pass from wavefunctions to boundary amplitudes. We present the results for boundary amplitudes constructed from the supersymmetric ground states $|\alpha\rangle$ and wavefunctions constructed from the $\mathcal{N}=(0,2)$ boundary conditions $D_F(s)$ and $D_C(s)$. These combinations are canonical in the sense that they do not depend on a choice of chamber or Lagrangian splitting. Other choices are found from the relations presented in previous sections.

We will derive the results using consistency between the decompositions into boundary amplitudes and wavefunctions considered thus far. We will introduce boundary condition representatives of the supersymmetric ground states and a derivation of the same results utilising supersymmetric localisation in section 5.

Let us then compare the decomposition of an overlap into wavefunctions and boundary amplitudes. For this purpose, it is most convenient to start from the decomposition into wavefunctions using the auxiliary Dirichlet boundary condition $D_F(s)$,

$$\langle B^l|B^r\rangle = \oint_{\mathrm{JK}} du\,\mathcal{Z}_{V,\Gamma_\varphi}\mathcal{Z}_C(s)\,\big\langle B^l|D_F(s)\big|B^l|D_F(s)\big\rangle\,\langle D_F(s)|B^r|D_F(s)|B^r\rangle\,. \tag{146}$$

Let us assume $B^l$ and $B^r$ prescribe a Neumann boundary condition for the vector multiplet. Then the poles contributing to the JK residue prescription arise entirely from the contribution $\mathcal{Z}_C(s)$ of the auxiliary chiral multiplets and are in 1-1 correspondence with supersymmetric vacua $\alpha$.

Let us describe this concretely, returning to the description of supersymmetric vacua in section 2.2. The contribution of the auxiliary chiral multiplets may be expressed in terms of the weight decomposition of the matter representation $T^*R$ as

$$\mathcal{Z}_C(s) = \prod_{\varrho\in T^*R}\frac{1}{\vartheta(w^\varrho)}\,, \tag{147}$$

where we have denoted the $G\times T_H\times T_t$ fugacities collectively as $w=(s,v,t)$, and weights as $\varrho=(\rho,\rho_H,\rho_t)$. The JK residue prescription for the elliptic genus [55,56] selects the rank $r$ poles given by

$$w^{\varrho_i} = 1\,,\quad i=1,\dots,r\,, \tag{148}$$

where the collection of $G$ weights $\{\rho_1,\dots,\rho_r\}$ obey the conditions outlined in section 2.2. Such collections are in 1-1 correspondence with supersymmetric vacuum $\alpha$. Recall we may invert the weights to obtain a unique value of the boundary gauge flat connection $u=u_\alpha$, and denote $s_\alpha = e^{2\pi iu_\alpha}$.

This implies the following crucial property of the contribution of the integrand:

$$(2\pi i)^r\mathop{\mathrm{Res}}_{u=u_\alpha}\mathcal{Z}_{V,\Gamma_\varphi}(s)Z_C(s) = \prod_{\lambda\in\Phi_\alpha}\frac{1}{\vartheta(a^\lambda)}\,. \tag{149}$$

In the above, the contribution of the $\mathcal{N} = (0, 2)$ vector multiplet and adjoint Fermi $\Gamma_\varphi$ play the role of the complex moment map and quotient in (32).

This is consistent with the decomposition into boundary amplitudes (118) provided the wavefunction $\langle D_F(s)|B\rangle$ evaluates to the boundary amplitude $\langle \alpha|B\rangle$ at $s = s_\alpha$. In summary, consistency demands that

$$\langle D_F(s_\alpha)|B\rangle = \langle \alpha|B\rangle, \tag{150}$$

$$(2\pi i)^r \operatorname*{Res}_{u=u_\alpha} \mathcal{Z}_{V,\Gamma_\varphi}(s) \langle D_C(s)|B\rangle = \prod_{\lambda \in \Phi_\alpha} \frac{1}{\vartheta(a^\lambda)} \langle \alpha|B\rangle. \tag{151}$$

Note that this implies the supersymmetric ground states $|\alpha\rangle$ lie in the same $Q$-cohomology classes as the boundary states generated by a boundary condition of the form $D_F(s_\alpha)$. We return to this observation in the next section.

It is also useful to consider the wavefunctions of the supersymmetric ground states themselves. Compatibility with the above results and the normalisation of supersymmetric ground states requires that

$$\langle D_F(s_\alpha)|\alpha\rangle = \prod_{\lambda \in \Phi_\alpha} \vartheta(a^\lambda), \tag{152}$$

$$(2\pi i)^r \operatorname*{Res}_{u=u_\alpha} \mathcal{Z}_{V,\Gamma_\varphi}(s) \langle D_C(s)|\alpha\rangle = 1. \tag{153}$$

### 4.5.1 Mathematical Interpretation

Let us now discuss the interpretation of the boundary wavefunctions in terms of equivariant elliptic cohomology. We have already seen that the boundary amplitudes $\langle \alpha|B\rangle$ transform as sections of holomorphic line bundles on $E_{T_f}^{(\alpha)}$ that glue to a section of a holomorphic line bundle on $E_T(X)$. The wavefunction repackages this information using the gauge theory description of $E_T(X)$ described in section 3.5.2.

First note that the auxiliary Dirichlet boundary conditions break the gauge symmetry, leaving a boundary $G$ flavour symmetry, which we have denoted $G_\partial$. The corresponding wavefunction $\langle D_F(s)|B\rangle$ therefore transforms as a section of a line bundle on the spectral curve $E_{T_f} \times E_G$, where the flat connection $s$ parametrises $E_G$. The associated boundary amplitudes obtained by setting $s = s_\alpha$,

$$\langle \alpha|B\rangle = \langle D_F(s_\alpha)|B\rangle, \tag{154}$$

represent the equivariant elliptic cohomology class obtained by pull back via the inclusion $c : E_T(X) \hookrightarrow E_F \times E_G$. In the mathematics literature we reference, *e.g.* [5,8,51,76] equivariant elliptic cohomology classes are often given in this 'off-shell' form, as classes on $E_{T_f} \times E_G$, with the pull back implicit.

### 4.5.2 Example

Let us return to supersymmetric QED and check the relation between the boundary amplitudes and wavefunctions of the Neumann boundary condition $N$ supported on $\mathbb{CP}^{N-1} \subset X$. The wavefunctions were given in (144). We then have $s_\alpha = v_\alpha t^{-1}$ and

$$\langle D_F(s_\alpha)|N|D_F(s_\alpha)|N\rangle = \prod_{\beta \neq \alpha} \vartheta(t^{-2}v_\alpha v_\beta^{-1}) \tag{155}$$

which agrees with the boundary amplitude $\langle \alpha|N|\alpha|N\rangle$ in (117). Similarly, we find

$$\vartheta(t^2)\eta(q)^2 \operatorname{Res}_{s=s_\alpha} \langle D_C(s)|N|D_C(s)|N\rangle = \prod_{\beta \neq \alpha} \frac{1}{\vartheta(v_\beta v_\alpha^{-1})}. \tag{156}$$

# 5 Exceptional Dirichlet

## 5.1 The Idea

In this section, we introduce another perspective on the supersymmetric ground states. The idea is to find a distinguished class of boundary conditions that are equivalent to a vacuum at $x^3 \to \pm\infty$, at least for the purpose of computations preserving the supercharge $Q$. This is illustrated in figure 11. This can lead to a convenient method to compute boundary amplitudes using supersymmetric localisation. Such boundary conditions preserving $\mathcal{N} = (2,2)$ supersymmetry were first considered in [15], and have been studied further in [20,66].

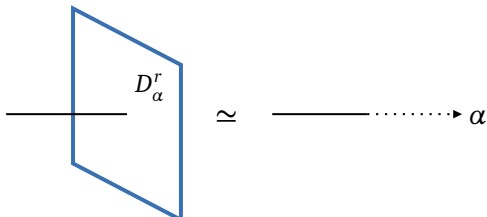

Figure 11: Vacuum boundary conditions.

## 5.2 $\mathcal{N} = (2,2)$ Exceptional Dirichlet

First consider generic mass parameters in some chamber $\mathfrak{C}$. We consider $\mathcal{N} = (2,2)$ boundary conditions $D_\alpha^r$, $D_\alpha^l$ which mimic a vacuum $\alpha$ at $x^3 \to \pm\infty$ for computations preserving the supercharge $Q$. Wrapping such boundary conditions on $E_\tau$ will produce boundary states in the same $Q$-cohomology class as the supersymmetric ground states $|\alpha\rangle_{\mathfrak{C}}$, ${}_{\mathfrak{C}}\langle\alpha|$.

In particular:

- The boundary amplitude ${}_{\mathfrak{C}}\langle\alpha|B\rangle = \langle D_\alpha^r|B\big|D_\alpha^r|B\rangle$ is the path integral on $E_\tau \times [-\ell, 0]$ with R-R boundary conditions with the right boundary condition $B$ at $x^3 = 0$ and the distinguished boundary condition $D_\alpha^l$ at $x^3 = -\ell$.

- The boundary amplitude $\langle B|\alpha\rangle_{\mathfrak{C}} = \langle B|D_\alpha^r\big|B|D_\alpha^r\rangle$ is the path integral on $E_\tau \times [0, \ell]$ with R-R boundary conditions with the left boundary condition $B$ at $x^3 = 0$ and the distinguished boundary condition $D_\alpha^r$ at $x^3 = \ell$.

If we can find UV gauge theory constructions of the distinguished boundary conditions $D_\alpha^l$, $D_\alpha^r$ this will provide a convenient method to compute boundary amplitudes using supersymmetric localisation on $E_\tau$ times an interval.

There are two important consistency checks on any proposal for such distinguished boundary conditions:

- From the perspective of a massive sigma model to $X$, the BPS equations for 2d $\mathcal{N} = (2,2)$ supersymmetry are gradient flow for the real moment map $h_m : X \to \mathbb{R}$. Therefore, to mimic a massive vacuum $\alpha$ at $x^3 \to \pm\infty$, the boundary conditions $D_\alpha^{r,l}$ must flow to Lagrangian branes supported on the repelling/attracting manifolds $X_\alpha^\mp \subset X$.

- By anomaly inflow, the boundary anomalies of $D_\alpha^r$, $D_\alpha^l$ must match the effective supersymmetric Chern-Simons couplings in the vacua $\alpha$. In our conventions, if we denote the boundary anomalies of $D_\alpha^r$ by $k_\alpha$, $k_\alpha^C$, $k_\alpha^H$, $\widetilde{k}_\alpha$, they should match the effective supersymmetric Chern-Simons couplings $\kappa_\alpha$, $\kappa_\alpha^C$, $\kappa_\alpha^H$, $\widetilde{\kappa}_\alpha$ introduced in section 2 in the chamber $\mathfrak{C}$. The boundary anomalies of $D_\alpha^l$ should be minus these.

If the latter condition is satisfied, it is guaranteed that overlaps with other boundary conditions $B^l, B^r$ will transform in the same way as the boundary amplitudes $\langle B^l | \alpha \rangle_{\mathfrak{C}}$, $_{\mathfrak{C}}\langle \alpha | B^r \rangle$ under large background gauge transformations.

### 5.2.1 Construction in Abelian Theories

A UV gauge theory construction of the distinguished boundary conditions $D^l_\alpha, D^r_\alpha$ in abelian 3d $\mathcal{N} = 4$ gauge theories was found in [15] and are known as exceptional Dirichlet. This proposal passes the first consistency check by construction. That they pass the second consistency check was proven in [20].

To construct exceptional Dirichlet boundary conditions, recall that in the vacuum $\alpha$ the real vector multiplet scalar $\sigma$ is uniquely determined by (having turned off $\epsilon$)

$$\rho_i \cdot \sigma + \rho_{H,i} \cdot m = 0 \,, \tag{157}$$

where $\varrho_i = (\rho_i, \rho_{H,i}, \rho_{t,i})$ with $i = 1, \dots, r$ are the set of weights associated to the vacuum $\alpha$. This in turn determines the effective real masses in the vacuum $\alpha$ of all remaining hypermultiplets. Fixing a chamber for the real mass parameters, we may split the hypermultiplet fields into those with zero, positive and negative mass.

For the right exceptional Dirichlet boundary condition, we define a splitting $\varepsilon^r_\alpha$ as in equation (122) such that $Y_{\varepsilon^r_\alpha}$ consist of hypermultiplet fields with negative real mass in the vacuum $\alpha$, or those which both have zero real mass and attain an expectation value in $\alpha$. Then the right exceptional Dirichlet boundary condition $D^r_\alpha$ is defined as follows.

- A $\mathcal{N} = (2,2)$ Dirichlet boundary condition for the vector multiplet. The boundary value of $\varphi$ is fixed by requiring the effective complex mass of all hypermultiplets with expectation values in $\alpha$ vanish. For $m_{\mathbb{C}} = 0$, $\varphi| = 0$.

- A Neumann-Dirichlet boundary condition for hypermultiplets

$$D_\perp X_{\varepsilon^r_\alpha}| = 0 \,, \quad Y_{\varepsilon^r_\alpha}| = \begin{cases} 0 \,, & \text{if } Y_{\varepsilon^r_\alpha} \text{ has negative real mass in } \alpha \,, \\ c \,, & \text{if } Y_{\varepsilon^r_\alpha} \text{ has zero real mass in } \alpha \text{ and attains a vev} \,, \end{cases} \tag{158}$$

  where $c$ is the expectation value in the vacuum $\alpha$.

The Higgs branch image of $D^r_\alpha$ is precisely the repelling set $X^-_\alpha \subset X$, since the hypermultiplet fields $X_{\varepsilon^r_\alpha}$ are exponentially suppressed under the inverse gradient flow.

For the left exceptional Dirichlet boundary condition $D^l_\alpha$, we instead take $Y_{\varepsilon^l_\alpha}$ to be chirals of positive real mass in the vacuum $\alpha$, or those which have zero real mass and attain a vev. Similarly, the image of the boundary condition $D^l_\alpha$ is $X^+_\alpha \subset X$.

Wrapping the theory on $E_\tau$, a feature common of both left and right boundary conditions is that the boundary gauge flat connection $u$ is fixed in terms of $z_H, z_t$ to the value determined by the vacuum, i.e. the unique solution of

$$\rho_i \cdot u + \rho_{H,i} \cdot z_H + \rho_{t,i} z_t = 0 \,. \tag{159}$$

This condition is not required to be invariant under shifts of $\mathbb{Z} + \tau\mathbb{Z}$, as the gauge symmetry is broken to a flavour symmetry at the boundary due to the Dirichlet boundary condition for the vector multiplet. We denote the distinguished value of the boundary holonomy $u_\alpha$, and also $s_\alpha = e^{2\pi i u_\alpha}$.

The above construction can be elegantly rephrased in terms of weights. The splitting corresponds to introducing the decomposition

$$Q = Q^+_\alpha \sqcup Q^-_\alpha \sqcup Q^0_\alpha \tag{160}$$

of the matter representation $Q = T^*R$ into weight spaces, which after the evaluation at the fixed point (32), correspond to positive, negative and zero weights respectively. Note that if a chiral $X$ has positive real mass, its corresponding weight in $T^*R$ is in fact $d/dX$ and thus corresponds to an element of $Q_\alpha^-$.

We note that

$$Q_\alpha^0 = \{\varrho_i, \varrho_i^* = -2e_t - \varrho_i \quad \text{for } i = 1, \ldots, r\}, \tag{161}$$

where $\varrho_i$ are the weights which label the vacuum $\alpha$, and $\varrho_i^*$ the weights corresponding to their partners from the same hypermultiplet. After evaluation at the vacuum, $w^{\varrho_i} = 1$ for $i = 1, \ldots, r$ (or equivalently $s = s_\alpha$), the character of $Q_\alpha^0$ is precisely cancelled by $\mu_{\mathbb{C}}$ and the gauge group quotient in (32). One also has

$$\varrho \in Q_\alpha^+ \quad \Leftrightarrow \quad \varrho^* \in Q_\alpha^- \tag{162}$$

again corresponding to pairs of chirals in the same hypermultiplet.

The polarisation is then rephrased in terms of weight spaces as

$$
\begin{aligned}
\varepsilon_\alpha^r : \quad & d/dX_{\varepsilon_\alpha^r} && \in && Q_\alpha^- \cup \{\varrho_i, i = 1, \ldots, r\}, \\
& d/dY_{\varepsilon_\alpha^r} && \in && Q_\alpha^+ \cup \{\varrho_i^*, i = 1, \ldots, r\}, \\
\varepsilon_\alpha^l : \quad & d/dX_{\varepsilon_\alpha^l} && \in && Q_\alpha^+ \cup \{\varrho_i, i = 1, \ldots, r\}, \\
& d/dY_{\varepsilon_\alpha^l} && \in && Q_\alpha^- \cup \{\varrho_i^*, i = 1, \ldots, r\}.
\end{aligned}
\tag{163}
$$

### 5.2.2 Anomalies

Let us derive the boundary anomalies of $D_\alpha^r$, and check they match the supersymmetric Chern-Simons levels in the vacuum $\alpha$.

First, the boundary anomalies involving the topological symmetry come purely from anomaly inflow and therefore trivially match the Chern-Simons couplings $\kappa_\alpha$, $\kappa_\alpha^C$. As in section 2, they coincide with the bilinear couplings appearing in the moment maps $h_m$ and $h_\epsilon$ at the fixed point $\alpha$. This matching was shown in detail for abelian theories in [20].

Let us therefore focus on anomalies arising from bulk fermions, computed using the results in [16]. We focus on $\mathcal{N} = 2$ flavour symmetries, and first compute the anomaly polynomial for zero boundary expectation values $c = 0$. The boundary condition initially supports an additional flavour symmetry $G_\partial$ with field strength $\mathbf{f}_\partial$, generated by global gauge transformations at the boundary. Using the description (163), the undeformed anomaly is

$$r\mathbf{f}_L^2 + \frac{1}{4} \prod_{\varrho \in Q_\alpha^+} (\varrho \cdot \mathbf{F})^2 + \frac{1}{4} \prod_{i=1}^r (\varrho_i \cdot \mathbf{F})^2 - \frac{1}{4} \prod_{\varrho \in Q_\alpha^-} (\varrho \cdot \mathbf{F})^2 - \frac{1}{4} \prod_{i=1}^r (\varrho_i^* \cdot \mathbf{F})^2, \tag{164}$$

where we have denoted $\mathbf{F} = (\mathbf{f}_\partial, \mathbf{f}_H, \mathbf{f}_L)$. Turning on the expectation value $c$, $G_\partial$ is broken and $\mathbf{f}_\partial$ is set to the value determined by solving $\varrho_i \cdot \mathbf{F} = 0$ for $i = 1, \ldots, r$. This is analogous to the substitution in the character (32).

Evaluating the above and re-introducing the terms from anomaly inflow, we recover

$$\mathcal{P}[D_\alpha^r] = \kappa_\alpha(\mathbf{f}_H, \mathbf{f}_C) + \kappa_\alpha^C(\mathbf{f}_L, \mathbf{f}_C) + \frac{1}{4} \sum_{\lambda \in \Phi_\alpha^+} (\lambda \cdot \mathbf{f})^2 - \frac{1}{4} \sum_{\lambda \in \Phi_\alpha^+} (\lambda \cdot \mathbf{f})^2, \tag{165}$$

which matches the Chern-Simons levels (37), i.e. $\mathcal{P}[D_\alpha^r] = \mathcal{P}_\alpha$.

One similarly recovers $\mathcal{P}[D_\alpha^l] = -\mathcal{P}_\alpha$, due to the opposite Lagrangian splitting and contribution from anomaly inflow due to the orientation of the boundary. This agrees with anomaly inflow from placing a massive supersymmetric vacuum $\alpha$ on the left.

### 5.2.3 Orthonormality

Another consistency check is to show that the interval partition functions of left and right exceptional Dirichlet boundary are orthonormal:

$$\langle D_\alpha^l | D_\beta^r \rangle = \delta_{\alpha,\beta} \,. \tag{166}$$

The configurations contributing to the localised path integral on $E_\tau \times I$ have constant profiles for the hypermultiplet scalars [19]. This implies that if $\alpha \neq \beta$, the boundary expectation values and holonomies are incompatible and break supersymmetry. If $\alpha = \beta$, taking $\ell \to 0$, the remaining fluctuating degrees of freedom consist of a neutral $\mathcal{N} = (2,2)$ chiral multiplet $(S, \bar{\Psi}_\varphi)$ of $R$-charge 0 and $r$ chiral multiplets $(\Phi_{X_{\varepsilon_\alpha^r}}, \Psi_{Y_{\varepsilon_\alpha^r}})$ for each $X_{\varepsilon_\alpha^r}$ of vanishing mass. The former is naively singular, with a contribution

$$\left( \frac{\vartheta(t^{-2})}{\vartheta(1)} \right)^r \,, \tag{167}$$

however this is cancelled by the contribution from the $r$ chiral multiplets

$$\prod_{i=1}^r \frac{\vartheta(w^{\varrho_i})}{\vartheta(w^{\varrho_i^*})} \Bigg|_{\substack{w^{\varrho_i} = 1 \\ i=1,..,r}} \,, \tag{168}$$

when evaluated at the value of the boundary holonomy of $G_\partial$ determined by both $D_\alpha^l$ and $D_\alpha^r$. This recovers the expected normalisation (166).

### 5.2.4 Example

Let us consider supersymmetric QED in the default chambers. The exceptional Dirichlet boundary conditions are given by Dirichlet for the vector multiplet and the following boundary condition for the hypermultiplets,

$$D_\alpha^r : \quad \begin{array}{ll} D_3 Y_\beta = 0\,, & X_\beta = c\delta_{\alpha\beta}\,, \quad \beta \le \alpha\,, \\ D_3 X_\beta = 0\,, & Y_\beta = 0\,, \qquad \beta > \alpha\,, \end{array} \tag{169}$$

$$D_\alpha^l : \quad \begin{array}{ll} D_3 X_\beta = 0\,, & Y_\beta = 0\,, \qquad \beta < \alpha\,, \\ D_3 Y_\beta = 0\,, & X_\beta = c\delta_{\alpha\beta}\,, \quad \beta \ge \alpha\,, \end{array} \tag{170}$$

where $c \neq 0$. In both cases, as $X_\alpha = c$, the effective real and complex mass parameters of this hypermultiplet field must vanish. This requires requires $\sigma = m_\alpha$ and $\varphi = m_{\alpha,\mathbb{C}}$, where $m_{\alpha,\mathbb{C}}$ are the complex mass parameters for $T_H$.[7]

Wrapping on $E_\tau$, the choice of vacuum $\alpha$ uniquely determines the value of the holonomy $u$ of the gauge field at the boundary

$$u = z_{H,\alpha} - z_t := u_\alpha \quad \Rightarrow \quad s v_\alpha^{-1} t = 1 \tag{171}$$

according to the hypermultiplet scalar $X_\alpha$ with a non-vanishing expectation value.

Next, we check the support of the right boundary conditions in $X = T^*\mathbb{CP}^{N-1}$. Let $\langle e_{\alpha_1} e_{\alpha_2} \cdots \rangle \subset \mathbb{CP}^{N-1} \subset X$ denote the projective subspace generated by the fundamental weights $e_{\alpha_1}, e_{\alpha_2}, \cdots$ of $T_H$. The supersymmetric vacua or fixed points are $\alpha = \langle e_\alpha \rangle$. Now

---

[7]Although we ultimately set the complex mass and FI parameters parameters to zero, which determines a fixed maximal torus $U(1)_H \times U(1)_C$, it is sometime convenient to include them when discussing boundary conditions and interfaces in flat space.

consider the subspaces $U_\alpha = \langle e_\alpha, \ldots, e_N \rangle$ and inclusions $\iota_\alpha : U_{\alpha+1} \to U_\alpha$. Then the right exceptional Dirichlet boundary conditions have support[8]

$$N^\perp U_\alpha - \iota_\alpha^* N^\perp U_\alpha, \qquad (172)$$

where $N^\perp$ denotes the co-normal bundle. This has the following interpretation:

- The coordinates $X_\beta$ for $\beta \geq \alpha$ parametrise the base $U_\alpha$.

- The coordinates $Y_\beta$ for $\beta < \alpha$ parametrise the co-normal directions to $U_\alpha$.

- The pull back $\iota_\alpha^* N^\perp U_\alpha$ is excluded due the constraint $X_\alpha = c \neq 0$.

This support is the attracting set $X_\alpha^-$ in the default chamber. Note that the closure of the attracting set is the whole co-normal bundle $\overline{X_\alpha^-} = N^\perp U_\alpha$. This is illustrated in terms of the hyper-toric diagram in figure 12. A similar argument shows that the left boundary conditions are supported on $X_\alpha^+$.

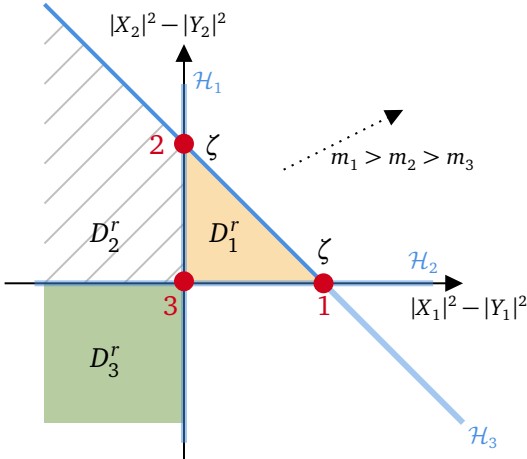

Figure 12: The hyper-toric diagram for the Higgs branch of supersymmetric QED with $N = 3$, see e.g. [15]. The slice $X_\beta Y_\beta = 0$ for all $\beta = 1, 2, 3$ is a fibration over $\mathbb{R}^2$ with typical fibre $(S^1)^2$, where one combination of the circles degenerates along each of the hyperplanes $\mathcal{H}_\beta$ defined by the vanishing of the hypermultiplet $(X_\beta, Y_\beta)$. The diagram is an illustration of the base. The fibre fully degenerates to a point at each vacuum $\{1, 2, 3\}$. The support of each exceptional Dirichlet boundary condition $D_\alpha^r$ in the default chamber is shown. For $D_1^r$ and $D_2^r$ we subtract the intersection with hyperplanes $\mathcal{H}_1$ and $\mathcal{H}_2$ respectively. For $D_3^r$ there is no such intersection. The direction of inverse gradient flow in the default chamber is shown by the dotted arrow.

Finally, we check the boundary anomalies reproduce the supersymmetric Chern-Simons coupling in the vacuum $\alpha$ in equations (23) and (25). We initially introduce separate field strengths $\mathbf{f}_V$ and $\mathbf{f}_A$ for the vector and axial R-symmetries. For the right boundary condition, starting with $c = 0$, the anomaly polynomial is

$$-\mathbf{f}_\partial \mathbf{f}_C - \mathbf{f}_V \mathbf{f}_A + \Big( \sum_{\beta \leq \alpha} (\mathbf{f}_H^\beta - \mathbf{f}_\partial) + \sum_{\beta > \alpha} (\mathbf{f}_\partial - \mathbf{f}_H^\beta) \Big) \mathbf{f}_A. \qquad (173)$$

---

[8]These Higgs branch images are found by taking the intersections of the Lagrangian submanifold of $T^* R \cong \mathbb{C}^{2N}$ specified by the splittings (169) and (170) with the moment map constraint $\mu_\mathbb{C} = 0$. One does not quotient by the gauge group because it is broken at the boundary by the Dirichlet boundary condition for the vector multiplet. Neither does one impose the real moment map $\mu_\mathbb{R} = \zeta_\mathbb{R}$, which arises as a D-term constraint, because it is absorbed into the boundary condition for $\sigma$. See section 3 of [15] for more details.

The first term comes from anomaly inflow from the mixed Chern-Simons term coupling the gauge symmetry to the topological symmetry, or equivalently the FI parameter. The second comes from gauginos, and the third from hypermultiplet fermions.

Now turning on $c \neq 0$, a combination of $T_H$, $U(1)_H$ and $G_\partial$ is broken. This can be seen as breaking $G_\partial$ and a re-definition of the boundary symmetries $T_H$ and $U(1)_V$,

$$
\begin{aligned}
J_H^\beta &= J_H^{\beta,\text{bulk}} + \delta^{\alpha\beta} J_G\,, \\
J_V &= J_{U(1)_H} - J_G\,.
\end{aligned}
\tag{174}
$$

In the anomaly polynomial, this sets $\mathbf{f}_\partial = \mathbf{f}_H^\alpha - \mathbf{f}_V$. Thus the boundary 't Hooft anomaly polynomial for the right exceptional Dirichlet boundary condition is

$$
\mathcal{P}[D_\alpha^r] = -\mathbf{f}_H^\alpha \mathbf{f}_C + \mathbf{f}_V \mathbf{f}_C + \Big( \sum_{\beta<\alpha}(\mathbf{f}_H^\beta - \mathbf{f}_H^\alpha) + \sum_{\beta>\alpha}(\mathbf{f}_H^\alpha - \mathbf{f}_H^\beta) \Big)\mathbf{f}_A + \mathbf{f}_V(2\alpha - N - 1)\mathbf{f}_A\,.
\tag{175}
$$

Note now $\mathbf{f}_H$ and $\mathbf{f}_V$ are field strengths for the boundary symmetries (174). This matches the supersymmetric Chern-Simons levels in the default chamber, after replacing $\mathbf{f}_A = -\mathbf{f}_L$ and $\mathbf{f}_V = \mathbf{f}_L$. Similarly $\mathcal{P}[D_\alpha^l] = -\mathcal{P}[D_\alpha^r]$.

## 5.3 $\mathcal{N} = (0,2)$ Exceptional Dirichlet

We now consider the case of vanishing mass parameters and construct boundary conditions $D_\alpha$, whose boundary states on $E_\tau$ lie in the same $Q$-cohomology class as the supersymmetric ground states $|\alpha\rangle$. Such boundary conditions will have the property that:

- The boundary amplitude $\langle \alpha | B | \alpha | B \rangle = \langle D_\alpha | B \rangle$ is given by the path integral on $E_\tau \times [-\ell, 0]$ with R-R boundary conditions with right boundary condition $B$ at $x^3 = 0$ and the distinguished boundary condition $D_\alpha$ at $x^3 = -\ell$.

- The boundary amplitude $\langle B | \alpha | B | \alpha \rangle = \langle B | D_\alpha \rangle$ is given by the path integral on $E_\tau \times [0, \ell]$ with R-R boundary conditions with left boundary condition $B$ at $x^3 = 0$ and the distinguished boundary condition $D_\alpha$ at $x^3 = \ell$.

Note that we have not introduced separate notation for left and right boundary conditions, as we will see momentarily that they take the same form. With explicit UV gauge theory constructions of such boundary conditions, this provides a convenient way to compute the boundary amplitudes via supersymmetric localisation. Recall these amplitudes glue to a section of a holomorphic line bundle on $E_T(X)$.

Three consistency checks on a proposal such a class of boundary conditions are:

- The boundary condition $D_\alpha$ should flow to Dirichlet boundary conditions in a massive sigma model to $X$ supported at the fixed points $\alpha \in X$.

- By anomaly inflow, the boundary anomalies should match the shifted supersymmetric Chern-Simons couplings given in (85),

$$
\begin{aligned}
|D_\alpha\rangle : & \quad \kappa_\alpha + \kappa_\alpha^C + \frac{1}{4}\sum_{\lambda\in\Phi_\alpha} \lambda\otimes\lambda\,, \\
\langle D_\alpha| : & \quad -\kappa_\alpha - \kappa_\alpha^C + \frac{1}{4}\sum_{\lambda\in\Phi_\alpha} \lambda\otimes\lambda\,,
\end{aligned}
\tag{176}
$$

which are independent of the chamber for the mass parameters.

- The corresponding boundary states are normalised with respect to $\mathcal{N} = (2,2)$ exceptional Dirichlet boundary conditions such that

$$
\begin{aligned}
\langle D_\alpha | B \rangle &= \langle D_\alpha^l | B \rangle \times \prod_{\lambda \in \Phi_\alpha^+} \vartheta(a^\lambda), \\
\langle B | D_\alpha \rangle &= \langle B | D_\alpha^r \rangle \times \prod_{\lambda \in \Phi_\alpha^-} \vartheta(a^\lambda),
\end{aligned}
\tag{177}
$$

in agreement with (104).

The last two compatibility checks are of course intimately related. The second ensures that the overlaps with boundary conditions $B^l$ and $B^r$ transform in the correct way under large background gauge transformations.

### 5.3.1 Construction

The $\mathcal{N} = (0,2)$ exceptional Dirichlet boundary condition $D_\alpha$ has a simple construction that is valid for any supersymmetric gauge theory, and is the same for both left and right. Decomposing into 3d $\mathcal{N} = 2$ supermultiplets, the boundary conditions are specified as follows:

- The vector multiplet has a Dirichlet boundary condition.

- The adjoint chiral multiplet $\varphi$ has a Neumann boundary condition (130).

- The chiral multiplets $X, Y$ are all assigned Dirichlet boundary conditions with boundary expectation values as in the vacuum $\alpha$, completely breaking the boundary $G_\partial$ symmetry. The remaining fluctuating degrees of freedom at the boundary are the $\mathcal{N} = (0,2)$ Fermi multiplets $\Psi_X$ and $\Psi_Y$.

The support of this boundary condition is the vacuum $\alpha \in X$.

We note that this construction is compatible with the formula (150), reproduced below

$$
\langle D_F(s_\alpha) | B \rangle = \langle \alpha | B \rangle,
\tag{178}
$$

relating boundary wavefunctions and amplitudes. The $\mathcal{N} = (0,2)$ exceptional Dirichlet boundary condition $D_\alpha$ is obtained from the $\mathcal{N} = (0,2)$ auxiliary Dirichlet boundary condition $D_F(s)$ by turning on expectations values for hypermultiplet scalars as in the vacuum $\alpha$. This fixes the boundary holonomy to $s = s_\alpha$.

### 5.3.2 Anomalies

Now consider the boundary anomalies of $D_\alpha$. Those involving the topological symmetry $T_C$ are the same as for the $\mathcal{N} = (2,2)$ exceptional Dirichlet boundary conditions and arise from anomaly inflow. The remaining anomalies arise from fermions in chiral multiplets surviving at the boundary. In terms of the matter representation $T^*R$, their contribution to the anomaly polynomial for $c = 0$ are

$$
-r\mathbf{f}_L^2 + \frac{1}{4} \prod_{\varrho \in T^*R} (\varrho \cdot \mathbf{F})^2,
\tag{179}
$$

where we have again denoted $\mathbf{F} = (\mathbf{f}_\partial, \mathbf{f}_H, \mathbf{f}_L)$. Turning on $c \neq 0$, we must again eliminate $\mathbf{f}_\partial$ by solving $\varrho_i \cdot \mathbf{F} = 0$. The above contribution becomes

$$
\frac{1}{4} \prod_{\lambda \in \Phi_\alpha} (\lambda \cdot \mathbf{f})^2.
\tag{180}
$$

Adding in the contributions from anomaly inflow, one obtains

$$|D_\alpha\rangle : \quad \mathcal{P} = +\kappa_\alpha(\mathbf{f}_H, \mathbf{f}_C) + \kappa_\alpha^C(\mathbf{f}_L, \mathbf{f}_C) + \frac{1}{4} \prod_{\lambda \in \Phi_\alpha} (\lambda \cdot \mathbf{f})^2 \,,$$

$$\langle D_\alpha| : \quad \mathcal{P} = -\kappa_\alpha(\mathbf{f}_H, \mathbf{f}_C) - \kappa_\alpha^C(\mathbf{f}_L, \mathbf{f}_C) + \frac{1}{4} \prod_{\lambda \in \Phi_\alpha} (\lambda \cdot \mathbf{f})^2 \,,$$
(181)

where the anomaly polynomials of the right and left boundary conditions are related by flipping the contributions $\kappa_\alpha$, $\kappa_\alpha^C$. This reproduces the expectation (176)

### 5.3.3 Orthogonality

We now check that the path integral on $E_\tau \times [0, \ell]$ with boundary conditions $D_\alpha$ at both boundaries reproduces the normalisation of supersymmetric ground states $|\alpha\rangle$ in (68). If $\alpha \neq \beta$, the partition function will vanish as before. Assuming the contrary, taking $\ell \to 0$, there is a contribution from Fermi multiplets $\Psi_X$, $\Psi_Y$,

$$\prod_{\varrho \in T^*R} \vartheta(w^\varrho)\Big|_{s=s_\alpha} = (\vartheta(1)\vartheta(t^2))^r \prod_{\lambda \in \Phi_\alpha} \vartheta(a^\lambda) \,.$$
(182)

The first factor has a zero of order $r$, but is cancelled by the remaining contribution of the adjoint chiral multiplets $S$ and $\Phi_\varphi$. In summary,

$$\langle D_\alpha | D_\beta | D_\alpha | D_\beta \rangle = \delta_{\alpha\beta} \prod_{\lambda \in \Phi_\alpha} \vartheta(a^\lambda) \,.$$
(183)

### 5.3.4 Example

Again let us consider supersymmetric QED in the default chambers. The $\mathcal{N} = (0, 2)$ exceptional Dirichlet boundary conditions are found by imposing a Dirichlet boundary condition for the 3d $\mathcal{N} = 2$ vector multiplet, a Neumann boundary condition for the chiral multiplet $\varphi = (\Phi_\varphi, \Gamma_\varphi)$, together with

$$\begin{array}{ll} D_3 \Psi_{X_\beta} = 0 \,, & X_\beta = c\delta_{\alpha\beta} \,, \\ D_3 \Psi_{Y_\beta} = 0 \,, & Y_\beta = 0 \,, \end{array} \quad \beta = 1, \dots, N \,,$$
(184)

for the hypermultiplets. The Higgs branch image is the fixed point $\alpha$, in which $X_\alpha \neq 0$. We do not impose $\varphi - m_{\alpha,\mathbb{C}} = 0$ as for $\mathcal{N} = (2, 2)$ exceptional Dirichlet boundary conditions, as here $\varphi$ is assigned a Neumann boundary condition.

Setting the expectation value to zero, $c = 0$, the boundary anomaly polynomial is, initially keeping track of separate $U(1)_V$ and $U(1)_A$ anomalies:

$$\mp \mathbf{f}_\partial \cdot \mathbf{f}_C - \frac{1}{2}\left(\mathbf{f}_V^2 + \mathbf{f}_A^2\right) + \frac{1}{4} \sum_{\beta=1}^{N} \left[ (\mathbf{f}_\partial - \mathbf{f}_H^\beta - \mathbf{f}_A)^2 + (-\mathbf{f}_\partial + \mathbf{f}_H^\beta - \mathbf{f}_A)^2 \right] \,,$$
(185)

where the $+$ sign is for $\langle\alpha|$ and the $-$ is for $|\alpha\rangle$. Turning on the expectation value, $c \neq 0$, we again make the redefinitions of boundary symmetries (174) and set $\mathbf{f}_\partial = \mathbf{f}_H^\alpha - \mathbf{f}_V$. Let us also pass to considering only 3d $\mathcal{N} = 2$ flavour symmetries, and set $\mathbf{f}_V = -\mathbf{f}_A = \mathbf{f}_L$. We obtain

$$\mp (\mathbf{f}_H^\alpha - \mathbf{f}_L) \cdot \mathbf{f}_C + \frac{1}{4} \sum_{\beta \neq \alpha} \left[ (\mathbf{f}_H^\beta - \mathbf{f}_H^\alpha)^2 + (-2\mathbf{f}_L - \mathbf{f}_H^\beta + \mathbf{f}_H^\alpha)^2 \right] \,,$$
(186)

which agrees with (176) with $\kappa_\alpha = -e_\alpha \otimes e_C$ and $\kappa_\alpha^C = e_t \otimes e_C$ as expected.

## 5.4 Boundary Amplitudes

The exceptional Dirichlet boundary conditions $D_\alpha^{l,r}$ and $D_\alpha$ provide an independent way to compute boundary amplitudes using supersymmetric localisation for the interval partition function [57]. In this section, we show such computations agree with the formulae derived via general consistency constraints in section 4.

We begin with the boundary amplitudes with vanishing mass parameters. From the explicit form of exceptional Dirichlet $D_\alpha$ boundary condition as $D_F(s)$ with boundary expectation values and $s = s_\alpha$, we immediately find that

$$\langle \alpha|B|\alpha|B\rangle = \langle D_F(s_\alpha)|B|D_F(s_\alpha)|B\rangle , \qquad \langle B|\alpha|B|\alpha\rangle = \langle B|D_F(s_\alpha)|B|D_F(s_\alpha)\rangle , \qquad (187)$$

as required by consistency in (150).

Now let us consider the boundary amplitudes with mass parameters in some chamber. Specialising to an abelian gauge theory, the exceptional Dirichlet boundary condition $D_\alpha^r$ is obtained from $D_\alpha$ by a coupling to the boundary superpotential

$$\int d^2x\, d\theta^+ \left( \Phi_\varphi| \cdot \Gamma_\varphi + \Psi_{X_{\varepsilon_\alpha^r}}| \cdot C_{\varepsilon^r} \right) , \qquad (188)$$

where $C_{\varepsilon_\alpha^r}$ and $\Gamma_\varphi$ are boundary chiral multiplets and an adjoint Fermi multiplet respectively. There is an implicit sum over each component of the polarisation $\varepsilon_\alpha^r$ that specifies the boundary condition $D_\alpha^r$. The boundary coupling implements a flip [16] of the boundary conditions for the chiral multiplets $\varphi, X, Y$ to those specified in $D_\alpha^r$.

The elliptic genus of the additional boundary contributions $C_\varepsilon, \Gamma_\varphi$ is

$$\vartheta(t^{-2})^r \prod_{\rho \in Q_\alpha^-} \frac{1}{\vartheta(w^\varrho)} \prod_{i=1}^r \frac{1}{\vartheta(w^{\varrho_i^*})} \Big|_{s=s_\alpha} = \prod_{\lambda \in \Phi_\alpha^-} \frac{1}{\vartheta(a^\lambda)} . \qquad (189)$$

We therefore have

$$\langle B|D_\alpha^r|B|D_\alpha^r\rangle = \prod_{\lambda \in \Phi_\alpha^-} \frac{1}{\vartheta(a^\lambda)} \langle B|D_\alpha|B|D_\alpha\rangle , \qquad (190)$$

which reproduces (177). A similar argument applies to boundary amplitudes involving the left exceptional Dirichlet boundary condition $D_\alpha^l$.

### 5.4.1 Lagrangian Branes

To illustrate the utility of this approach, we derive the formulae proposed in section 4.2.3 for the boundary amplitudes of boundary conditions flowing to smooth Lagrangian branes $L \subset X$ in the sigma model to $X$.

An $\mathcal{N} = (2,2)$ boundary condition $N_L$ flowing to a smooth Lagrangian brane $L \subset X$ can be constructed by imposing Neumann boundary conditions for the vector multiplet, together with a standard boundary condition for the hypermultiplet specified by a polarisation $\varepsilon_L$. The Lagrangian $L$ is the image under the hyperKähler quotient of the Lagrangian $Q_L \subset Q = T^*R$ specified by the polarisation.

First note that for the boundary amplitude with $D_\alpha$ to be non-vanishing, the polarisation $\varepsilon_L$ must be compatible with the vacuum $\alpha$. This means any hypermultiplet scalar which has a non-zero expectation value in the vacuum $\alpha$ is one of the $\{X_{\varepsilon_L}\}$, and not the $\{Y_{\varepsilon_L}\}$. Equivalently, this means that $Q_L$ contains the weights $\varrho_i$, $i = 1, \dots r$ which label the vacuum $\alpha$.

Let us then consider the boundary amplitude $\langle N_L|D_\alpha\rangle$. Sending the length of the interval to zero, the remaining degrees of freedom on $E_\tau$ consist of the $\mathcal{N} = (0,2)$ adjoint chiral multiplet

$\Phi_\varphi$ and the Fermi multiplets $\Psi_{Y_{\varepsilon_L}}$. The holonomy of the gauge connection is fixed to $s_\alpha$. Thus,

$$
\begin{aligned}
\langle N_L | D_\alpha | N_L | D_\alpha \rangle &= \frac{1}{\vartheta(t^{-2})^r} \prod_{\varrho \in Q_L} \vartheta(w^{\varrho *}) \bigg|_{s=s_\alpha} \\
&= \prod_{\lambda \in \Phi_\alpha(L)} \vartheta(a^{\lambda *}) \\
&= \prod_{\lambda \in \Phi_\alpha(L)^\perp} \vartheta(a^\lambda).
\end{aligned}
\tag{191}
$$

Note that the contribution from $\Phi_\varphi$ is cancelled by the Fermi multiplets paired with hyper-multiplet scalars which get expectation values in the vacuum $\alpha$. This reproduces the formula proposed in equation (109), which is the elliptic genus of Fermi multiplets parametrising $(T_\alpha L)^\perp \subset T_\alpha$. These boundary amplitudes represent the equivariant elliptic cohomology class of $L \subset X$. Note the result vanishes unless $\{\varrho_1, \dots, \varrho_r\} \subset Q_L$. The boundary amplitude $\langle D_\alpha | N_L | D_\alpha | N_L \rangle$ gives the same answer.

Let us now assume the left boundary condition $N_L$ is compatible with mass parameters in some chamber, and consider the boundary amplitude $\langle N_L | D_\alpha^r | N_L | D_\alpha^r \rangle$. Sending the length of the interval to zero, the remaining degrees of freedom on $E_\tau$ consist of the $\mathcal{N} = (2, 2)$ chiral multiplets compatible with both the splitting $\varepsilon_\alpha^r$ of $D_\alpha^r$ and $\varepsilon_L$ of $N_L$. Assuming $\{\varrho_1, \dots, \varrho_r\} \subset Q_L$, they are precisely the $\mathcal{N} = (2, 2)$ chirals containing scalars dual to the weights in

$$
Q_L \cap Q_\alpha^-.
\tag{192}
$$

At the fixed point, these become the weights in $\Phi_\alpha^-(L) \subset \Phi_\alpha$ of the tangent space $T_\alpha L \subset T_\alpha X^-$. Therefore $\langle N_L | D_\alpha^r | N_L | D_\alpha^r \rangle$ is given by the $(2, 2)$ elliptic genus

$$
\langle N_L | D_\alpha^r | N_L | D_\alpha^r \rangle = \prod_{\varrho \in Q_L \cap Q_\alpha^-} \frac{\vartheta(w^{\varrho *})}{\vartheta(w^\varrho)} \bigg|_{s=s_\alpha} = \prod_{\lambda \in \Phi_\alpha^-(L)} \frac{\vartheta(a^{\lambda *})}{\vartheta(a^\lambda)}
\tag{193}
$$

and similarly

$$
\langle D_\alpha^l | N_L | D_\alpha^l | N_L \rangle = \prod_{\varrho \in Q_L \cap Q_\alpha^+} \frac{\vartheta(w^{\varrho *})}{\vartheta(w^\varrho)} \bigg|_{s=s_\alpha} = \prod_{\lambda \in \Phi_\alpha^+(L)} \frac{\vartheta(a^{\lambda *})}{\vartheta(a^\lambda)}.
\tag{194}
$$

This reproduces the boundary amplitudes (108).

### 5.4.2 Example

Let us return to the Neumann boundary condition $N$ for supersymmetric QED that flows to the compact Lagrangian brane $L = \mathbb{CP}^{N-1} \subset X$. In the default chamber, this corresponds to the polarisation $\varepsilon_L = \{+, \cdots, +\}$.

In computing the boundary amplitudes with $D_\alpha$, the remaining degrees of freedom on $E_\tau$ are the $N$ Fermi multiplets $\Psi_{Y_\beta}$, $\beta = 1, \dots, N$, and a neutral chiral multiplet $\Phi_\varphi$. Thus

$$
\begin{aligned}
\langle D_\alpha | N_L | D_\alpha | N_L \rangle &= \frac{1}{\vartheta(t^{-2})} \prod_{\beta=1}^{N} \vartheta(t^{-1} s v_\beta^{-1}) \bigg|_{s=v_\alpha t^{-1}} \\
&= \prod_{\beta \neq \alpha} \vartheta(t^{-2} v_\alpha v_\beta^{-1}),
\end{aligned}
\tag{195}
$$

with an identical result for $\langle N_L | D_\alpha | N_L | D_\alpha \rangle$. This reproduces the previous formula (117). Note that the parameter $t$ corresponds to the left-moving boundary R-symmetry $T_L = U(1)_V - U(1)_A$,

where $U(1)_A$ is the boundary axial R-symmetry defined in section 4.2.4 and $U(1)_V$ is the boundary vector R-symmetry defined in (174).

The boundary condition $N$ is compatible with mass parameters in any chamber. In the overlap $\langle N|D_\alpha^r|N|D_\alpha^r\rangle$, there are no fluctuating degrees of freedom from the vector multiplet. The remaining contribution comes from the $\mathcal{N} = (2,2)$ chiral multiplets containing the scalars $X_\beta$ for $\beta > \alpha$ in the default chamber, evaluated at $u = u_\alpha$. Thus:

$$\langle N|D_\alpha^r|N|D_\alpha^r\rangle = \prod_{\beta > \alpha} \frac{\vartheta(t^{-2\frac{v_\alpha}{v_\beta}})}{\vartheta(\frac{v_\beta}{v_\alpha})}, \tag{196}$$

which reproduces the previous formula (115).

## 5.5 Wavefunctions of Exceptional Dirichlet

We now consider the wavefunctions of exceptional Dirichlet boundary conditions, either on the left or right, and at the origin or in a chamber of the mass parameter space,

$$\langle D_\alpha^l|D_C(s)\big|D_\alpha^l|D_C(s)\rangle, \qquad \langle D_C(s)|D_\alpha^r\big|D_C(s)|D_\alpha^r\rangle, \tag{197}$$

$$\langle D_\alpha|D_C(s)|D_\alpha|D_C(s)\rangle, \qquad \langle D_C(s)|D_\alpha|D_C(s)|D_\alpha\rangle. \tag{198}$$

Here we use the $\mathcal{N} = (0,2)$ boundary condition $D_C$ to write down wavefunctions, since it is compatible with the non-vanishing expectation value for $X_\alpha$.

A common feature is that, similarly to the auxiliary Dirichlet boundary conditions (140), the wavefunction vanishes unless $s = s_\alpha$. Provided $s = s_\alpha$, collapsing the interval there is fluctuating adjoint $\mathcal{N} = (0,2)$ chiral multiplet $S$, whose elliptic genus becomes singular. Using identical reasoning to the discussion surrounding equations (141) to (143) we replace this contribution by

$$(-)^r \frac{\delta^{(r)}(u - u_\alpha)}{\eta(q)^{2r}}, \tag{199}$$

where the delta function is understood as a contour prescription around an order $r$ pole at $u = u_\alpha$ with unit residue.

Let us first consider the wavefunction of the $\mathcal{N} = (0,2)$ exceptional Dirichlet boundary conditions $D_\alpha$. In addition to the contribution above, for both left and right boundary conditions, the only other contribution comes from the adjoint chiral multiplet $\Phi_\varphi$. In summary,

$$\langle D_\alpha|D_C(s)|D_\alpha|D_C(s)\rangle = \langle D_C(s)|D_\alpha|D_C(s)|D_\alpha\rangle = \frac{\delta^{(r)}(u - u_\alpha)}{(\vartheta(t^2)\eta(q)^2)^r}. \tag{200}$$

This wavefunction obeys the expected property (153).

For the $\mathcal{N} = (2,2)$ exceptional Dirichlet boundary conditions, there are additional contributions of the $\mathcal{N} = (0,2)$ chiral multiplets arising from the hypermultiplet fields with Neumann boundary conditions. For the right boundary condition $D_\alpha^r$, their contribution may be written in terms of $T^*R$ weights as

$$\prod_{i=1}^r \frac{1}{\vartheta(w^{\varrho_i^*})} \prod_{\varrho \in Q_\alpha^-} \frac{1}{\vartheta(w^\varrho)}\bigg|_{s=s_\alpha} = \frac{1}{\vartheta(t^{-2})^r} \prod_{\lambda \in \Phi_\alpha^-} \frac{1}{\vartheta(a^\lambda)}. \tag{201}$$

Thus

$$\langle D_C(s)|D_\alpha^r|D_C(s)|D_\alpha^r\rangle = \frac{\delta^{(r)}(u - u_\alpha)}{(\eta(q)^2\vartheta(t^2))^r \prod\limits_{\lambda \in \Phi_\alpha^-} \vartheta(a^\lambda)}, \tag{202}$$

and similarly

$$\left\langle D_\alpha^l | D_C(s) \middle| D_\alpha^l | D_C(s) \right\rangle = \frac{\delta^{(r)}(u - u_\alpha)}{(\eta(q)^2 \vartheta(t^2))^r \prod\limits_{\lambda \in \Phi_\alpha^+} \vartheta(a^\lambda)}. \tag{203}$$

These wavefunctions satisfy the relative normalisations of the boundary states created by the $\mathcal{N} = (2,2)$ and $\mathcal{N} = (0,2)$ boundary conditions (177).

It is easy to check that these wavefunctions are consistent with the formulae for boundary amplitudes (150). For example, let us consider the right $\mathcal{N} = (0,2)$ exceptional Dirichlet $D_\alpha$. Then

$$\begin{aligned}
\langle D_\alpha | B | D_\alpha | B \rangle &= \oint du (\eta(q)^2 \vartheta(t^2))^r \, \langle D_\alpha | D_C(s) | D_\alpha | D_C(s) \rangle \, \langle D_F(s) | B | D_F(s) | B \rangle \\
&= \langle D_F(s_\alpha) | B | D_F(s_\alpha) | B \rangle \,,
\end{aligned} \tag{204}$$

where $s_\alpha = e^{2\pi i u_\alpha}$ as before. This again is simply the evaluation of the 'Fermi' wavefunction of $B$ evaluated at values of the boundary holonomy $s$ fixed by the vacuum. The collection $\{\langle D_\alpha | B | D_\alpha | B \rangle\}_{\alpha \in \text{f.p.}}$ is guaranteed to glue to a single holomorphic line bundle on $E_T(X)$. Analogous statements hold for left and $(2,2)$ exceptional Dirichlet boundary conditions.

Let us also briefly check that the wavefunctions are consistent with orthogonality. Note that if $\alpha \neq \beta$, the contour prescriptions are not compatible. On the other hand, we have

$$\left\langle D_\alpha^l | D_\alpha^r \middle| D_\alpha^l | D_\alpha^r \right\rangle = \oint du \left( \eta(q)^2 \vartheta(t^2) \right)^r \mathcal{Z}_\Gamma \left\langle D_\alpha^l | D_C(s) \middle| D_\alpha^l | D_C(s) \right\rangle \left\langle D_C(s) | D_\alpha^r \middle| D_C(s) | D_\alpha^r \right\rangle = 1\,. \tag{205}$$

This is because in $Z_\Gamma$, at $u = u_\alpha$ there is a zero of order $r$ arising from $(0,2)$ Fermis which become neutral in the vacuum $\alpha$, multiplied by a factor $(\eta(q)^2 \vartheta(t^2))^r \prod_{\lambda \in \Phi_\alpha} \vartheta(a^\lambda)$. The $\eta(q)^2$ comes from the non-zero factor in $\vartheta(1)$, and $\vartheta(t^2)$ from the Fermi multiplets in the same hypermultiplets as the Fermis which become neutral. Recalling the description of $\delta^{(r)}(u - u_\alpha)$ in the exceptional Dirichlet wavefunctions as a contour prescription around a pole at $u = u_\alpha$ of order $r$ and unit residue, we recover the correct normalisation above.

### 5.5.1 Example

We return to the example of supersymmetric QED. The wavefunctions of $D_\alpha$ are obtained from (200) by setting $r = 1$.

For the wavefunction of $D_\alpha^r$, from equation (169) the remaining fluctuating degrees of freedom are Fermi multiplets $\Phi_{Y_\beta}$ for $\beta \leq \alpha$ and $\Phi_{X_\beta}$ for $\beta > \alpha$. These contribute

$$\prod_{\beta \leq \alpha} \frac{-1}{\vartheta(ts^{-1} v_\beta)} \prod_{\beta > \alpha} \frac{-1}{\vartheta(ts v_\beta^{-1})} \Bigg|_{s v_\alpha^{-1} t = 1} \tag{206}$$

and combining with the contribution of the chiral $S$,

$$\langle D_C(s) | D_\alpha^r | D_C(s) | D_\alpha^r \rangle = \frac{\delta(u - u_\alpha)}{\eta(q)^2 \vartheta(t^2) \prod\limits_{\beta < \alpha} \vartheta(t^{-2} \frac{v_\alpha}{v_\beta}) \prod\limits_{\beta > \alpha} \vartheta(\frac{v_\beta}{v_\alpha})}\,, \tag{207}$$

which agrees with the general formula with repelling weights $\Phi_\alpha^-$ in (46).

Similarly for $D_\alpha^l$, from (170), contributing degrees of freedom from the hypermultiplets are the $\mathcal{N} = (2,2)$ chirals $\Phi_{X_\beta}$ for $\beta < \alpha$ and $\Psi_{Y_\beta}$ for $\beta \geq \alpha$. So similarly

$$\langle D_\alpha^l | D_C(s) | D_\alpha^l | D_C(s) \rangle = \frac{\delta(u - u_\alpha)}{\eta(q)^2 \vartheta(t^2) \prod\limits_{\beta < \alpha} \vartheta(\frac{v_\beta}{v_\alpha}) \prod\limits_{\beta > \alpha} \vartheta(t^{-2} \frac{v_\alpha}{v_\beta})}\,, \tag{208}$$

where the denominator is constructed from the attracting weights $\Phi_\alpha^+$.

## 5.6 Mirror Image

An interesting problem is to understand the mirror dual of the $\mathcal{N} = (2,2)$ exceptional Dirichlet boundary conditions $\{D_\alpha\}$ of a theory $\mathcal{T}$. A hint is provided by the Coulomb branch image of these boundary conditions. Let us denote the Coulomb branch of $\mathcal{T}$ by $X^!$, known in the mathematics literature as the symplectic dual. The dual theory $\widetilde{\mathcal{T}}$ has a Higgs branch $X^!$, and Coulomb branch $X$. There is a canonical isomorphism of fixed points, and we denote them by $\{\alpha\}$ in both $X$ and $X^!$. These are the images of the same massive vacua of $\mathcal{T}$ on its Higgs and Coulomb branches.

The mirror dual boundary condition for $\widetilde{\mathcal{T}}$ must have a Higgs branch image in $X^!$ coinciding with the Coulomb branch image in $\mathcal{T}$ of $D_\alpha$. We consider the latter first, in the case of $\mathcal{T} = SQED[N]$. The generalisation to arbitrary abelian theories follows similarly.

For generic real masses, and zero complex masses, the Coulomb branch $X^!$ of $\mathcal{T}$ is the $A_{N-1}$ surface (a resolution of the singularity $\mathbb{C}^2/\mathbb{Z}_N$). The Coulomb branch image of $D_\alpha^r$ is supported on the fibre $\varphi = 0$, which we denote $\mathcal{S}_0$. This is a fibration, with base $\mathbb{R}$ parametrised by $\sigma$, and typical fibre $S^1$ by the dual photon $\gamma$. The photon circle shrinks where hypermultiplets become massless, i.e. when $\{\sigma - m_\alpha = 0\}$. These are the images of the vacua $\{\alpha\}$ on $X^!$. Thus $\mathcal{S}_0$ is a chain of $N-1$ copies of $\mathbb{P}^1$ capped on both ends by a copy of $\mathbb{C}$, see figure 13.

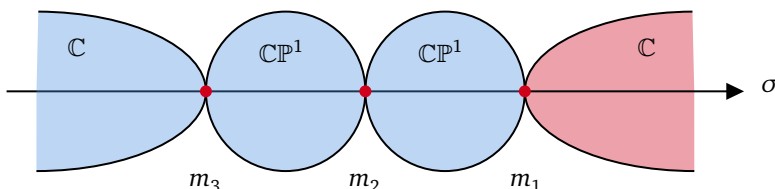

Figure 13: The slice $\mathcal{S}_0$ of the Coulomb branch of SQED[3], in our choice of $\mathfrak{C}_H$. The Coulomb branch image of $D_1^r$ necessarily has support contained in the blue region, and $D_1^l$ in the red. The real moment map for $T_C$, $h_\zeta = -\zeta \cdot \sigma$ decreases from left to right, in our choice of $\mathfrak{C}_C$.

The semi-classical analysis of section 3.4 of [15] implies the Coulomb branch support of $D_\alpha^r$ is necessarily contained in the locus of $\mathcal{S}_0$ where the effective real mass of $X_\alpha$ is negative, i.e. $\sigma - m_\alpha < 0$. This consists of the union of all the divisors $\mathbb{P}^1$ and copy of $\mathbb{C}$ to the left of $\alpha$ in $\mathcal{S}_0$ in figure 13. These are not just the repelling Lagrangian $(X^!)_\alpha^-$ for the Morse flow generated by $h_\zeta = -\zeta \cdot \mu_C = -\zeta \cdot \sigma$ on $X^!$, which is just the single $\mathbb{P}^1$ (or $\mathbb{C}$ for $\alpha = N$) to the left of $\alpha$. Analogous statements hold for $D_\alpha^l$.

Realising $X^!$ as the Higgs branch of $\widetilde{\mathcal{T}}$, this locus is precisely the support of the stable envelope [5,13] of the $A_{N-1}$ surface, in the chambers $\widetilde{\mathfrak{C}}_H$ and $\widetilde{\mathfrak{C}}_C$ of mirror mass and FI parameters under the identifications $(m_\alpha, \zeta) = (\widetilde{\zeta}_\alpha, -\widetilde{m})$. The Morse flow on $X^!$ is with respect to the function $\widetilde{h}_{\widetilde{m}} = \widetilde{m} \cdot \widetilde{\mu}_H$, which is equal to $h_C$ after these identifications.

The above suggests that the mirror boundary conditions in $\widetilde{\mathcal{T}}$ should be supported on the stable envelope associated to the fixed point $\alpha$. We have not proved this yet; the semi-classical analysis only implies a necessary condition for the Coulomb branch support. However, in the next section we produce the mirror dual boundary conditions explicitly by studying a mirror symmetry interface, and show that indeed the Higgs branch image of the mirror dual boundary condition coincides with stable envelopes.

For abelian theories, where $X$ and $X^!$ are hyper-toric varieties, the stable envelope contains the attracting Lagrangian $(X^!)_\alpha^-$, but is generically larger [67]. Thus the dual of exceptional Dirichlet is generically not another exceptional Dirichlet.

# 6 Enriched Neumann

An important construction in elliptic equivariant cohomology is elliptic stable envelopes, introduced in [5] and studied further in [6, 8, 68, 69]. This can be regarded as a nice basis of elliptic equivariant cohomology, or an assignment of an elliptic cohomology class to each fixed point.

We proceed to construct boundary conditions that realise elliptic stable envelopes in two steps. We first review the construction of a mirror symmetry interface between two mirror dual theories. Then, colliding the interface with the exceptional Dirichlet boundary conditions we considered in section 5, we recover a class of Neumann boundary conditions $N_\alpha^r$ enriched by boundary $\mathbb{C}^*$-valued $(2,2)$ chiral multiplets. These are labelled by vacua, and generate states in $Q$-cohomology which coincide with the elliptic stable envelopes. We check this proposition carefully by computing the wavefunctions and boundary amplitudes, showing agreement with the expressions in the mathematical literature. We give explicit constructions for abelian theories, leaving non-abelian examples to future work.

## 6.1 Mirror Symmetry Interface

We consider the $\mathcal{N} = (2,2)$ mirror symmetry interface between theories $\mathcal{T}, \widetilde{\mathcal{T}}$, which flows to a trivial interface in the IR whilst exchanging Higgs and Coulomb branch data [15]. The mirrors of boundary conditions can be constructed by collision with the interface. Our aim is to construct boundary conditions mirror to the exceptional Dirichlet $D_\alpha^r$ and $\widetilde{D}_\alpha^r$. This provides an alternative basis of supersymmetric ground states and reproduces the construction of elliptic stable envelopes from supersymmetric gauge theory.

### 6.1.1 Definition

We consider the mirror symmetry interface between a pair of mirror abelian gauge theories $\mathcal{T}$ on the left and $\widetilde{\mathcal{T}}$ on the right.

- The theory $\mathcal{T}$ has $G = U(1)^r$ and $R = \mathbb{C}^N$ with $r \times N$ charge matrix $Q$. It has a Higgs branch flavour symmetry $T_H = U(1)^{N-r}$ with $(N-r) \times N$ charge matrix $q$.

- The theory $\widetilde{\mathcal{T}}$ has $\widetilde{G} = U(1)^{N-r}$ and $\widetilde{R} = \mathbb{C}^N$ with $(N-r) \times N$ charge matrix $\widetilde{Q}$. It has a Higgs branch flavour symmetry $\widetilde{T}_H = U(1)^r$ with $r \times N$ charge matrix $\widetilde{q}$.

The Coulomb branch flavour symmetries are $T_C = \widetilde{T}_H$ and $\widetilde{T}_C = T_H$ and the mass and FI parameters are related by $(\widetilde{\zeta}, \widetilde{m}) = (m, -\zeta)$. The charge matrices are related by

$$\begin{pmatrix} \widetilde{q} \\ \widetilde{Q} \end{pmatrix}^T = \begin{pmatrix} Q \\ q \end{pmatrix}^{-1} \tag{209}$$

and further details can be found in [15]. Note that the multiplets in $\widetilde{\mathcal{T}}$ are twisted vector multiplets and twisted hypermultiplets, with the roles of $SU(2)_H$ and $SU(2)_C$ interchanged.

As is ubiquitous with $\mathcal{N} = (2,2)$ boundary conditions, the mirror symmetry interface depends on choice of polarisation $\varepsilon \in \{\pm\}^N$, see equation 122. We begin by imposing right Neumann boundary conditions $N_\varepsilon$ on $x^3 \leq 0$ and left Neumann boundary conditions $\widetilde{N}_{-\varepsilon}$ on $x^3 \geq 0$,

$$\begin{aligned} N_\varepsilon: \quad & D_3 X_\varepsilon = 0, \quad Y_\varepsilon = 0, \\ \widetilde{N}_{-\varepsilon}: \quad & D_3 \widetilde{X}_{-\varepsilon} = 0, \quad \widetilde{Y}_{-\varepsilon} = 0. \end{aligned} \tag{210}$$

We then introduce $N$ 2d $\mathcal{N} = (2,2)$ chiral multiplets $\Phi_i$ and their T-duals $\widetilde{\Phi}_i$ at $x^3 = 0$, whose scalar components $\phi_i, \widetilde{\phi}_i$ are valued in $\mathbb{C}^* \cong \mathbb{R} \times S^1$. Finally, we introduce boundary superpotentials[9]

$$W = \sum_{i=1}^{N} X_{\varepsilon_i} | e^{-\varepsilon_i \phi_i} - \phi_i (\widetilde{Q}^i \cdot | \widetilde{\varphi} + \widetilde{q}^i \cdot \widetilde{m}_{\mathbb{C}}),$$

$$\widetilde{W} = \sum_{i=1}^{N} e^{\varepsilon_i \widetilde{\phi}_i} | \widetilde{X}_{-\varepsilon_i} - (Q^i \cdot \varphi | + q^i \cdot m_{\mathbb{C}}) \widetilde{\phi}_i, \tag{211}$$

where $\varphi, \widetilde{\varphi}$ are the complex vector multiplet scalars. We have also introduced complex mass and FI parameters with $(\widetilde{\zeta}_{\mathbb{C}}, \widetilde{m}_{\mathbb{C}}) = (m_{\mathbb{C}}, -\zeta_{\mathbb{C}})$ as this is useful to discuss anomalies, with the understanding that we will ultimately set them to zero.

The superpotentials identify

- $G$, $T_H$ and $U(1)_V$ as translation symmetries of $\phi$, or winding symmetries of $\widetilde{\phi}$,

- $\widetilde{G}$, $T_C$ and $U(1)_A$ as winding symmetries of $\phi$, or translation symmetries of $\widetilde{\phi}$.

The charges of $\phi_i, \widetilde{\phi}_i$ are fixed by the first terms in the superpotentials (211). The second terms break these symmetries explicitly. They may be interpreted as $2d$ $\theta$-angles encoding the contribution to mixed anomalies. This interface reproduces the mirror map between Higgs and Coulomb branch chiral rings [15].

Another consistency check is that the mirror symmetry interface is anomaly-free for any polarisation $\varepsilon$. This requires anomalies of the Neumann boundary conditions $N_\varepsilon$ and $\widetilde{N}_{-\varepsilon}$ are cancelled by those of the boundary chiral multiplets $\Phi_i$, $\widetilde{\Phi}_i$. The latter can be seen from the superpotentials. Recall that the combination

$$\widetilde{Q}^i \cdot \widetilde{\varphi} + \widetilde{q}^i \cdot \widetilde{m}_{\mathbb{C}} \tag{212}$$

appearing in the superpotential $W$ is the effective complex mass of $\widetilde{\phi}_i$ and encodes its charges under its translation symmetries $\widetilde{G}$, $T_C$. The superpotential $W$ displays a mixed anomaly under translation symmetries $G$, $T_H$ or $U(1)_V$ of $\phi_i$. Similarly,

$$Q^i \cdot \varphi + q^i \cdot m_{\mathbb{C}} \tag{213}$$

is the effective complex mass for $\phi_i$ and encodes its charges under its translation symmetries $G$, $T_H$ and one obtains the same mixed anomaly by shifting $\widetilde{\phi}_i$ under $\widetilde{G}$, $T_C$ or $U(1)_A$.

Overall, the contribution to the mixed anomaly from $\phi_i$, $\widetilde{\phi}_i$ is

$$\sum_{i=1}^{N} \left( \mathbf{f}_V - \varepsilon_i (Q^i \cdot \mathbf{f} + q^i \cdot \mathbf{f}_H) \right) \left( \mathbf{f}_A + \varepsilon_i (\widetilde{Q}^i \cdot \widetilde{\mathbf{f}} + \widetilde{q}^i \cdot \widetilde{\mathbf{f}}_H) \right) - 2N \mathbf{f}_A \mathbf{f}_V. \tag{214}$$

We have denoted by $\widetilde{\mathbf{f}}$ the field strength for $\widetilde{G}$ and identify $\widetilde{\mathbf{f}}_H = -\mathbf{f}_C$ under mirror symmetry. In the summation, the first and second factors are the charges of the operators $e^{-\varepsilon_i \phi_i}$ and $e^{\varepsilon_i \widetilde{\phi}_i}$, which create translation and winding modes respectively for $\phi_i$. The final term comes from the fermions $\psi_\pm^{\phi_i}$ in the $\mathcal{N} = (2,2)$ chiral multiplet $\Phi_i$, which are only charged under $R$ symmetries.[10] This precisely cancels the mixed anomalies coming from the boundary conditions $N_\varepsilon$ and $\widetilde{N}_{-\varepsilon}$ at the interface.

---

[9]We note that such multiplets and similar superpotentials appear ubiquitously in Hori-Vafa mirror symmetry [70]. In the superpotentials we have represented $\mathcal{N} = (2,2)$ multiplets by their scalar components, and will continue to make the same abuse of notation in the remainder of this section.

[10]Note that the action of T-duality on $\Phi_i$ dualises the scalar $\phi_i$, and leaves the fermions $\psi_\pm^{\phi_i}$ alone. The dual scalar $\widetilde{\phi}_i$ and the same fermions comprise the dual $\mathcal{N} = (2,2)$ twisted chiral multiplet to $\Phi_i$.

### 6.1.2 Example

We will consider the mirror symmetry interface for $\mathcal{T}$ supersymmetric QED with $N$ flavours. The mirror $\widetilde{\mathcal{T}}$ is an abelian $A_{N-1}$-type quiver gauge theory with $G = U(1)^{N-1}$, and $X^!$ is the resolution of the $A_{N-1}$ singularity $\mathbb{C}^2/\mathbb{Z}_N$. We choose charge matrices corresponding to quiver conventions, as illustrated in figure 14.

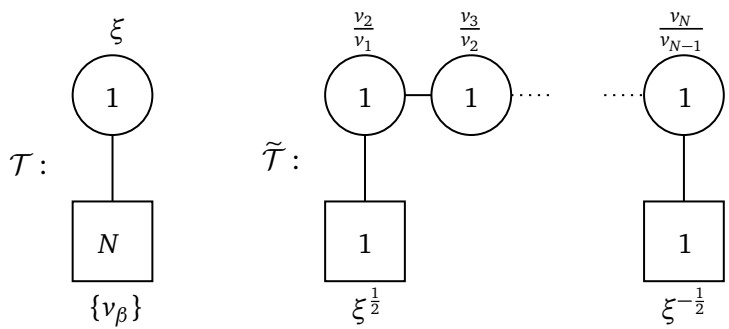

Figure 14: Quiver diagrams for supersymmetric QED and its mirror.

The mass and FI parameters are identified according to $(m_\alpha, \zeta) = (\widetilde{\zeta}_\alpha, -\widetilde{m})$ and similarly for the chambers. In the default chambers $\mathfrak{C}_C = \{\zeta > 0\}$ and $\mathfrak{C}_H = \{m_1 > m_2 > \ldots > m_N\}$ for supersymmetric QED, the vacua $\alpha$ in the mirror are

$$\begin{aligned}
\widetilde{Y}_\beta &= \sqrt{m_\beta - m_\alpha} \quad &\text{for } \beta &< \alpha, \\
\widetilde{X}_\beta &= \sqrt{m_\alpha - m_\beta} \quad &\text{for } \beta &> \alpha,
\end{aligned} \tag{215}$$

with all other hypermultiplet fields vanishing.

Let us derive the weight space decomposition of the tangent space $T_\alpha X^!$. We denote the matter representation of $\widetilde{G} \times \widetilde{T}_H \times T_t$ of $\widetilde{T}$ by $T^*\widetilde{R}$, and work with the fundamental weights of $\mathcal{T}$. Note $T_t$ transforms the holomorphic symplectic form of $X^!$ with weight $-2$.

$$\text{Ch } T^*\widetilde{R} = t \sum_{\beta=1}^{N} \left( \frac{\tilde{s}_\beta}{\tilde{s}_{\beta-1}} + \frac{\tilde{s}_{\beta-1}}{\tilde{s}_\beta} \right), \tag{216}$$

$$\text{Ch }\widetilde{\mathfrak{g}}_{\mathbb{C}} + t^2 \text{Ch }\widetilde{\mathfrak{g}}_{\mathbb{C}}^* = (N-1)(1+t^2), \tag{217}$$

where we have identified $\tilde{s}_0 = \xi^{\frac{1}{2}}$ and $\tilde{s}_N = \xi^{-\frac{1}{2}}$. Note that $\xi$ is the formal parameter we have associated to $T_C$, and will be identified as $e^{2\pi i z_C}$ in our computations on $E_\tau \times I$. The choice of vacuum (215) determines

$$\begin{aligned}
t^{-1}\frac{\tilde{s}_{\beta-1}}{\tilde{s}_\beta} &= 1, \quad \beta < \alpha, \\
t^{-1}\frac{\tilde{s}_\beta}{\tilde{s}_{\beta-1}} &= 1, \quad \beta > \alpha,
\end{aligned} \quad \Rightarrow \quad \begin{aligned}
\tilde{s}_\beta &= t^{-\beta}\xi^{\frac{1}{2}}, \quad &\beta < \alpha, \\
\tilde{s}_\beta &= t^{-(N-\beta)}\xi^{-\frac{1}{2}}, \quad &\beta \geq \alpha.
\end{aligned} \tag{218}$$

Thus one has

$$\begin{aligned}
\text{Ch } T_\alpha X^! &= \text{Ch } T^*\widetilde{R} - \text{Ch }\widetilde{\mathfrak{g}}_{\mathbb{C}} + t^2\text{Ch }\widetilde{\mathfrak{g}}_{\mathbb{C}}|_\alpha \\
&= \xi^{-1}t^{-N+2\alpha} + \xi t^{N-2\alpha+2}.
\end{aligned} \tag{219}$$

Therefore in our choice of $\mathfrak{C}_C$:

$$\widetilde{\Phi}_\alpha^+ = \{e_C + (N-2\alpha+2)e_t\}, \qquad \widetilde{\Phi}_\alpha^- = \{-e_C + (-N+2\alpha)e_t\}. \tag{220}$$

## 6.2 Enriched Neumann

We now use the mirror symmetry interface to derive the mirror of the collection of right exceptional Dirichlet boundary conditions $\widetilde{D}_\alpha^r$. The mirror is a collection of right Neumann boundary conditions $N_\alpha^r$, defined by a polarisation and enriched by $\mathbb{C}^*$-valued chiral multiplets coupled via boundary superpotentials and twisted superpotentials. We refer to these boundary conditions as enriched Neumann.

It was conjectured in [15] that the mirror of exceptional Dirichlet boundary conditions are again exceptional Dirichlet. This proposal reproduces the same boundary chiral rings, anomalies and Higgs branch support for generic complex FI parameters. However, it fails to capture the correct Higgs branch support with vanishing complex FI parameter and quarter-BPS boundary operators contributing to the general half superconformal index. Enriched Neumann may flow to exceptional Dirichlet in special cases, but this is not generically the case.

In summary, as anticipated in appendix B of [15], the mirror of exceptional Dirichlet boundary conditions are enriched Neumann boundary conditions.

### 6.2.1 Definition and Derivation

We focus here on the case where $\mathcal{T}$ is supersymmetric QED, the extension to general abelian theories considered in [15] follows straightforwardly. We use the default chambers $\mathfrak{C}_C = \{\zeta > 0\}$ and $\mathfrak{C}_H = \{m_1 > m_2 > \ldots > m_N\}$ with corresponding chambers $\widetilde{\mathfrak{C}}_C$, $\widetilde{\mathfrak{C}}_H$ in the mirror obtained under the identifications $(m_\alpha, \zeta) = (\widetilde{\zeta}_\alpha, -\widetilde{m})$.

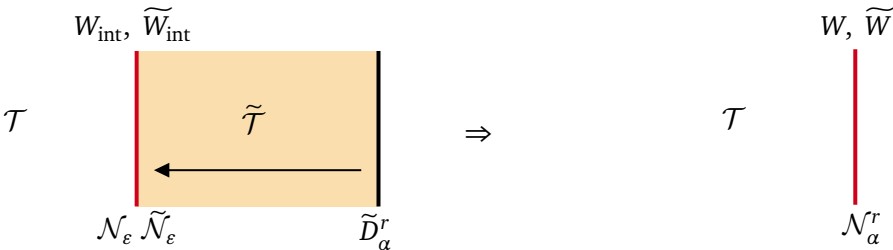

Figure 15: Collision of exceptional Dirichlet boundary conditions in $\widetilde{\mathcal{T}}$ with the mirror interface to derive enriched Neumann boundary conditions in $\mathcal{T}$.

We start from the right exceptional Dirichlet boundary conditions for $\widetilde{\mathcal{T}}$ and collide with the mirror interface to derive right exceptional Neumann boundary conditions for $\mathcal{T}$. This is shown in figure 15. The right exceptional Dirichlet boundary conditions $\widetilde{D}_\alpha^r$ in the default chamber are

$$
\begin{aligned}
D_3 \widetilde{X}_\beta = 0, \quad & \widetilde{Y}_\beta = \widetilde{c}_\beta, \quad \beta = 1, \ldots, \alpha - 1, \\
D_3 \widetilde{Y}_\alpha = 0, \quad & \widetilde{X}_\alpha = 0, \\
D_3 \widetilde{Y}_\beta = 0, \quad & \widetilde{X}_\beta = \widetilde{c}_\beta, \quad \beta = \alpha + 1, \ldots N,
\end{aligned}
\tag{221}
$$

where $\widetilde{c}_\beta$ are non-vanishing constants. The boundary condition on the complex scalars in the twisted vector multiplet requires

$$
\widetilde{\varphi}_\beta = \begin{cases} +\frac{1}{2}\zeta_{\mathbb{C}} & , \quad \beta < \alpha, \\ -\frac{1}{2}\zeta_{\mathbb{C}} & , \quad \beta \geq \alpha, \end{cases}
\tag{222}
$$

where $\zeta_{\mathbb{C}} = -\widetilde{m}_{\mathbb{C}}$ is the complex FI parameter.

It is convenient to tailor the choice of polarisation in the mirror symmetry interface with the exceptional Dirichlet boundary condition. To collide with the right exceptional Dirichlet

$\widetilde{D}_\alpha^r$ associated to the vacuum $\alpha$ it is convenient to use the polarisation $\varepsilon = (+\ldots+-\ldots-)$, where the first $-$ is in position $\alpha$. The interface then has superpotentials

$$W_{\text{int}} = \sum_{\beta<\alpha}\left(X_\beta|e^{-\phi_\beta} - \phi_\beta|\widetilde{M}_{\mathbb{C}}^\beta\right) + \sum_{\beta\geq\alpha}\left(Y_\beta|e^{\phi_\beta} - \phi_\beta|\widetilde{M}_{\mathbb{C}}^\beta\right),$$

$$\widetilde{W}_{\text{int}} = \sum_{\beta<\alpha}\left(e^{\widetilde{\phi}_\beta}|\widetilde{Y}_\beta - M_{\mathbb{C}}^\beta|\widetilde{\phi}_\beta\right) + \sum_{\beta\geq\alpha}\left(e^{-\widetilde{\phi}_\beta}|\widetilde{X}_\beta - M_{\mathbb{C}}^\beta|\widetilde{\phi}_\beta\right),$$

(223)

where $M_{\mathbb{C}}^\beta = \varphi - m_{\alpha,\mathbb{C}}$, $\widetilde{M}_{\mathbb{C}}^\beta = \widetilde{\varphi}_\beta - \widetilde{\varphi}_{\beta-1}$ denote the total complex masses of $X^\beta$, $\widetilde{X}^\beta$. Here we have abused notation and denoted $\widetilde{\varphi}_0 = -\widetilde{\varphi}_N = \frac{1}{2}\zeta_{\mathbb{C}}$.

On collision with the right exceptional Dirichlet boundary condition $\widetilde{D}_\alpha^r$, we obtain a right Neumann boundary condition for $\mathcal{T}$, coupled to the boundary (twisted) superpotential

$$W = \sum_{\beta<\alpha}\left(X_\beta|e^{-\phi_\beta}\right) + \left(Y_\alpha|e^{\phi_\alpha} - (-\zeta_{\mathbb{C}})\phi_\alpha\right) + \sum_{\beta>\alpha}\left(Y_\beta|e^{\phi_\beta}\right),$$

$$\widetilde{W} = \sum_{\beta<\alpha}\left(\widetilde{c}_\beta e^{\widetilde{\phi}_\beta} - M_{\mathbb{C}}^\beta|\widetilde{\phi}_\beta\right) + \left(-M_{\mathbb{C}}^\alpha\widetilde{\phi}_\alpha\right) + \sum_{\beta>\alpha}\left(\widetilde{c}_\beta e^{-\widetilde{\phi}_\beta} - M_{\mathbb{C}}^\beta|\widetilde{\phi}_\beta\right).$$

(224)

Colliding with the exceptional Dirichlet boundary breaks the gauge symmetry $\widetilde{G} = U(1)^{N-1}$ at the interface, shifting the $U(1)_A$ and $T_C$ weights of boundary operators charged under $\widetilde{G}$. From the perspective of $\mathcal{T}$, this can be seen as redefinition of the boundary $U(1)_A$ and $T_C$ symmetries by the addition of a generator of $_\partial|\widetilde{G}$. This redefinition only alters the charges of the boundary operators constructed from $\Phi_\alpha, \widetilde{\Phi}_\alpha$, which are are modified to those in table 1. The fermions $\psi_\pm^{\phi_\alpha}$ are not charged under the gauge symmetry $\widetilde{G}$ and are therefore unaffected by this shift.

Table 1: Charges of operators in the boundary chiral multiplet $\Phi_\alpha$ in the construction of right enriched Neumann boundary condition $N_\alpha^r$ in supersymmetric QED.

| Operator | $G$ | $T_{H,\alpha}$ | $T_C$ | $U(1)_V$ | $U(1)_A$ | Index |
|----------|-----|----------------|-------|----------|----------|-------|
| $e^{\phi_\alpha}$ | 1 | $-1$ | 0 | 1 | 0 | $sv_\alpha^{-1}t$ |
| $e^{-\widetilde{\phi}_\alpha}$ | 0 | 0 | 1 | 0 | $-(N-2\alpha)$ | $\xi t^{N-2\alpha}$ |
| $\psi_\pm^{\phi_\alpha}$ | 0 | 0 | 0 | $-1$ | $\mp 1$ | $1, t^{-2}$ |

We can now integrate out the boundary chiral multiplets $\Phi_\beta$ with $\beta \neq \alpha$. Let us do this for $\beta > \alpha$; the $\beta < \alpha$ case is treated similarly. The term $\widetilde{c}_\beta e^{-\widetilde{\phi}_\beta}$ in the twisted superpotential removes $e^{-\phi_\beta}$, promoting $\eta_\beta \equiv e^{\phi_\beta}$ to a $\mathbb{C}$-valued chiral multiplet. We can also integrate out $\widetilde{\phi}_\beta$ using $\partial W/\partial\widetilde{\phi}_\beta = 0$. The remaining contributions to the superpotentials are

$$W_\beta = Y_\beta|\eta_\beta,$$

$$\widetilde{W}_\beta = M_{\mathbb{C}}^\beta|\left(\log M_{\mathbb{C}}^\beta - 1\right) - M_{\mathbb{C}}^\beta|\log(-\widetilde{c}_\beta).$$

(225)

The boundary superpotential imposes $X_\beta| = 0$. The first term in the twisted superpotential is the 1-loop correction from integrating out the boundary chiral multiplet $\eta_\beta$. In combination they implement a flip of the boundary condition for the hypermultiplet $(X_\alpha, Y_\beta)$ - see section 5.3 of [15]. The remaining second term in the twisted superpotential contributes to the complexified $2d$ FI parameter.

This remaining boundary is a right Neumann boundary condition (110) for the vector multiplet, together with the Lagrangian splitting for hypermultiplets

$$D_\perp Y_\beta = 0, \quad X_\beta = 0, \quad \beta \leq \alpha,$$

$$D_\perp X_\beta = 0, \quad Y_\beta = 0, \quad \beta > \alpha,$$

(226)

coupled to boundary $\mathbb{C}^*$-valued chirals multiplets $\Phi_\alpha, \widetilde{\Phi}_\alpha$ with charges summarised in table 1 via the boundary superpotential and twisted superpotential

$$W = Y_\alpha |e^{\phi_\alpha} - (-\zeta_\mathbb{C})\phi_\alpha, \quad \widetilde{W} = -(\varphi| - m_{\alpha,\mathbb{C}})\widetilde{\phi}_\alpha - t_{2d}\varphi|. \qquad (227)$$

The boundary condition supports a complexified 2d FI parameter[11] $t_{2d}$, such that the boundary conditions on the real scalar $\sigma$ and dual photon $\gamma$ are

$$\sigma + i\gamma = t_{2d}. \qquad (228)$$

For later, we note that due to the first term in the boundary superpotential only $e^{-\mathfrak{m}\widetilde{\phi}_\alpha}$ with $\mathfrak{m} \geq 0$ are genuine boundary chiral operators. We refer to this collection of boundary conditions as enriched Neumann and denote them by $N_\alpha^r$.

We can similarly construct the left enriched Neumann boundary conditions $N_\alpha^l$, mirror to the exceptional Dirichlet boundary conditions $\widetilde{D}_\alpha^l$. We find that $N_\alpha^l$ is defined by a left $\mathcal{N} = (2,2)$ Neumann boundary condition for the vector multiplet (110), together with the Lagrangian splitting for the hypermultiplets

$$\begin{aligned} D_\perp X_\beta = 0, \quad Y_\beta = 0, \quad \beta < \alpha, \\ D_\perp Y_\beta = 0, \quad X_\beta = 0, \quad \beta \geq \alpha, \end{aligned} \qquad (229)$$

coupled to a boundary $\mathbb{C}^*$-valued chiral multiplet $\Phi_\alpha, \widetilde{\Phi}_\alpha$ with charges summarised in table 2 via boundary superpotentials

$$W = |Y_\alpha e^{\phi_\alpha} - \zeta_\mathbb{C}\phi_\alpha, \quad \widetilde{W} = -\widetilde{\phi}_\alpha(|\varphi - m_{\alpha,\mathbb{C}}). \qquad (230)$$

Note the opposite sign of the contribution $\zeta_\mathbb{C}\phi_\alpha$ to the boundary superpotential compared to (227), which reflects a twist operator with opposite topological charge.

Table 2: Charges of operators in the boundary chiral multiplet $\Phi_\alpha$ in the construction of right enriched Neumann boundary condition $N_\alpha^l$ in supersymmetric QED.

| Operator | $G$ | $T_{H,\alpha}$ | $T_C$ | $U(1)_V$ | $U(1)_A$ | Index |
|----------|-----|----------------|-------|----------|----------|-------|
| $e^{\phi_\alpha}$ | $-1$ | $1$ | $0$ | $1$ | $0$ | $s\nu_\alpha^{-1}t$ |
| $e^{-\widetilde{\phi}_\alpha}$ | $0$ | $0$ | $-1$ | $0$ | $N+2-2\alpha$ | $\xi^{-1}t^{-N-2+2\alpha}$ |
| $\psi_\pm^{\phi_\alpha}$ | $0$ | $0$ | $0$ | $-1$ | $\mp 1$ | $1, t^{-2}$ |

We now provide two consistency checks by computing the boundary 't Hooft anomalies and the Higgs branch support of the enriched Neumann boundary conditions $N_\alpha^r$.

### 6.2.2 Anomalies

We now compute the boundary mixed 't Hooft anomalies for the enriched Neumann boundary conditions $N_\alpha^r$. As the mirror interface is trivial in the IR and anomaly free, this will agree by construction with the anomalies of exceptional Dirichlet boundary conditions $\widetilde{D}_\alpha^r$ in the mirror. Moreover, as the anomalies of exceptional Dirichlet boundary conditions reproduce the effective Chern-Simons terms in the vacua $\alpha$, which match under mirror symmetry, this must reproduce the anomalies of the exceptional Dirichlet $D_\alpha^r$ in the same theory. We will check this for supersymmetric QED.

---

[11]In the above derivation, $t_{2d} = -\sum_{\beta<\alpha} \log(-\widetilde{c}_\beta) + \sum_{\beta>\alpha} \log(-\widetilde{c}_\beta)$.

A key observation is that the $\mathbb{C}^*$-valued chiral multiplets have mixed anomalies between translation and winding symmetries. This can be seen, for example, from the twisted superpotential, which contains a coupling between $\widetilde{\Phi}_\alpha$ and the twisted chiral field strength multiplet containing $\varphi$, as well as the background multiplet for $U(1)_\alpha \subset T_H$ containing $m_{\alpha,\mathbb{C}}$. Thus $\widetilde{\Phi}_\alpha$ can be interpreted as a dynamical complexified theta angle. The mixed anomalies can be seen by shifting $\widetilde{\phi}_\alpha$ or performing a gauge or flavour transformation.

With this in mind, we compute the anomaly polynomial of $N_\alpha^r$. First, the contributions from the boundary conditions on the bulk vector and hypermultiplets are computed following [16], with the result

$$\mathcal{P}(N_\alpha^r)_{\text{bulk}} = -\mathbf{f}\mathbf{f}_C + \mathbf{f}_V\mathbf{f}_A + \Big( -\sum_{\beta\leq\alpha}(\mathbf{f}-\mathbf{f}_H^\beta) + \sum_{\beta>\alpha}(\mathbf{f}-\mathbf{f}_H^\beta)\Big)\mathbf{f}_A. \tag{231}$$

The first term comes from anomaly inflow from the bulk mixed Chern-Simons term between $G \cong U(1)$ to $T_C \cong U(1)$, or equivalently the bulk FI coupling. The second term comes from the gauginos surviving on Neumann boundary condition for the vector multiplet. The remaining terms arise from fermions in the hypermultiplets. Second, the contribution of the boundary $\mathbb{C}^*$-valued matter can be computed using table 1,

$$\mathcal{P}(N_\alpha^r)_{\text{bdy}} = (\mathbf{f}-\mathbf{f}_H^\alpha + \mathbf{f}_V)(\mathbf{f}_C - (N-2\alpha)\mathbf{f}_A) - 2\mathbf{f}_A\mathbf{f}_V, \tag{232}$$

where the first term comes from mixed translation-winding anomalies of $\phi_\alpha$ and $\widetilde{\phi}_\alpha$, and the second from boundary fermions $\psi_\pm^{\phi_\alpha}$. Summing the two contributions,

$$\begin{aligned}
\mathcal{P}[N_\alpha^r] = &-\mathbf{f}_H^\alpha\mathbf{f}_C + \Big( \sum_{\beta<\alpha}(\mathbf{f}_H^\beta - \mathbf{f}_H^\alpha) + \sum_{\beta>\alpha}(\mathbf{f}_H^\alpha - \mathbf{f}_H^\beta)\Big)\mathbf{f}_A \\
&+ \mathbf{f}_V\mathbf{f}_C + \mathbf{f}_V(2\alpha - N - 1)\mathbf{f}_A.
\end{aligned} \tag{233}$$

Note that, beautifully, the gauge anomalies from bulk supermultiplets have been completely cancelled by the boundary chiral multiplets. This coincides with the boundary mixed 't Hooft anomalies for the exceptional Dirichlet boundary conditions $D_\alpha^r$, or equivalently the mixed supersymmetric Chern-Simons terms in a supersymmetric massive vacuum $\alpha$.

### 6.2.3 Higgs Branch Support

We now determine the Higgs branch support of the enriched Neumann boundary conditions $N_\alpha^r$. To contrast with exceptional Dirichlet, it is convenient to first consider $\zeta_\mathbb{C} \neq 0$, where the Higgs branch is a complex deformation of $T^*\mathbb{CP}^{N-1}$. We then set $\zeta_\mathbb{C} = 0$ to recover $X = T^*\mathbb{CP}^{N-1}$.

First, the boundary superpotential requires

$$X_\alpha| = -\frac{\partial W}{\partial Y_\alpha}| = e^{\phi_\alpha}, \qquad \frac{\partial W}{\partial\phi_\alpha} = Y_\alpha|e^{\phi_\alpha} + \zeta_\mathbb{C} = 0. \tag{234}$$

Combined with the boundary condition on the hypermultiplets (226), this implies the support of $N_\alpha^r$ is the quotient of the submanifold of $T^*\mathbb{C}^N$:

$$X_\beta = 0 \ (\beta < \alpha), \qquad X_\alpha Y_\alpha = \zeta_\mathbb{C}, \qquad Y_\beta = 0 \ (\beta > \alpha), \tag{235}$$

by $G = U(1)$.

For non-zero complex FI parameter $\zeta_\mathbb{C} \neq 0$, this is the attracting submanifold of the vacuum $\alpha$ in the complex deformation of the Higgs branch and coincides with the support of

exceptional Dirichlet $D_\alpha^r$. However, sending $\zeta_\mathbb{C}$ to zero, for enriched Neumann we now have $X_\alpha = 0$ and/or $Y_\alpha = 0$ and the support becomes

$$N^\perp U_\alpha \cup N^\perp U_{\alpha+1} = \overline{X_\alpha^-} \cup \overline{X_{\alpha+1}^-}, \tag{236}$$

where we have recycled the notation from section 5. This differs from the support of exceptional Dirichlet, which remains the attracting submanifold $X_\alpha^- \subset X$. The support of enriched Neumann and exceptional Dirichlet boundary conditions for supersymmetric QED with 3 hypermultiplets are contrasted in figure 16.

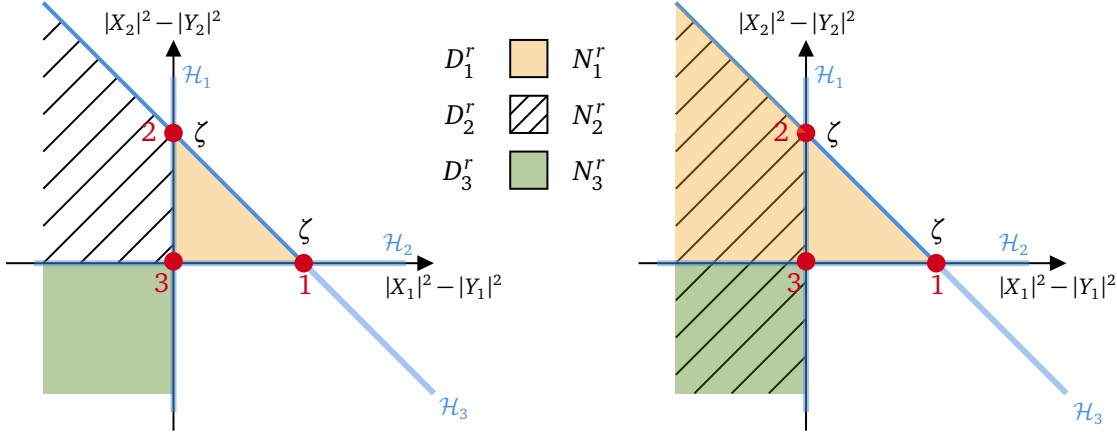

Figure 16: The support of right enriched Neumann boundary conditions (right) and exceptional Dirichlet (left) for supersymmetric QED with $N = 3$ hypermultiplets. See figure 12 for the interpretation of the hyper-toric diagram.

The support of enriched Neumann boundary conditions $N_\alpha^r$ coincides with the cohomological stable envelopes of $X = T^*\mathbb{CP}^{N-1}$ introduced in [13] in the default chamber. We will see shortly that the boundary amplitudes of enriched Neumann on $E_\tau$ reproduce the corresponding elliptic stable envelopes. A similar computation shows that the support of left enriched Neumann boundary conditions is the cohomological stable envelope in the opposite chamber for the mass parameters.

Let us summarise this picture more broadly for exceptional Dirichlet $D_\alpha^r$ and enriched Neumann $N_\alpha^r$ boundary conditions. Although the names of these boundary conditions refer to explicit UV constructions in abelian theories, we expect the same considerations to apply to analogous distinguished sets of boundary conditions labelled by vacua $\alpha$ in theories satisfying our assumptions in section 2.

- Exceptional Dirichlet $D_\alpha^r$

    - $\zeta_\mathbb{C}$ generic: supported on the attracting submanifold $X_{\alpha,\zeta_\mathbb{C}}^- \subset X_{\zeta_\mathbb{C}}$.

    - $\zeta_\mathbb{C} = 0$: supported on the attracting submanifold $X_\alpha^- \subset X$.

    - While $X_{\alpha,\zeta_\mathbb{C}}^- \subset X_{\zeta_\mathbb{C}}$ is closed for generic $\zeta_\mathbb{C}$, $X_\alpha^- \subset X$ is generally not closed at $\zeta_\mathbb{C} = 0$. The closure $\overline{X_\alpha^-}$ is not generally stable under perturbations away from $\zeta_\mathbb{C} = 0$. In other words, the Higgs branch support jumps upon turning on a complex FI parameter.

- Enriched Neumann $N_\alpha^r$

    - $\zeta_\mathbb{C}$ generic: supported on the attracting set $X_{\alpha,\zeta_\mathbb{C}}^- \subset X_{\zeta_\mathbb{C}}$.

– $\zeta_{\mathbb{C}} = 0$: supported on the stable envelope $\mathrm{Stab}(\alpha) \subset X$, which contains $X_{\alpha}^{-}$ and generically a union of some $X_{\beta}^{-}$ where $h_m|_{\beta} > h_m|_{\alpha}$.

– The support is given by closing $X_{\alpha,\zeta_{\mathbb{C}}}^{-} \subset X_{\zeta_{\mathbb{C}}}$ in the whole family of complex deformations labelled by $\zeta_{\mathbb{C}} \in H^2(X, \mathbb{C})$, including $\zeta_{\mathbb{C}} = 0$. In other words, the Higgs branch is smoothly deformed upon turning on a complex FI parameter.

It is straightforward to extend our derivation to general abelian theories, whose Higgs branches are hyper-toric varieties. For a vacuum $\alpha$, denote by $S \subset \{1, \dots, N\}$ the subset of size $r$ corresponding to hypermultiplets $(X_i, Y_i)$ of zero real mass in $\alpha$. Then we may define the polarisation

$$(\bar{\varepsilon}_{\alpha}^{r})_j = \begin{cases} -(\varepsilon_{\alpha}^{r})_j & j \in S, \\ (\varepsilon_{\alpha}^{r})_j & j \notin S. \end{cases} \tag{237}$$

In [15] it was shown that for $\zeta_{\mathbb{C}} = 0$, the Higgs branch image of $D_{\alpha}^{r}$ is the toric variety associated to the chamber $\Delta_{S, \bar{\varepsilon}_{\alpha}^{r}}$ of the hyper-toric diagram of $X$, using notation therein. Repeating our analysis for a general abelian theory, we find that the image of enriched Neumann boundary conditions $\mathcal{N}_{\alpha}^{r}$, for $\zeta_{\mathbb{C}} = 0$, are toric varieties associated to the orthants $V_{S, \varepsilon_{\alpha}^{r}}$, obeying

$$V_{S, \varepsilon_{\alpha}^{r}} = \bigcup_{\substack{\varepsilon \mid \varepsilon_j = (\varepsilon_{\alpha}^{r})_j \\ \forall j \notin S}} \Delta_{\varepsilon} \supset \Delta_{S, \bar{\varepsilon}_{\alpha}^{r}}. \tag{238}$$

These are precisely the cohomological stable envelopes of hyper-toric varieties [67].

We note that the cohomological stable envelopes appeared as an orthonormal basis of boundary conditions of $\mathcal{N} = 4$ quantum mechanics in a work by one of the authors [71], containing 1d analogues of some of the results we discuss in the remainder of this paper.

Finally, we mention that the difference between exceptional Dirichlet and enriched Neumann boundary conditions is also detected by the general half index [16–18] counting quarter-BPS boundary local operators. We will explore this topic, connecting with the mathematical literature on vertex functions [72] and exploring their mirror symmetry properties [73–75] in a future work [21].

### 6.2.4 Enriched Neumann for the Dual

We can also employ these techniques to construct enriched Neumann boundary conditions for $\widetilde{\mathcal{T}}$, which will realise elliptic stable envelopes of the resolution of the $A_{N-1}$-type singularity. We will construct the left enriched Neumann boundary conditions $\widetilde{N}_{\alpha}^{l}$ by acting with the same mirror symmetry interface on the left exceptional Dirichlet boundary conditions $D_{\alpha}^{l}$ given in (170).

To do so, we collide with a duality interface specified by the (twisted) superpotential

$$\begin{aligned} W &= \sum_{\beta < \alpha} \left( Y_{\beta} | e^{-\phi_{\beta}} - \phi_{\beta} | \widetilde{M}_{\mathbb{C}}^{\beta} \right) + \sum_{\beta \geq \alpha} \left( X_{\beta} | e^{-\phi_{\beta}} - \phi_{\beta} | \widetilde{M}_{\mathbb{C}}^{\beta} \right), \\ \widetilde{W} &= \sum_{\beta < \alpha} \left( e^{-\widetilde{\phi}_{\beta}} | \widetilde{X}_{\beta} - M_{\mathbb{C}}^{\beta} | \widetilde{\phi}_{\beta} \right) + \sum_{\beta \geq \alpha} \left( e^{\widetilde{\phi}_{\beta}} | \widetilde{Y}_{\beta} - M_{\mathbb{C}}^{\beta} | \widetilde{\phi}_{\beta} \right). \end{aligned} \tag{239}$$

Colliding with $D_{\alpha}^{l}$, similarly to before, the $\alpha^{\text{th}}$ terms flip the boundary condition for the twisted hyper $(\widetilde{X}_{\alpha}, \widetilde{Y}_{\alpha})$ in $\widetilde{\mathcal{T}}$. In summary, the dual is found to be the left enriched Neumann boundary condition, which we denote by $\widetilde{N}_{\alpha}^{l}$, specified by the splitting:

$$\begin{aligned} D_{\perp} \widetilde{X}_{\beta} = 0, \quad \widetilde{Y}_{\beta} = 0, \quad \beta \leq \alpha, \\ D_{\perp} \widetilde{Y}_{\beta} = 0, \quad \widetilde{X}_{\beta} = 0, \quad \beta > \alpha, \end{aligned} \tag{240}$$

coupled to $N-1$ boundary $\mathbb{C}^*$-valued $\mathcal{N}=(2,2)$ twisted chirals $\widetilde{\Phi}_\beta$ for $\beta \neq \alpha$, and their $T$-duals, via (twisted) boundary superpotentials

$$
\begin{aligned}
W &= -\sum_{\beta<\alpha} \phi_\beta |(\widetilde{\varphi}_\beta - \widetilde{\varphi}_{\beta-1}) - \sum_{\beta>\alpha} \phi_\beta |(\widetilde{\varphi}_\beta - \widetilde{\varphi}_{\beta-1}), \\
\widetilde{W} &= \sum_{\beta<\alpha} \left( e^{-\widetilde{\phi}_\beta} |\widetilde{X}_\beta - (m_{\alpha,\mathbb{C}} - m_{\beta,\mathbb{C}}) \widetilde{\phi}_\beta \right) + \sum_{\beta>\alpha} \left( e^{\widetilde{\phi}_\beta} |\widetilde{Y}_\beta - (m_{\alpha,\mathbb{C}} - m_{\beta,\mathbb{C}}) \widetilde{\phi}_\beta \right),
\end{aligned}
\tag{241}
$$

where we have abused notation and let $\widetilde{\varphi}_0 = -\widetilde{\varphi}_N = \frac{1}{2}\zeta_\mathbb{C}$. As before, collision with $D_\alpha^l$ shifts the charges of operators at the boundary charged under the gauge symmetry of $\mathcal{T}$. These are just the scalars $\phi_\beta$ in the $\mathbb{C}^*$ valued chirals $\Phi_\beta$. Note from the perspective of $\widetilde{\mathcal{T}}$ the valid twist operators are $e^{\mathfrak{n}\phi_\beta}$ for $\beta < \alpha$ and $e^{-\mathfrak{n}\phi_\beta}$ for $\beta > \alpha$, where $\mathfrak{n} \in \mathbb{Z}_{\geq 0}$. The shift in charges can be encoded in the substitution $s v_\alpha^{-1} t = 1$ in the characters of these operators:

$$
\begin{aligned}
e^{\phi_\beta} &: \quad s v_\beta^{-1} t \to v_\alpha v_\beta^{-1} \qquad \text{for } \beta < \alpha, \\
e^{-\phi_\beta} &: \quad s^{-1} v_\beta t \to v_\alpha^{-1} v_\beta t^2 \quad \text{for } \beta > \alpha.
\end{aligned}
\tag{242}
$$

One can also determine the Higgs branch support of the above left enriched Neumann boundary conditions for $\widetilde{\mathcal{T}}$, following section 5 of [15]. The calculation is similar to that of $\mathcal{T}$. The result is that the Higgs branch image is given by the $\widetilde{G} = U(1)^{N-1}$ quotient of the following Lagrangian of the matter representation $\widetilde{Q}$

$$
\begin{aligned}
\widetilde{Y}_\alpha &= 0, \quad D_\perp \widetilde{X}_\alpha = 0, \\
\widetilde{X}_\beta \widetilde{Y}_\beta &= m_{\alpha,\mathbb{C}} - m_{\beta,\mathbb{C}} \quad \beta \neq \alpha.
\end{aligned}
\tag{243}
$$

In our case, we set the complex masses to zero, $m_\mathbb{C} = 0$.

It is then not hard to check that the Higgs branch support of $\widetilde{N}_\alpha^l$ in the $A_{N-1}$ surface matches the putative Coulomb branch images of $D_\alpha^l$ described in section 5.6. In particular, the slice $\mathcal{S}_0$ of the Coulomb branch of $\mathcal{T}$ is mapped by mirror duality to a slice $S^0 = \{\widetilde{X}_\gamma \widetilde{Y}_\gamma = 0 \quad \forall \gamma = 1, \dots, N\}$, and the axis labelled by the Coulomb branch moment map $\sigma$ is mapped to the dual Higgs branch moment map $\widetilde{\mu}_H = \frac{1}{2}(|\widetilde{X}_N|^2 - |\widetilde{Y}_N|^2 + |\widetilde{X}_1|^2 - |\widetilde{Y}_1|^2)$. This is illustrated in the figure 17 for the case with 3 hypermultiplets.

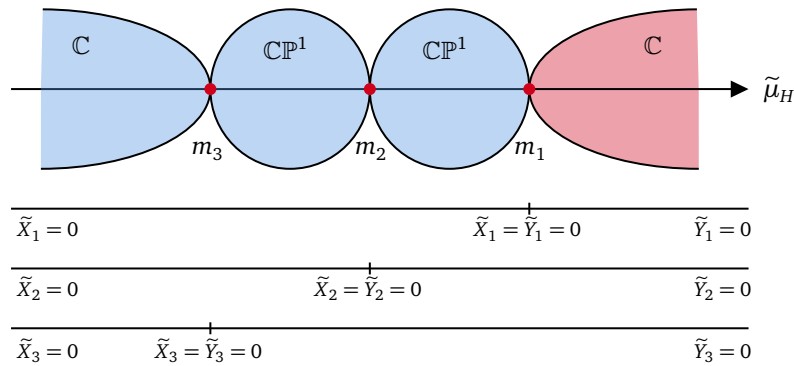

Figure 17: The slice $\mathcal{S}^0$ of the Higgs branch of $\widetilde{\mathcal{T}}$. The support of $\widetilde{N}_\alpha^r$ is the blue region, and $\widetilde{N}_\alpha^l$ the red, for $\alpha = 1$. The real moment map $h_{\widetilde{m}} = \widetilde{m} \cdot \widetilde{\mu}_H = -\zeta \cdot \widetilde{\mu}_H$ on the Higgs branch of $\widetilde{\mathcal{T}}$ decreases from left to right in our choice of $\widetilde{\mathfrak{C}}_H = \mathfrak{C}_C$.

## 6.3 Amplitudes and Wavefunctions

We now compute the boundary amplitudes and wavefunctions of the mirror symmetry interface and enriched Neumann boundary conditions. We will encounter expressions for elliptic stable envelopes, which will be expanded on in section 7.

### 6.3.1 Mirror Symmetry Interface

We first consider the wavefunction of the mirror symmetry interface, obtained by forming a sandwich with auxiliary Dirichlet boundary conditions. This provides an integration kernel that may be used to compute the action of mirror symmetry on the wavefunctions of other boundary conditions. We will identify this kernel with the mother function in equivariant elliptic cohomology.

Let us then consider the partition function on $E_\tau \times [-\ell, \ell]$, with the mirror interface at $x^3 = 0$, a left reference Dirichlet boundary condition for $\mathcal{T}$ at $x^3 = -\ell$, and a right reference Dirichlet boundary condition for $\widetilde{\mathcal{T}}$ at $x^3 = \ell$. This is illustrated in figure 18.

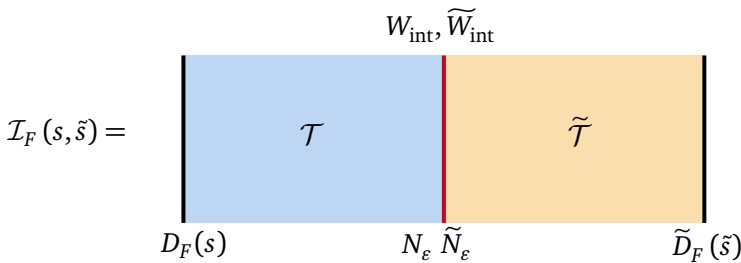

Figure 18: The path integral on $E_\tau \times [-\ell, \ell]$ which yields the mother function.

There is a choice of $\mathcal{N} = (2,2)$ or $\mathcal{N} = (0,2)$ reference Dirichlet boundary conditions, which will lead to different normalisations of the mirror interface kernel function. To ensure the set-up is compatible with real masses in any chamber and also to match results in the mathematical literature [8, 76], we will choose the $\mathcal{N} = (0,2)$ boundary conditions:

$$
\begin{aligned}
D_F(s) : \quad & |\Phi_X = |\Phi_Y = 0, \\
\widetilde{D}_F(\tilde{s}) : \quad & \Phi_{\widetilde{X}}| = \Phi_{\widetilde{Y}}| = 0.
\end{aligned}
\tag{244}
$$

Here $s_a$, $a = 1, \ldots, r$ are holonomies around $E_\tau$ for $\mathcal{T}$, and $\tilde{s}_{a'}$, $a' = 1, \ldots, N-r$ for $\widetilde{\mathcal{T}}$. Recall that these boundary conditions also impose a Dirichlet boundary condition for the 3d $\mathcal{N} = 2$ vector multiplet, where the gauge connections at the boundary are set equal to $s_a$ and $\tilde{s}_{a'}$ respectively, and a Neumann boundary condition for the adjoint chirals $\varphi$, $\widetilde{\varphi}$.

We denote the kernel function with this choice of auxiliary Dirichlet boundary conditions as

$$
\mathcal{I}_F(s, \tilde{s}) = \langle D_F(s) | \mathcal{I} | \widetilde{D}_F(\tilde{s}) \rangle.
\tag{245}
$$

The kernel functions for other choices of reference Dirichlet boundary conditions are related by overall normalisations as in section 4. This kernel function may be computed explicitly for abelian gauge theories and coincides with the mother function [8, 76] in equivariant elliptic cohomology for hyper-toric varieties.

Here we focus on the wavefunction when $\mathcal{T}$ is supersymmetric QED with $N$ flavours and $\widetilde{\mathcal{T}}$ is mirror the abelian $A_{N-1}$-type quiver gauge theory, the extension to the abelian case is straightforward. This will receive contributions from the following sources:

- Colliding the left Dirichlet boundary condition $D_F(s)$ with the right Neumann boundary condition $N_\varepsilon$, there remains Fermi multiplets $\Psi_{Y_\varepsilon}$. Similarly, colliding the right Dirichlet boundary condition $\widetilde{D}_F(\tilde{s})$ with the left Neumann boundary condition $\widetilde{N}_{-\varepsilon}$ leaves behind Fermi multiplets $\Psi_{\widetilde{Y}_{-\varepsilon}}$.

- The remaining fluctuating degrees of freedom from colliding the boundary conditions for the vector multiplets are adjoint chiral multiplets $\Phi_\varphi$, $\Phi_{\widetilde{\varphi}}$, with scalar components $\varphi$ and $\widetilde{\varphi}$.

- The $\mathbb{C}^*$-valued $\mathcal{N} = (2,2)$ boundary chiral multiplets $\Phi_\beta$.

Combining the elliptic genera of these contributions will yield the kernel function of the mirror symmetry interface.

We enumerate the elliptic genera of these contributions in turn. First, the contributions from the $\mathcal{N} = (0,2)$ Fermi multiplets arising from bulk hypermultiplets are

$$\Psi_{Y_\varepsilon} : \quad \prod_{\beta=1}^{N} \vartheta(t^{-1}(sv_\beta^{-1})^{\varepsilon_\beta}), \qquad \Psi_{\widetilde{Y}_{-\varepsilon}} : \quad \prod_{\beta=1}^{N} \vartheta(t(\tilde{s}_\beta/\tilde{s}_{\beta-1})^{-\varepsilon_\beta}). \tag{246}$$

The contributions from the adjoint $(0,2)$ chirals are

$$\Phi_\varphi : \quad \vartheta(t^{-2})^{-1}, \qquad \Phi_{\widetilde{\varphi}} : \quad \vartheta(t^2)^{-(N-1)}. \tag{247}$$

The contribution from $\mathbb{C}^*$-valued chiral multiplets $\{\Phi_\beta\}_{\beta=1,\ldots,N}$ is more difficult to compute. The description of the duality interface is slightly non-Lagrangian as it involves boundary superpotentials and twisted superpotentials coupling to $\Phi_\beta$ and its T-dual $\widetilde{\Phi}_\beta$ simultaneously. However, since these couplings are exact for the elliptic genus, it is reasonable to treat the contributions as isolated 2d $\mathbb{C}^*$-valued chiral multiplets.

It is unclear how to compute the R-R sector elliptic genus of these chiral multiplets using supersymmetric localisation. Instead, we first compute their elliptic genus in the NS-NS sector by employing an operator counting argument, before performing a spectral flow back to the R-R sector. This computation is performed in appendix B. The result is

$$\Phi, \widetilde{\Phi} \quad : \quad \prod_{\beta=1}^{N} \frac{\vartheta(t^2)\vartheta((sv_\beta^{-1})^{-\varepsilon_\beta}(\tilde{s}_\beta/\tilde{s}_{\beta-1})^{\varepsilon_\beta})}{\vartheta(t(sv_\beta^{-1})^{-\varepsilon_\beta})\vartheta(t^{-1}(\tilde{s}_\beta/\tilde{s}_{\beta-1})^{\varepsilon_\beta})}. \tag{248}$$

We note that the arguments of the theta-functions in the denominators correspond to the weights of the boundary operators $e^{-\varepsilon_\beta \phi_\beta}$ and $e^{\varepsilon_\beta \widetilde{\phi}_\beta}$.

Combining the three contributions (246), (247) and (248), yields the result

$$\mathcal{I}_F(s, \tilde{s}) = -\prod_{\beta=1}^{N} \vartheta\left((sv_\beta^{-1})^{-\varepsilon_\beta}(\tilde{s}_\beta/\tilde{s}_{\beta-1})^{\varepsilon_\beta}\right). \tag{249}$$

We emphasise that different choices of polarisation $\varepsilon$ in the definition of the mirror interface yield the same mirror interface kernel up to a sign, when sandwiched between the $\mathcal{N} = (0,2)$ reference Dirichlet boundary conditions. Up to an overall sign and a re-definition of gauge and R-symmetries, this is the mother function for $T^*\mathbb{P}^{N-1}$ [8, 76].

Had we sandwiched with reference $\mathcal{N} = (2,2)$ Dirichlet boundary conditions $D_{-\varepsilon}(s)$ and $\widetilde{D}_\varepsilon(\tilde{s})$, upon shrinking the interval the remaining fluctuating degrees of freedom are solely the $\mathbb{C}^*$-valued chirals, and the interface kernel would just yield (248).

We now explain why, from our physical perspective, it is natural to identify the duality interface as generating an element of the $T_f$ equivariant elliptic cohomology of $X \times X^!$, i.e. a holomorphic line bundle over the variety $\mathrm{Ell}_{T_f}(X \times X^!)$.

One may view the partition function of the interface as that of a doubled theory $\mathcal{T} \times \widetilde{\mathcal{T}}$ on $E_\tau \times [0, \ell]$, by folding together the two theories $\mathcal{T}$ and $\widetilde{\mathcal{T}}$. The doubled theory consists of two sectors which are decoupled in the bulk, but are coupled at the duality interface which becomes a boundary condition.

The Higgs branch of $\mathcal{T} \times \widetilde{\mathcal{T}}$ is $X \times X^!$, with an equivariant action of $T_f = T_C \times T_H \times T_t$. For the purposes of this discussion, we denote the image of the massive vacua of $\mathcal{T}$ on its Higgs branch $X$ by $\alpha$, and its Coulomb branch $X^!$ by $\tilde{\alpha}$. Then $\mathcal{T} \times \widetilde{\mathcal{T}}$ has $N^2$ vacua given by a choice

of $(\alpha, \tilde{\alpha})$. Following section 3, the equivariant elliptic cohomology variety of $X \times X^!$, or spectral curve for the space of supersymmetric ground states of $\mathcal{T} \times \widetilde{\mathcal{T}}$, is:

$$\mathrm{Ell}_{T_f}(X \times X^!) := \left( \bigsqcup_{(\alpha, \tilde{\alpha})} E_{T_f}^{(\alpha, \tilde{\alpha})} \right) \Big/ (\Delta \times \widetilde{\Delta}). \tag{250}$$

In the above

- $E_{T_f}^{(\alpha, \tilde{\alpha})} \cong \Gamma_f \otimes_{\mathbb{Z}} E_\tau$ are $N^2$ copies of the torus of background flat connections for $T_f$, associated to the supersymmetric vacua $(\alpha, \tilde{\alpha})$.

- $\Delta \times \widetilde{\Delta}$ identifies:

    - The copies $E_{T_f}^{(\alpha, \tilde{\gamma})}$ and $E_{T_f}^{(\beta, \tilde{\gamma})}$ for all $\tilde{\gamma}$, at points $\lambda \cdot z \in \mathbb{Z} + \tau\mathbb{Z}$, where $\lambda \in \Phi_\alpha \cap (-\Phi_\beta)$ labels an internal edge of the GKM diagram of $X$.

    - The copies $E_{T_f}^{(\gamma, \tilde{\alpha})}$ and $E_{T_f}^{(\gamma, \tilde{\beta})}$ for all $\gamma$, at points $\tilde{\lambda} \cdot \tilde{z} \in \mathbb{Z} + \tau\mathbb{Z}$ where $\tilde{\lambda} \in \widetilde{\Phi}_{\tilde{\alpha}} \cap (-\widetilde{\Phi}_{\tilde{\beta}})$ labels an internal edge of the GKM diagram of $X^!$. Here $\tilde{z} = (z_C, z_t)$.

As a boundary condition for $\mathcal{T} \times \widetilde{\mathcal{T}}$, the interface naturally defines a $Q$-cohomology class, or supersymmetric ground state. The mother function $\mathcal{I}_F(s, \tilde{s})$ is now interpreted as its wavefunction, with reference boundary condition $D_F(s) \times \widetilde{D}_F(\tilde{s})$. See figure 19.

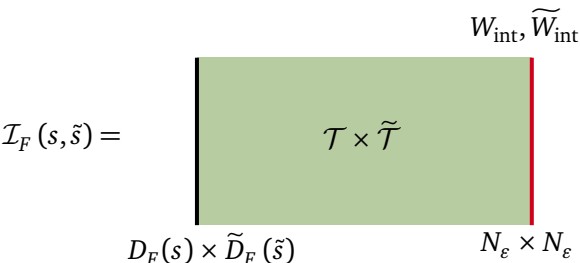

Figure 19: The wavefunction of the duality interface.

By evaluating $\mathcal{I}_F(s, \tilde{s})$ at the values of $s$ and $\tilde{s}$ specified by $(\alpha, \tilde{\alpha})$, one recovers an $N \times N$ matrix of boundary amplitudes, glued along its rows according to the GKM description of section 3 for $\mathcal{T}$, and its columns according to that of $\widetilde{\mathcal{T}}$. These form a section of a holomorphic line bundle over $\mathrm{Ell}_{T_f}(X \times X^!)$. We do not reproduce the amplitudes explicitly here in the interest of brevity, since we shall see shortly that we effectively compute them as the boundary amplitudes of enriched Neumann boundary conditions.

### 6.3.2 Enriched Neumann

We will first compute the wavefunctions of enriched Neumann boundary conditions from first principles. We then show that colliding the mirror symmetry interface with exceptional Dirichlet boundary conditions produces the same wavefunctions using the kernel function.

We first consider the wavefunction of right boundary enriched Neumann,

$$\left\langle D_F(s) | N_\alpha^r \big| D_F(s) | N_\alpha^r \right\rangle. \tag{251}$$

After taking the interval length $\ell \to 0$, the remaining fluctuating degrees of freedom are the chiral multiplet $\Phi_\varphi$, Fermi multiplets $\Psi_{X_\gamma}$ for $\gamma \leq \alpha$, $\Psi_{Y_\gamma}$ for $\gamma > \alpha$, and the boundary $\mathbb{C}^*$ chiral

multiplet $\Phi_\alpha$. The contribution of the $\mathbb{C}^*$ chiral multiplet[12] is computed using the same method as for the mother function and gives the contribution (248) for $\gamma = \alpha$, except with $\widetilde{s} = \widetilde{s}|_\alpha$. We return to this observation in the next section. Combining these contributions, we find

$$
\begin{aligned}
\left\langle D_F(s)|N_\alpha^r\big|D_F(s)|N_\alpha^r\right\rangle &= \frac{1}{\vartheta(t^{-2})}\prod_{\gamma\leq\alpha}\vartheta(t^{-1}s^{-1}v_\gamma)\frac{\vartheta(t^2)\vartheta(sv_\alpha^{-1}t\,\xi t^{N-2\alpha})}{\vartheta(sv_\alpha^{-1}t)\vartheta(\xi t^{N-2\alpha})}\prod_{\gamma>\alpha}\vartheta(t^{-1}sv_\gamma^{-1}), \\
&= \prod_{\gamma<\alpha}\vartheta(t^{-1}s^{-1}v_\gamma)\frac{\vartheta(sv_\alpha^{-1}t\,\xi t^{N-2\alpha})}{\vartheta(\xi t^{N-2\alpha})}\prod_{\gamma>\alpha}\vartheta(t^{-1}sv_\gamma^{-1}).
\end{aligned}
\tag{252}
$$

This expression matches the elliptic stable envelope for $X = T^*\mathbb{P}^{N-1}$ [5],

$$
\left\langle D_F(s)|N_\alpha^r\big|D_F(s)|N_\alpha^r\right\rangle = \mathrm{Stab}(\alpha)_{\mathfrak{C}_H,\xi}.
\tag{253}
$$

Recall that the boundary amplitudes at the origin of mass parameter space are constructed using $(0,2)$ exceptional Dirichlet, and are, using any of the equivalent results of sections 4 and 5:

$$
\begin{aligned}
\left\langle\beta|N_\alpha^r\big|\beta|N_\alpha^r\right\rangle &= \left\langle D_F(v_\beta t^{-1})|N_\alpha^r\big|D_F(v_\beta t^{-1})|N_\alpha^r\right\rangle \\
&= \prod_{\gamma<\alpha}\vartheta(v_\gamma v_\beta^{-1})\frac{\vartheta(v_\beta v_\alpha^{-1}\xi t^{N-2\alpha})}{\vartheta(\xi t^{N-2\alpha})}\prod_{\gamma>\alpha}\vartheta(t^{-2}v_\beta v_\gamma^{-1}).
\end{aligned}
$$

The collection $\{\left\langle\beta|N_\alpha^r\big|\beta|N_\alpha^r\right\rangle\}_{\beta=1,\dots,N}$ glues to give a section of a line bundle over $E_T(X) = \mathrm{Ell}_T(X)\times E_{T_C}$. This is consistent with our description of supersymmetric ground states in section 3. Since $\vartheta(1) = 0$, this matrix of boundary amplitudes is lower-triangular: it vanishes for $\beta < \alpha$. This reflects that the Higgs branch support of the enriched Neumann boundary condition $N_\alpha^r$ contains only the fixed points $\beta \geq \alpha$.

The boundary amplitudes constructed using $\mathcal{N} = (2,2)$ exceptional Dirichlet boundary conditions $D_\alpha^l$ are simply related to the $(0,2)$ boundary amplitudes by a normalisation:

$$
\begin{aligned}
\left\langle D_\beta^l|N_\alpha^r\big|D_\beta^l|N_\alpha^r\right\rangle &= \left\langle D_F(s_\beta)|N_\alpha^r\big|D_F(s_\beta)|N_\alpha^r\right\rangle\prod_{\lambda\in\Phi_\beta^+}\frac{1}{\vartheta(a^\lambda)} \\
&= \begin{cases}
\dfrac{\vartheta(t^2)\vartheta(\frac{v_\beta}{v_\alpha}\xi t^{N-2\alpha})}{\vartheta(\frac{v_\beta}{v_\alpha})\vartheta(\xi t^{N-2\alpha})}\displaystyle\prod_{\alpha<\gamma<\beta}\frac{\vartheta(t^{-2}\frac{v_\beta}{v_\gamma})}{\vartheta(\frac{v_\gamma}{v_\beta})} & \text{if } \beta>\alpha, \\
1 & \text{if } \beta=\alpha, \\
0 & \text{if } \beta<\alpha.
\end{cases}
\end{aligned}
\tag{254}
$$

The quasi-periodicities of the non-vanishing overlaps are consistent with the anomalies of the boundary conditions $D_\beta^l$, $N_\alpha^r$, and are determined by the difference of Chern-Simons levels $K_\alpha - K_\beta$. This matrix of boundary amplitudes is nothing but the pole-subtraction matrix introduced in [5]. The reason for this identification is explained in [21].

One can similarly compute the wavefunction of enriched Neumann for the mirror $\widetilde{\mathcal{T}}$. We consider the wavefunction of left enriched Neumann boundary condition,

$$
\left\langle\widetilde{N}_\alpha^l|\widetilde{D}_F(\widetilde{s})\big|\widetilde{N}_\alpha^l|\widetilde{D}_F(\widetilde{s})\right\rangle.
\tag{255}
$$

---

[12]Note that similar ratios of theta functions appeared in [77] in the IR treatment of holomorphic blocks: partition functions on $S^1\times HS^2$. There they arise as replacements of contributions of mixed Chern-Simons levels, motivated by quasi-periodicity and meromorphicity arguments. They occur naturally in our UV perspective as contributions of $\mathbb{C}^*$-valued matter on the boundary torus $E_\tau = \partial(S^1\times HS^2)$. We return to this observation in a future work [21]

After taking interval length $\ell \to 0$, there are the following contributions from Fermi and chiral multiplets arising from the boundary conditions for bulk supermultiplets,

$$
\begin{aligned}
\Psi_{\widetilde{Y}_\gamma} &: \vartheta(t\tilde{s}_\gamma/\tilde{s}_{\gamma-1}) \qquad \gamma \leq \alpha, \\
\Psi_{\widetilde{X}_\gamma} &: \vartheta(t\tilde{s}_{\gamma-1}/\tilde{s}_\gamma) \qquad \gamma > \alpha, \\
\Phi_{\tilde{\varphi}} &: \vartheta(t^2)^{-(N-1)}.
\end{aligned}
\tag{256}
$$

The contribution of the $N-1$ $\mathbb{C}^*$-valued twisted chiral multiplets is

$$
\prod_{\gamma<\alpha} \frac{\vartheta(t^2)\vartheta(t^{-1}\frac{\tilde{s}_{\gamma-1}}{\tilde{s}_\gamma}\frac{v_\alpha}{v_\gamma})}{\vartheta(t^{-1}\frac{\tilde{s}_{\gamma-1}}{\tilde{s}_\gamma})\vartheta(\frac{v_\alpha}{v_\gamma})} \prod_{\gamma>\alpha} \frac{\vartheta(t^2)\vartheta(t^{-1}\frac{\tilde{s}_\gamma}{\tilde{s}_{\gamma-1}}t^2\frac{v_\gamma}{v_\alpha})}{\vartheta(t^{-1}\frac{\tilde{s}_\gamma}{\tilde{s}_{\gamma-1}})\vartheta(t^2\frac{v_\gamma}{v_\alpha})}.
\tag{257}
$$

Putting these contributions together

$$
\langle \widetilde{N}_\alpha^l | \widetilde{D}_F(\tilde{s}) | \widetilde{N}_\alpha^l | \widetilde{D}_F(\tilde{s}) \rangle = \prod_{\gamma<\alpha} \frac{\vartheta(t^{-1}\frac{\tilde{s}_{\gamma-1}}{\tilde{s}_\gamma}\frac{v_\alpha}{v_\gamma})}{\vartheta(\frac{v_\gamma}{v_\alpha})} \vartheta\left(t\frac{\tilde{s}_\alpha}{\tilde{s}_{\alpha-1}}\right) \prod_{\gamma>\alpha} \frac{\vartheta(t\frac{\tilde{s}_\gamma}{\tilde{s}_{\gamma-1}}\frac{v_\gamma}{v_\alpha})}{\vartheta(t^{-2}\frac{v_\alpha}{v_\gamma})},
\tag{258}
$$

which, recalling the identifications $(m,\zeta) = (\tilde{\zeta}, -\tilde{m})$, may be identified as the elliptic stable envelope for the $A_{N-1}$ surface [5, 8]

$$
\langle \widetilde{N}_\alpha^l | \widetilde{D}_F(\tilde{s}) | \widetilde{N}_\alpha^l | \widetilde{D}_F(\tilde{s}) \rangle = \widetilde{\text{Stab}}(\alpha)_{\widetilde{\mathfrak{C}}_H^{opp}, \tilde{\xi}^{-1}}
\tag{259}
$$

in the chambers

$$
\widetilde{\mathfrak{C}}_H = \{\tilde{m} < 0\}, \qquad \widetilde{\mathfrak{C}}_C = \{\tilde{\zeta}_1 > \ldots > \tilde{\zeta}_N\}.
\tag{260}
$$

The inverted $\tilde{\xi}$ and flipped mass chamber in (259) are consistent with the fact we are considering wavefunctions of left boundary conditions.

The boundary amplitudes of $\widetilde{N}_\alpha^l$ are given by evaluating (258) at (218), the value of the boundary $\widetilde{G}$ fugacities $\tilde{s}$ in the vacuum $\beta$ of $\widetilde{T}$ fixed by the expectation values (221):

$$
\langle \widetilde{N}_\alpha^l | \widetilde{D}_\beta | \widetilde{N}_\alpha^l | \widetilde{D}_\beta \rangle = \begin{cases} 0 & \text{if } \beta > \alpha, \\ (-1)^{N-1}\vartheta(\xi^{-1}t^{-N+2\alpha}) & \text{if } \beta = \alpha, \\ (-1)^{N-\alpha+\beta-1}\vartheta(t^2)\frac{\vartheta(\frac{v_\beta}{v_\alpha}\xi^{-1}t^{-N+2\beta})}{\vartheta(\frac{v_\alpha}{v_\beta})} \prod_{\beta<\gamma<\alpha} \frac{\vartheta(t^{-2}\frac{v_\alpha}{v_\gamma})}{\vartheta(\frac{v_\gamma}{v_\alpha})} & \text{if } \beta < \alpha. \end{cases}
\tag{261}
$$

This is upper-triangular, reflecting the fact that $\widetilde{\mathcal{N}}_\alpha^l$ contains only the fixed points $\beta \leq \alpha$, see figure 17. These amplitudes glue to a section of line bundle over $\text{Ell}_{\widetilde{T}}(X^!) \times E_{T_H} \equiv E_{\widetilde{T}}(X^!)$, where we have defined $\widetilde{T} = T_C \times T_t$. This is the extended elliptic cohomology of $X^!$, or equivalently the spectral curve for $\widetilde{\mathcal{T}}$.

### 6.3.3 Elliptic Stable Envelopes from Duality Interface

We now derive the same wavefunctions and boundary amplitudes for enriched Neumann boundary conditions by applying the mirror symmetry interface to exceptional Dirichlet. We shall see this has a well-defined mathematical origin in elliptic cohomology.

Let us first recover the wavefunction of the enriched Neumann boundary condition $N_\alpha^r$ by colliding the interface on the right with $\widetilde{D}_\alpha^r$. The relevant set-up is illustrated in figure 20.

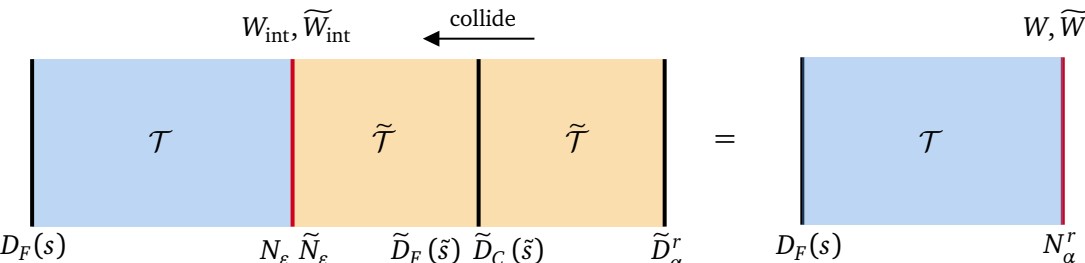

Figure 20: Collision with duality interface = Restriction of mother function

To evaluate the wavefunction, we cut the path integral using auxiliary Dirichlet boundary conditions in the mirror theory. We choose the polarisation $\varepsilon = \{+ \ldots + - \ldots -\}$ for the mirror interface, where the first minus sign is at position $\alpha$, as in equation (223). This gives a decomposition of the wavefunction of enriched Neumann as follows,

$$\left\langle D_F(s) | N_\alpha^r \middle| D_F(s) | N_\alpha^r \right\rangle = \oint \frac{d\tilde{s}}{2\pi i \tilde{s}} \left( \eta(q)^2 \vartheta(t^{-2}) \right)^{N-1} \mathcal{I}_F(s, \tilde{s}) \left\langle \widetilde{D}_F(\tilde{s}) | \widetilde{D}_\alpha^r \middle| \widetilde{D}_F(\tilde{s}) | \widetilde{D}_\alpha^r \right\rangle, \tag{262}$$

where

$$\mathcal{I}_F(s, \tilde{s}) = (-)^\alpha \prod_{\beta=1}^{N} \vartheta\left( s v_\beta^{-1} \frac{\tilde{s}_{\beta-1}}{\tilde{s}_\beta} \right), \tag{263}$$

and the minus sign comes from the choice of polarisation. The wavefunction of exceptional Dirichlet in the mirror is

$$\begin{aligned}
\left\langle \widetilde{D}_F(\tilde{s}) | \widetilde{D}_\alpha^r \middle| \widetilde{D}_F(\tilde{s}) | \widetilde{D}_\alpha^r \right\rangle &= \frac{\delta^{(N-1)}(\tilde{u} - \tilde{u}|_\alpha)}{(\eta(q)^2 \theta(t^{-2}))^{N-1} \prod_{\lambda \in \tilde{\Phi}_\alpha^-} \vartheta(\tilde{a}^\lambda)} \\
&= \frac{\delta^{(N-1)}(\tilde{u} - \tilde{u}|_\alpha)}{(\eta(q)^2 \theta(t^{-2}))^{N-1} \vartheta(\xi^{-1} t^{2\alpha-N})},
\end{aligned} \tag{264}$$

where we have used the tangent weights (220). The delta function is regarded as a contour prescription around a rank $N-1$ pole at (218) which is the value in the vacuum $\alpha$ in the mirror theory. Evaluating the residue reproduces the wavefunction (252). A similar computation yields the wavefunction of left enriched Neumann boundary conditions in the mirror (258).

Let us now briefly review the constructions in the mathematical literature, see in particular section 6 of [8], of which this is the precise physical analog. For this discussion we deviate slightly in notation, and denote the image of the massive vacua of $\mathcal{T}$ on its Higgs branch $X$ by $\alpha$ and its Coulomb branch $X^!$ by $\tilde{\alpha}$. Regarding $X \times X^!$ as a $T_f = T_C \times T_H \times T_t$ variety, there are equivariant embeddings

$$X = X \times \{\tilde{\alpha}\} \xrightarrow{i_{\tilde{\alpha}}} X \times X^! \xleftarrow{i_\alpha} \{\alpha\} \times X^! = X^!. \tag{265}$$

Taking the $T_f$ equivariant elliptic cohomology

$$\mathrm{Ell}_{T_f}(X \times \{\tilde{\alpha}\}) = \mathrm{Ell}_T(X) \times \mathrm{Ell}_{T_C}(\{\tilde{\alpha}\}) = \mathrm{Ell}_T(X) \times E_{T_C} = E_T(X). \tag{266}$$

The latter is precisely what we identified as the extended elliptic cohomology in section 3. Similarly

$$\mathrm{Ell}_{T_f}(\{\alpha\} \times X^!) = \mathrm{Ell}_{\widetilde{T}}(X^!) \times E_{T_H} = E_{\widetilde{T}}(X^!), \tag{267}$$

where we have defined $\widetilde{T} = T_C \times T_t$, such that $\mathrm{Ell}_{\widetilde{T}}(X^!)$ is the extended elliptic cohomology of $X^!$, and the associated spectral curve for the space of supersymmetric ground states of $\widetilde{\mathcal{T}}$.

The functoriality of elliptic cohomology then gives maps

$$E_T(X) \xrightarrow{i_\alpha^*} \mathrm{Ell}_{T_f}(X \times X^!) \xleftarrow{i_\alpha^*} E_{\widetilde{T}}(X^!). \tag{268}$$

The main result of [8] is that there exists a holomorphic section $\mathfrak{m}$ of a particular line bundle on $\mathrm{Ell}_{T_f}(X \times X^!)$ which can be pulled back by the maps above to give[13]

$$\frac{(i_\alpha^*)^*(\mathfrak{m})}{\prod_{\lambda \in \widetilde{\Phi}_\alpha^-} \vartheta(e^{2\pi i \lambda \cdot \tilde{z}})} = \mathrm{Stab}(\alpha)_{\mathfrak{C}_H, \xi}, \qquad \frac{(i_\alpha^*)^*(\mathfrak{m})}{\prod_{\lambda \in \Phi_\alpha^+} \vartheta(e^{2\pi i \lambda \cdot z})} = \widetilde{\mathrm{Stab}}(\alpha)_{\widetilde{\mathfrak{C}}_H^{opp}, \widetilde{\xi}_{-1}}, \tag{269}$$

which are sections of line bundles on $E_T(X)$ and $E_{\widetilde{T}}(X^!)$ respectively.

Note that more precisely, in the above we have used the 'wavefunction' representation of the sections/classes as described in section 4.5.1. Evaluation of $s, \tilde{s}$ and $(s, \tilde{s})$ at fixed points will give collections of amplitudes which glue to sections of line bundles over $E_T(X), E_{\widetilde{T}}(X^!)$ and $\mathrm{Ell}_{T_f}(X \times X^!)$ as appropriate.

The physical correspondence is now as follows. The existence of the mother function $\mathfrak{m}$ is equivalent to the existence of a duality interface between theories $\mathcal{T}$ and $\widetilde{\mathcal{T}}$. The mother function $\mathfrak{m}$ is itself the wavefunction $\mathcal{I}_F(s, \tilde{s})$, or the mirror duality interface kernel, described previously. The pullbacks, or fixed point evaluations (269) are simply the collision of the duality interface with exceptional Dirichlet boundary conditions.

## 6.4 Orthogonality and Duality of Stable Envelopes

We demonstrated in section 5 that left and right $\mathcal{N} = (2, 2)$ exceptional Dirichlet boundary conditions are orthonormal, as expected as they represent normalisable supersymmetric ground states at $x \to \pm\infty$. Since enriched Neumann boundary conditions are mirror to exceptional Dirichlet, we also expect that they are orthonormal,

$$\left\langle N_\alpha^l | N_\beta^r \middle| N_\alpha^l | N_\beta^r \right\rangle = \delta_{\alpha\beta}. \tag{270}$$

Let us demonstrate this in supersymmetric QED.

First we write down the wavefunction of left enriched Neumann, using the same reasoning as before. The overlap with auxiliary Dirichlet boundary conditions receives contributions from the Fermi multiplets $\Psi_{Y_\beta}$ for $\beta < \alpha$ and $\Psi_{X_\beta}$ for $\beta \geq \alpha$, the chiral $\Phi_\varphi$, as well as the $\mathbb{C}^*$ boundary chiral $\Phi_\alpha$. The result is

$$\left\langle N_\alpha^l | D_F(s) \middle| N_\alpha^l | D_F(s) \right\rangle = \prod_{\beta < \alpha} \vartheta(t^{-1} s v_\beta^{-1}) \frac{\vartheta(s v_\alpha^{-1} t \xi^{-1} t^{-N-2+2\alpha})}{\vartheta(\xi^{-1} t^{-N-2+2\alpha})} \prod_{\beta > \alpha} \vartheta(t^{-1} s^{-1} v_\beta). \tag{271}$$

We note the relation

$$\begin{aligned}
\left\langle N_\alpha^l | D_F(s) \middle| N_\alpha^l | D_F(s) \right\rangle &= \left\langle D_F(s) | N_{\iota \cdot \alpha}^r \middle| D_F(s) | N_{\iota \cdot \alpha}^r \right\rangle \Big|_{v_\beta \mapsto v_{\iota \cdot \beta}, \, \xi \mapsto \xi^{-1}} \\
&= \mathrm{Stab}(\iota \cdot \alpha)_{\mathfrak{C}_H, \xi^{-1}} \Big|_{v_\beta \mapsto v_{\iota \cdot \beta}} \\
&= \mathrm{Stab}(\alpha)_{\mathfrak{C}_H^{opp}, \xi^{-1}},
\end{aligned} \tag{272}$$

where $\iota : \{1, \ldots, N\} \mapsto \{N, \ldots, 1\}$ is the longest permutation in $S_N$. This is as expected, changing orientation from right to left in the Morse flow is given by inverting the chamber to $\mathfrak{C}_H^{opp}$ for

---

[13]The inversion of chamber and 'Kähler' parameter in the second equation is due simply to our choice of convention for identification of mass and FI parameters compared to [8]. The division by weights in the attracting/repelling weight spaces is related to the holomorphic normalisations of elliptic stable envelopes described therein.

mass parameters. The inversion of $\xi$ arises from anomaly inflow from an oppositely oriented boundary.

With this wavefunction in hand, we can compute the overlap of left and right enriched Neumann boundary conditions by cutting the path integral,

$$\left\langle N_\alpha^l | N_\beta^r \Big| N_\alpha^l | N_\beta^r \right\rangle = \oint_{\substack{s=v_\gamma t^{-1} \\ \gamma=1,\ldots,N}} \frac{ds}{2\pi i s} \, \eta(q)^2 \vartheta(t^2) Z_C(s) \left\langle N_\alpha^l | D_F(s) \Big| N_\alpha^l | D_F(s) \right\rangle \left\langle D_F(s) | N_\beta^r \Big| D_F(s) | N_\beta^r \right\rangle .$$

(273)

The integrand is periodic in $s$, reflecting the absence of gauge anomalies at either boundary. The JK contour selects poles in $Z_C(s)$ at $s = v_\gamma t^{-1}$ with $\gamma = 1,\ldots,N$. This generates the decomposition into boundary amplitudes,

$$\left\langle N_\alpha^l | N_\beta^r \Big| N_\alpha^l | N_\beta^r \right\rangle = \sum_{\gamma=1}^N \left\langle N_\alpha^l | \gamma \Big| N_\alpha^l | \gamma \right\rangle \left\langle \gamma | N_\alpha^r \Big| \gamma | N_\alpha^r \right\rangle \prod_{\lambda \in \Phi_\gamma} \frac{1}{\vartheta(a^\lambda)} .$$

(274)

Note that $\left\langle N_\alpha^l | \gamma \Big| N_\alpha^l | \gamma \right\rangle$ vanishes for $\gamma > \alpha$, and $\left\langle \gamma | N_\alpha^r \Big| \gamma | N_\alpha^r \right\rangle$ for $\gamma < \beta$, as in our construction of the pole subtraction matrix. Thus if $\alpha < \beta$, the overlap vanishes trivially. It is straightforward to check the diagonal components $\alpha = \beta$ evaluate to 1. For $\alpha > \beta$, the vanishing reduces to a $(\alpha - \beta + 1)$-term theta function identity. These identities are straightforward to check for small $N$ by hand, and can be checked using a computer for larger values of $N$. One thus reproduces orthonormality (270).

Mathematically, this is precisely the duality result of Aganagic and Okounkov [5]

$$\sum_{\gamma \in \text{f.p.}} \frac{\text{Stab}(\alpha)_{\mathfrak{C}_H^{opp}, \xi^{-1}} \Big|_\gamma \text{Stab}(\beta)_{\mathfrak{C}_H, \xi} \Big|_\gamma}{\prod_{\lambda \in \Phi_\gamma} \vartheta(a^\lambda)} = \delta_{\alpha\beta}$$

(275)

of orthonormal elliptic cohomology classes, realised as the orthonormality of left and right enriched Neumann boundary conditions in fixed chambers $\mathfrak{C}_H$ and $\mathfrak{C}_C$.

# 7 Janus Interfaces

Thus far we have tacitly assumed that the real FI and mass parameters are constant. However, the set-up is consistent with varying profiles $\zeta(x^3)$, $m(x^3)$ for the mass parameters in a way that preserves the supercharge $Q$. The computation of boundary amplitudes and overlaps are independent of the profiles of the FI and mass parameters in the $x^3$-direction, except through constraints that their terminal or asymptotic values place on compatible boundary conditions.

Indeed, such non-trivial profiles for the mass parameters are required to correctly interpret some of the computations we have performed. For example, suppose we have left and right boundary conditions $B^l$, $B^r$ compatible with chambers $\mathfrak{C}$, $\mathfrak{C}'$.[14] Then computing the overlap $\langle B^l | B^r \rangle$ as a partition function on $E_\tau \times [-\ell, \ell]$, we must assume a non-trivial profile $m(x^3)$ for the mass parameters interpolating between the chambers $\mathfrak{C}$ at $x^3 = 0$ and $\mathfrak{C}'$ at $x^3 = \ell$.

Since the computations are independent of the profile $m(x^3)$, we can imagine that the mass parameters vary from $\mathfrak{C}$ to another $\mathfrak{C}'$ in a vanishingly small region $\Delta \subset [-\ell, \ell]$. This is known as a supersymmetric Janus interface, which we denote by $J_{\mathfrak{C}, \mathfrak{C}'}$. The overlap of boundary conditions is more correctly expressed as a correlation function of the Janus interface $\langle B^l | J_{\mathfrak{C}, \mathfrak{C}'} | B^r \rangle$. This picture also applies to boundary amplitudes.

---

[14]We drop the $H$ subscript on the chamber $\mathfrak{C}_H$ of mass parameters in the rest of this section.

In this section, we review some aspects of Janus interfaces and re-visit some of the computations done thus far in a new light, especially in the study of exceptional Dirichlet and enriched Neumann boundary conditions. We explain how the computation of correlation functions

$$\langle N_\alpha^{\mathfrak{c}} | J_{\mathfrak{c},\mathfrak{c}'} | N_\beta^{\mathfrak{c}'} \rangle \tag{276}$$

reproduces chamber R-matrices. We exploit independence of the profile $m(x^3)$ to express the R-matrices in terms of elliptic stable envelopes and explain why they satisfy Yang-Baxter.

## 7.1 Explicit Constructions

Supersymmetric Janus interfaces have been studied extensively in the literature, beginning with the case of $4d$ $\mathcal{N} = 4$ Super Yang-Mills [78–81].

Let us consider Janus interfaces from a 3d $\mathcal{N} = 2$ perspective (we follow the notation of [57]) and consider a background vector multiplet $(A^F, \sigma^F, \lambda^F, \bar{\lambda}^F, D^F)$ for a flavour symmetry $F$. The configurations preserving half of the supercharges $Q_+^{++}$, $Q_+^{--}$ are given by

$$A^F = a^F dx^3, \quad \sigma^F = \sigma^F(x^3), \quad D^F = i\partial_3 \sigma^F,$$
$$\lambda^F = 0, \quad \bar{\lambda}^F = 0. \tag{277}$$

Crucially, the profile $\sigma^F(x^3)$ for the real vector multiplet scalar is allowed to vary in the $x^3$-direction provided a compensating profile for the auxiliary field $D^F(x^3)$ is turned on.

In the following, we argue that computations preserving the supercharge $Q = Q_+^{++} + Q_+^{--}$ are independent of the profile in the bulk $x^3 \in (-\ell, \ell)$, and depends only on the terminal values $\sigma^F(\pm\ell)$. We do this by showing that perturbing the action by $\delta\sigma^F(x^3)$ such that $\delta\sigma^F(\pm\ell) = 0$ results in a $Q$-exact deformation of the action. The usual localisation argument then finishes the argument.

We apply this to the case where $F = T_H$, $T_C$ and the vector multiplet scalar $\sigma^F = m, \zeta$. However, since in this paper we fix a chamber for the FI parameter and thus a Higgs branch $X$, we are primarily interested in varying profiles for the mass parameters.

### 7.1.1 Mass Janus

Let us first consider the case of $F = T_H$ with varying mass parameters $\sigma^H = m(x^3)$ and background auxiliary field $D^H = im'(x^3)$. We consider the action of a 3d $\mathcal{N} = 2$ chiral multiplet $(\phi, \bar{\phi}, \psi, \bar{\psi}, F, \bar{F})$ charged under the flavour symmetry $T_H$. The part of the Lagrangian that depends on $m(x^3)$ is

$$\mathcal{L}_m = \bar{\phi} m^2(x^3) \phi + 2\bar{\phi}\sigma m(x^3)\phi - \bar{\phi} m'(x^3)\phi + i\bar{\psi} m(x^3)\psi, \tag{278}$$

where $m$ and $\sigma$ act in the appropriate $G$ and $G_H$ representations. A variation in the profile $m(x^3) \to m(x^3) + \delta m(x^3)$ results in a change in the Lagrangian

$$\delta\mathcal{L}_m = \bar{\phi}\delta m^2 \phi + 2\bar{\phi}\delta m(\sigma + m)\phi + \bar{\phi}\delta m \partial_3 \phi + \partial_3\bar{\phi}\delta m\phi - \partial_3(\bar{\phi}\delta m\phi) + i\bar{\psi}\delta m\psi. \tag{279}$$

It is not hard to show that

$$Q \cdot V_m = \delta\mathcal{L}_m + \partial_3(\bar{\phi}\delta m\phi) - i(\bar{\phi}\delta m F + \bar{F}\delta m\phi) - \bar{\phi}\delta m^2 \phi, \tag{280}$$

where

$$V_m = \frac{\delta m}{2i} \left( \bar{\phi}\delta m\psi_- + \bar{\psi}_-\delta m\phi \right). \tag{281}$$

Although it appears as given that $\delta\mathcal{L}_m$ is not $Q$-exact, we now show that the last two terms in $Q \cdot V_m$ can be absorbed into a re-definition of the auxiliary fields.

Let $\mathcal{Z}_m$ denote the partition function with the initial mass profile $m(x^3)$, and $S[\Phi]_m$ the corresponding action. Then by the usual localisation argument, we have

$$\mathcal{Z}_m = \int D\Phi e^{-S[\Phi]_m} = \int D\Phi e^{-S[\Phi]_m - Q \cdot \int V_m}, \qquad (282)$$

where $\Phi$ denotes collectively all the dynamical fields.

The auxiliary fields $F$ and $\bar{F}$, included to ensure closure of the supersymmetry algebra on fermions, appear in the action only through the quadratic term $\int_{E_\tau \times I} \bar{F} F$. Denoting $\Phi'$ to be the collection of fields without $F, \bar{F}$, and $S[\Phi']_m$ the action minus this term, then

$$S[\Phi]_m + Q \cdot V_m = S[\Phi']_m + \int_{E_\tau \times I} \delta \mathcal{L}_m + \partial_3(\bar{\phi} \delta m \phi) + (\bar{F} - i\bar{\phi}\delta m)(F - i\delta m \phi), \qquad (283)$$

using (280). Substituting the above into (282) and redefining

$$\bar{F} - i\bar{\phi}\delta m \to \bar{F}, \qquad F - i\delta m \phi \to F, \qquad (284)$$

then we have[15]

$$\mathcal{Z}_m = \int D\Phi' DF D\bar{F} e^{-S[\Phi']_m - \int (\bar{F}F + \delta \mathcal{L}_m) - \int \partial_3(\bar{\phi}\delta m \phi)}$$

$$= \int D\Phi e^{-S[\Phi]_{m+\delta m} - \int \partial_3(\bar{\phi}\delta m \phi)}. \qquad (285)$$

So provided that $\delta m(\pm\ell) = 0$, it is true that $\mathcal{Z}_{m+\delta m} = \mathcal{Z}_m$. We have shown that $\mathcal{Z}_m$ is independent of the interior profile of $m(x^3)$ and only on the values $m(\pm\ell)$ at the end points of the interval.

### 7.1.2 FI Janus

The FI parameter consists of the real scalar of a background $\mathcal{N} = 2$ vector multiplet $(A^C, \sigma^C, \lambda^C, \bar{\lambda}^C, D^C)$ coupled to the gauge symmetry via a mixed Chern-Simons term. In an $\mathcal{N} = 4$ theory, it is the real scalar $\zeta = \zeta^{+-}$ in an $\mathcal{N} = 4$ twisted vector multiplet, coupled to a $\mathcal{N} = 4$ vector multiplet for the gauge symmetry via an $\mathcal{N} = 4$ CS term.

In either case, we will be interested in giving the background scalar $\sigma^C$ a non-trivial profile in $x^3$, i.e $\zeta(x^3)$. For this to be BPS we require $D^C = i\partial_3 \sigma^C$. Therefore the (bulk) FI term is

$$S_{\text{FI}} = \int_{E_\tau \times I} i\zeta(x^3) D - \zeta'(x^3)\sigma. \qquad (286)$$

For the coupling of the gauge field to a background for $A^C$, see e.g. [19, 20].[16] Note that

$$Q \cdot S_{\text{FI}} = \frac{1}{2} \int_{E_\tau \times I} \left( \zeta \bar{\epsilon}\gamma^\mu D_\mu \lambda + \zeta D_\mu \bar{\lambda}\gamma^\mu \epsilon - \zeta'\bar{\epsilon}\lambda + \zeta'\bar{\lambda}\epsilon \right)$$

$$= -\frac{1}{2} \int_{E_\tau \times I} \partial_3 \left( \zeta(\lambda_+ + \bar{\lambda}_+) \right), \qquad (287)$$

---

[15]We have assumed that since the redefinition (284) is linear, the measure is invariant. There is an additional assumption in the following. In Euclidean signature all fields are complexified and barred and unbarred fields are independent. In the path integral there is a choice of middle dimensional contour in field space for the bosonic fields, the canonical choice relating bar and unbarred fields by complex conjugation: $\bar{F} = F^\dagger$. Then the term $\int \bar{F}F$ is positive definite in the action. The redefinition (284) deforms away from this contour. To retain a positive definite action we assume the contour can be deformed back to the canonical choice without changing the answer.

[16]These references include couplings for NS-NS boundary conditions on $E_\tau$: the analogous results for the R-R sector are obtained from dropping the dependence on $E_\tau$ coordinates in the boundary conditions and spinors, and taking the curvature of $HS^2$ to infinity.

so that $S_{\text{FI}}$ is supersymmetric under $Q$, provided Neumann boundary conditions for the vector multiplet (implying $\lambda_+ = \bar\lambda_+ = 0$) are prescribed at both boundaries. In passing to the second line we have dropped the total derivatives in the $x^{1,2}$ directions, as $\lambda$ and $\bar\lambda$ obey R-R boundary conditions. For Dirichlet boundary conditions one must add boundary Chern-Simons terms to preserve supersymmetry, as in [20]. We will not consider this case.

Under a variation $\zeta(x^3) \to \zeta(x^3) + \delta\zeta(x^3)$, the variation is $Q$-exact up to boundary terms:

$$
\begin{aligned}
\delta S_{\text{FI}} &= \int_{E_\tau \times I} i\,\delta\zeta (D - i\partial_3\sigma) - \partial_3(\delta\zeta\,\sigma) \\
&= Q \cdot \left[ \int_{E_\tau \times I} \frac{i\,\delta\zeta}{4} (\bar\lambda_- - \lambda_-) \right] - \int_{E_\tau \times I} \partial_3(\delta\zeta\,\sigma).
\end{aligned}
\tag{288}
$$

Thus the path integral is independent of the interior profile of the FI parameter $\zeta(x^3)$. If $\sigma$ is fixed to a non-zero boundary 2d FI parameter $\sigma| = t_{2\mathrm{d}}$, as is the case for the enriched Neumann boundary conditions encountered in this work, then the above shows that the partition function does depend on the terminal value(s) of the FI Janus $\zeta(\pm\ell)$.

## 7.2 Mass Parameter Janus

Let us first reconsider the boundary amplitudes of exceptional Dirichlet and enriched Neumann boundary conditions. Both sets of boundary conditions are defined in the presence of mass parameters in some chamber $\mathfrak{C}$. As we now keep track of chamber structure let us write the left boundary conditions compatible with mass parameters this chamber as $\langle D_\alpha^{\mathfrak{C}}|$, $\langle N_\alpha^{\mathfrak{C}}|$, and right boundary conditions as $|D_\alpha^{\mathfrak{C}}\rangle$, $|N_\alpha^{\mathfrak{C}}\rangle$.

We considered the boundary amplitudes representing equivariant cohomology classes on $X$ by taking overlaps with the supersymmetric ground states $|\alpha\rangle$ defined at vanishing mass parameters. We would previously have denoted such boundary amplitudes as $\langle\beta|D_\alpha^{\mathfrak{C}}\rangle$, $\langle\beta|N_\alpha^{\mathfrak{C}}\rangle$. However, in light of the discussion above, it is more appropriate to regard them as correlation functions

$$
\begin{aligned}
\langle\alpha|\mathcal{J}_{0,\mathfrak{C}}|D_\beta^{\mathfrak{C}}\rangle &= \Theta(\Phi_\beta^{\mathfrak{C},+})\delta_{\alpha\beta}, & \langle D_\alpha^{\mathfrak{C}}|\mathcal{J}_{\mathfrak{C},0}|\beta\rangle &= \Theta(\Phi_\beta^{\mathfrak{C},-})\delta_{\alpha\beta}, \\
\langle\alpha|\mathcal{J}_{0,\mathfrak{C}}|N_\beta^{\mathfrak{C}}\rangle &= \text{Stab}(\beta)_{\mathfrak{C},\xi}\big|_\alpha, & \langle N_\alpha^{\mathfrak{C}}|\mathcal{J}_{\mathfrak{C},0}|\beta\rangle &= \text{Stab}(\alpha)_{\mathfrak{C}^{opp},\xi^{-1}}\big|_\beta,
\end{aligned}
\tag{289}
$$

where $\mathcal{J}_{0,\mathfrak{C}}$ and $\mathcal{J}_{\mathfrak{C},0}$ are Janus interfaces interpolating between vanishing mass parameters and mass parameters in the chamber $\mathfrak{C}$, from left to right and vice versa. In the above we have emphasised the chamber dependence in the sets of positive and negative weights $\Phi_\alpha^{\mathfrak{C},\pm}$. Additionally we have used the shorthand where if $W$ is a set of weights, then

$$
\Theta(\pm W) \equiv \prod_{\lambda \in W} \vartheta(a^\lambda)^{\pm 1}.
\tag{290}
$$

It will also be convenient to define the matrix of boundary amplitudes

$$
S_{\alpha\beta}^{\mathfrak{C},\xi} = \text{Stab}(\beta)_{\mathfrak{C},\xi}\big|_\alpha.
\tag{291}
$$

We may then use the orthogonality of exceptional Dirichlet and enriched Neumann boundary conditions to write

$$
\begin{aligned}
\langle\alpha|J_{0,\mathfrak{C}} &= \langle D_\alpha^{\mathfrak{C}}|\Theta(\Phi_\alpha^{\mathfrak{C},+}) & J_{\mathfrak{C},0}|\beta\rangle &= \Theta(\Phi_\beta^{\mathfrak{C},-})|D_\beta^{\mathfrak{C}}\rangle \\
&= \sum_{\gamma \leq \alpha} \langle N_\gamma^{\mathfrak{C}}|S_{\alpha\gamma}^{\mathfrak{C},\xi}, & &= \sum_{\gamma \geq \beta} S_{\beta\gamma}^{\mathfrak{C}^{opp},\xi^{-1}}|N_\gamma^{\mathfrak{C}}\rangle,
\end{aligned}
\tag{292}
$$

where in the second line we recall for example that $S_{\beta\gamma}^{\mathfrak{C}}$ is lower triangular with respect to the partial ordering induced by the Morse flow with respect to mass parameters in the chamber $\mathfrak{C}$.

The first line for exceptional Dirichlet is the correct interpretation of the relationship between supersymmetric ground states with $m = 0$ and $m \in \mathfrak{C}$ derived using the infinite-dimensional quantum mechanics model in section 3. Namely, in transporting the supersymmetric ground states from $m = 0$ to $m \in \mathfrak{C}$, the supersymmetric ground states are related by a factor (69).

The second equation shows that the elliptic stable envelope provides the matrix elements that express how to decompose the supersymmetric ground states at $m = 0$ in terms of states generated by enriched Neumann boundary conditions when transported to $m \in \mathfrak{C}$.

Let us go one step further and apply these equations to a general boundary condition compatible with mass parameters in the chamber $\mathfrak{C}$. We first consider a right boundary condition, which we denote at $m = 0$ by $B$ and at $m \in \mathfrak{C}$ by $B^{\mathfrak{C}}$. We find immediately:

$$
\begin{aligned}
\langle \alpha|B\rangle &= \Theta(\Phi_{\alpha}^{+})\langle D_{\alpha}^{\mathfrak{C}}|B^{\mathfrak{C}}\rangle\,, \\
\langle \alpha|B\rangle &= \sum_{\beta} S_{\alpha\beta}^{\mathfrak{C},\xi}\langle N_{\beta}^{\mathfrak{C}}|B^{\mathfrak{C}}\rangle\,.
\end{aligned}
\tag{293}
$$

Note that both of these equations relate quantities on the right, $\{\langle D_{\alpha}^{\mathfrak{C}}|B^{\mathfrak{C}}\rangle\}$ and $\{\langle N_{\alpha}^{\mathfrak{C}}|B^{\mathfrak{C}}\rangle\}$, which transform as sections of holomorphic line bundles on $E_T(\{\alpha\}) = \bigsqcup_{\alpha} E_{T_f}^{(\alpha)}$ to boundary amplitudes on the left that glue to a section of a line bundle on $E_T(X)$.

This is the physical realisation of the construction in [5] which realises the stable envelope as map of sections of holomorphic line bundles on $\bigsqcup_{\alpha} E_{T_f}^{(\alpha)}$ to those on $E_T(X)$.

### 7.3 $R$-matrices

Correlation functions of particular interest are the overlap of enriched Neumann $N_{\alpha}^{\mathfrak{C}'}$, $N_{\beta}^{\mathfrak{C}}$ in a pair of distinct chambers. Following the discussion above, we regard this as a correlation function

$$
R_{\alpha\beta}^{\mathfrak{C}',\mathfrak{C}} := \langle N_{\alpha}^{\mathfrak{C}'}|\mathcal{J}_{\mathfrak{C}',\mathfrak{C}}|N_{\beta}^{\mathfrak{C}}\rangle\,.
\tag{294}
$$

Note that if we choose $\mathfrak{C} = \mathfrak{C}'$ then $R_{\alpha,\beta}^{\mathfrak{C},\mathfrak{C}} = \delta_{\alpha\beta}$, as the enriched Neumann boundary conditions are orthonormal.

These correlation functions obey an important property as a consequence of the independence of the profile $m(x^3)$ connecting the chambers $\mathfrak{C}$, $\mathfrak{C}'$. Suppose that $\mathfrak{C}$, $\mathfrak{C}'$ are not neighbouring chambers, meaning they are not separated by a single hyperplane $\mathbb{W}_{\alpha\beta} \subset \mathfrak{t}_H$. Then by deforming the profile $m(x^3)$, we can decompose

$$
\mathcal{J}_{\mathfrak{C}',\mathfrak{C}} = \mathcal{J}_{\mathfrak{C}',\mathfrak{C}_1}\mathcal{J}_{\mathfrak{C}_1,\mathfrak{C}_2}\cdots\mathcal{J}_{\mathfrak{C}_n,\mathfrak{C}}\,,
\tag{295}
$$

as a composition of elementary Janus interfaces connecting neighbouring chambers. By expanding in enriched Neumann boundary conditions in each of the intermediate chambers $\mathfrak{C}_1, \cdots, \mathfrak{C}_n$, we find

$$
R_{\alpha\beta}^{\mathfrak{C}',\mathfrak{C}} = \sum_{\gamma_1,\gamma_2,\dots,\gamma_n} R_{\alpha\gamma_1}^{\mathfrak{C}',\mathfrak{C}_1}R_{\gamma_1\gamma_2}^{\mathfrak{C}_1,\mathfrak{C}_2}\cdots R_{\gamma_n\beta}^{\mathfrak{C}_n,\mathfrak{C}}\,.
\tag{296}
$$

Moreover, different decompositions must yield the same result due to invariance under deformations of the profile $m(x^3)$. Special cases of this relation include the Yang-Baxter equation and unitarity condition obeyed by R-matrices of quantum integrable models. Therefore, following [71], we refer to these correlation functions as chamber R-matrices.

The chamber R-matrices can be computed directly using supersymmetric localisation on $E_{\tau} \times [0,\ell]$ with the left and right enriched Neumann boundary conditions. However, it is also

convenient to decompose the result in terms of boundary amplitudes. For this purpose, we decompose the Janus interface as

$$\mathcal{J}_{\mathfrak{C}',\mathfrak{C}} = \mathcal{J}_{\mathfrak{C}',0}\mathcal{J}_{0,\mathfrak{C}}, \tag{297}$$

where we think about smooth deforming the profile $m(x^3)$ as illustrated in figure 21. This

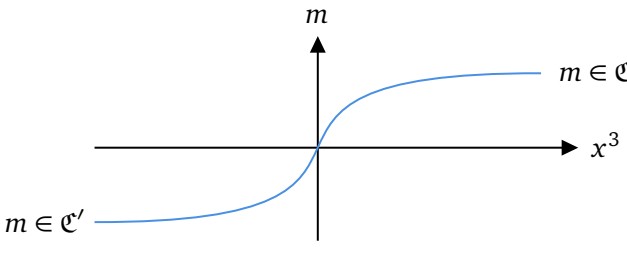

Figure 21

allows us to decompose the chamber R-matrices as follows

$$
\begin{aligned}
R_{\alpha\beta}^{\mathfrak{C}',\mathfrak{C}} &= \langle N_\alpha^{\mathfrak{C}'}|\mathcal{J}_{\mathfrak{C}',\mathfrak{C}}|N_\beta^{\mathfrak{C}}\rangle \\
&= \langle N_\alpha^{\mathfrak{C}'}|\mathcal{J}_{\mathfrak{C}',0}\mathcal{J}_{0,\mathfrak{C}}|N_\beta^{\mathfrak{C}}\rangle \\
&= \sum_\gamma \Theta(-\Phi_\gamma)\langle N_\alpha^{\mathfrak{C}'}|\mathcal{J}_{\mathfrak{C}',0}|\gamma\rangle\langle\gamma|\mathcal{J}_{0,\mathfrak{C}}|N_\beta^{\mathfrak{C}}\rangle \\
&= (S^{\mathfrak{C}'})_{\alpha\gamma}^{-1}S_{\gamma\beta}^{\mathfrak{C}},
\end{aligned} \tag{298}
$$

which reproduces the construction of chamber R-matrices in terms of elliptic stable envelopes introduced in [5]. We have used the orthogonality of enriched Neumann boundary conditions:

$$\Theta(-\Phi_\beta)\langle N_\alpha^{\mathfrak{C}}|\mathcal{J}_{\mathfrak{C},0}|\beta\rangle = \Theta(-\Phi_\beta)S_{\beta\alpha}^{\mathfrak{C}^{opp},\xi^{-1}} = (S^{\mathfrak{C},\xi})_{\alpha\beta}^{-1}, \tag{299}$$

as discussed in section 6.4.

Note that one could have instead proceeded here by decomposing into wavefunctions to get an integral formula:

$$R_{\alpha\beta}^{\mathfrak{C}',\mathfrak{C}} = \oint \frac{ds}{2\pi i s}\mathcal{Z}_V\mathcal{Z}_C(s)\langle N_\alpha^{\mathfrak{C}'}|D_F(s)\rangle\langle D_F(s)|N_\beta^{\mathfrak{C}}\rangle. \tag{300}$$

One could also arrive at the same integral formula by collapsing the interval directly and evaluating the elliptic genus of the effective $2d$ theory consisting of boundary $\mathbb{C}^*$ chiral multiplets and surviving bulk matter, as in section 4. Of course, evaluating the residues decomposes this into boundary amplitudes.

### 7.3.1 Example: Supersymmetric QED

Let us specialise to our example of supersymmetric QED. Let $\mathfrak{C}_0 = \{m_1 > m_2 > \ldots > m_N\}$ be our initial choice of chamber. A generic chamber $\mathfrak{C}$ can therefore be expressed as $\mathfrak{C} = \pi \cdot \mathfrak{C}_0$ where $\pi \in S_N$. In a way analogous to the arguments in section 6.4, it is not hard to see that the wavefunction of a right enriched Neumann boundary condition in the chamber $\mathfrak{C}$ is:

$$\langle D_F(s)|N_\alpha^{\mathfrak{C}}\rangle = \text{Stab}(\pi^{-1}\cdot\alpha)_{\mathfrak{C}_0,\xi}\big|_{v_\beta\to v_{\pi\cdot\beta}}. \tag{301}$$

It is of course also possible to derive this directly by using the mirror symmetry interface.

Let us now compute the chamber $R$-matrix explicitly for supersymmetric QED with $N = 2$. The default chamber is $\mathfrak{C} = \{m_1 > m_2\}$. The only non-trivial chamber R-matrix $R^{\mathfrak{C}',\mathfrak{C}}_{\alpha\beta}$ comes from choosing the opposite chamber $\mathfrak{C}' = \{m_1 < m_2\}$.

Using the boundary amplitudes, obtained via (272) and (301),

$$R^{\mathfrak{C}',\mathfrak{C}} = S^{\mathfrak{C}'-1} S^{\mathfrak{C}} = \begin{pmatrix} \frac{1}{\vartheta(\frac{v_2}{v_1})} & \frac{\vartheta(t^{-2})\vartheta(\xi^{-1}\frac{v_2}{v_1})}{\vartheta(\xi^{-1})\vartheta(\frac{v_1}{v_2})\vartheta(t^{-2}\frac{v_2}{v_1})} \\ 0 & \frac{1}{\vartheta(t^{-2}\frac{v_2}{v_1})} \end{pmatrix} \begin{pmatrix} \vartheta(t^{-2}\frac{v_1}{v_2}) & 0 \\ \frac{\vartheta(t^{-2})\vartheta(\xi\frac{v_2}{v_1})}{\vartheta(\xi)} & \vartheta(\frac{v_1}{v_2}) \end{pmatrix}, \tag{302}$$

which may be simplified by the 3-term theta function identity $\vartheta(ab)\vartheta(a/b)\vartheta(c)^2 + \text{cyclic} = 0$ to recover the $\mathfrak{sl}_2$ $R$-matrix

$$R^{\mathfrak{C}',\mathfrak{C}} = \begin{pmatrix} \frac{\vartheta(\xi t^{-2})\vartheta(\xi t^2)\vartheta(\frac{v_1}{v_2})}{\vartheta(\xi)^2\vartheta(t^{-2}\frac{v_2}{v_1})} & \frac{\vartheta(t^{-2})\vartheta(\xi^{-1}\frac{v_2}{v_1})}{\vartheta(\xi^{-1})\vartheta(t^{-2}\frac{v_2}{v_1})} \\ \frac{\vartheta(t^{-2})\vartheta(\xi\frac{v_2}{v_1})}{\vartheta(\xi)\vartheta(t^{-2}\frac{v_2}{v_1})} & \frac{\vartheta(\frac{v_1}{v_2})}{\vartheta(t^{-2}\frac{v_2}{v_1})} \end{pmatrix}, \tag{303}$$

as in [5]. Similarly, by computing $\text{Stab}(\alpha)_{\mathfrak{C}'}\big|_\beta$, the $R$-matrix in the opposite direction is given by

$$R^{\mathfrak{C},\mathfrak{C}'} = \begin{pmatrix} \frac{\vartheta(\frac{v_2}{v_1})}{\vartheta(t^{-2}\frac{v_1}{v_2})} & \frac{\vartheta(t^{-2})\vartheta(\xi\frac{v_1}{v_2})}{\vartheta(\xi)\vartheta(t^{-2}\frac{v_1}{v_2})} \\ \frac{\vartheta(t^{-2})\vartheta(\xi^{-1}\frac{v_1}{v_2})}{\vartheta(\xi^{-1})\vartheta(t^{-2}\frac{v_1}{v_2})} & \frac{\vartheta(\xi t^{-2})\vartheta(\xi t^2)\vartheta(\frac{v_2}{v_1})}{\vartheta(\xi)^2\vartheta(t^{-2}\frac{v_1}{v_2})} \end{pmatrix}. \tag{304}$$

Then using the same cyclic identity as before, we recover the unitarity condition

$$R^{\mathfrak{C}',\mathfrak{C}} R^{\mathfrak{C},\mathfrak{C}'} = \mathbb{I}_{2x2} \tag{305}$$

obeyed by $R$-matrices, as expected.

# Acknowledgements

The authors would like to thank Tudor Dimofte, Hunter Dinkins, David Tong, Justin Hilburn, Atul Sharma, and Ziruo Zhang for useful discussions on the topics in this paper. We are particularly grateful to Samuel Crew for useful discussions and collaboration on part of this project. The work of MB is supported by the EPSRC Early Career Fellowship EP/T004746/1 "Supersymmetric Gauge Theory and Enumerative Geometry" and the STFC Research Grant ST/T000708/1 "Particles, Fields and Spacetime". The work of DZ is partially supported by STFC consolidated grants ST/P000681/1, ST/T000694/1.

# A Gluing Property

In this appendix we demonstrate the gluing property of line bundles $\{\mathcal{L}'_\alpha\}$ required in section 3.5.1. First, we recall some canonical results on line bundles over elliptic curves [82], using the rephrasing in [83, 84] of these results in the language of factors of automorphy.

It will be convenient to regard the background flat connections $z_f = (z_C, z_H, z_t)$ as coordinates on $\mathfrak{t}_f^{\mathbb{C}} = \mathfrak{t}_f \otimes_{\mathbb{R}} \mathbb{C}$. Then global background gauge transformations $z_f \rightarrow z_f + \nu_f + \mu_f \tau$ form a group $\Lambda_f$ of deck transformations so that we have a universal covering:

$$\mathfrak{t}_f^{\mathbb{C}} \rightarrow \mathfrak{t}_f^{\mathbb{C}}/\Lambda_f = E_{T_f}, \qquad z_f \mapsto [z_f]. \tag{306}$$

A factor of automorphy is a holomorphic function $F : \Delta_f \times \mathfrak{t}_f^{\mathbb{C}} \to \mathbb{C}^*$ obeying

$$F(z_f, \nu_f + \mu_f \tau + \nu_f' + \mu_f' \tau) = F(z_f + \nu_f + \mu_f \tau, \nu_f' + \mu_f' \tau) F(z_f, \nu_f + \mu_f \tau). \tag{307}$$

There is an equivalence relation on factors of automorphy, where $F \sim F'$ if there exists a holomorphic function $H : \mathfrak{t}_f^{\mathbb{C}} \to \mathbb{C}^*$ obeying

$$H(z_f + \nu_f + \mu_f \tau) F(z_f, \nu_f + \mu_f \tau) = F'(z_f, \nu_f + \mu_f \tau) H(z_f). \tag{308}$$

Importantly, isomorphism classes of line bundles over $E_{T_f}$ are in 1-1 correspondence with equivalence classes $[F]$. Sections of a line bundle associated to $F$ are 1-1 with holomorphic functions $s : \mathfrak{t}_f^{\mathbb{C}} \to \mathbb{C}$ satisfying

$$s(z_f + \nu_f + \mu_f \tau) = F(z_f, \nu_f + \mu_f \tau) s(z_f). \tag{309}$$

In our set-up, the line bundles $\mathcal{L}_\alpha'$ have factors of automorphy

$$F_\alpha(z_f, \nu_f + \mu_f \tau) = e^{-2\pi i \left( K_\alpha'(z_f, \mu_f) + K_\alpha'(\mu_f, z_f) + \tau K_\alpha'(\mu_f, \mu_f) \right)}, \tag{310}$$

where $K_\alpha'$ are the shifted supersymmetric Chern-Simons couplings as in equation (85). It is not difficult to check that $F_\alpha$ obeys property (307).

Now consider a weight $\lambda \in \Phi_\alpha \cup (-\Phi_\beta)$ labelling an interior edge of the GKM diagram. Mass parameters $x$ lying on the hyperplane $\mathbb{W}_\lambda = \{\lambda \cdot x = 0\}$ generate a codimension 1 subtorus $T^\lambda$ of $T$, which leaves point-wise fixed the irreducible curve $\Sigma_\lambda \cong \mathbb{P}^1$ connecting the fixed points $\alpha$ and $\beta$. The weight $\lambda$ then defines a doubly-periodic array of hyperplanes in $\mathfrak{t}_f^{\mathbb{C}}$ where

$$\lambda \cdot z \in \mathbb{Z} + \tau \mathbb{Z}, \tag{311}$$

recalling that $z = (z_H, z_t)$. Let us suppose $\lambda \cdot z = a + b\tau$ for $a, b \in \mathbb{Z}$. This breaks the group of large background gauge transformations to $\Lambda_f^\lambda$, whose elements obey $\lambda \cdot \nu = \lambda \cdot \mu = 0$, where $\mu = (\mu_H, \mu_t)$ etc. Then we may define

$$E_{T_f}^\lambda = \{z_f \in \mathfrak{t}_f^{\mathbb{C}} | \lambda \cdot z = a + b\tau\}/\Lambda_f^\lambda \hookrightarrow E_{T_f}, \tag{312}$$

a codimension-one subtorus of $E_{T_f}$, where we have left the $a, b$ dependence implicit. Analogous definitions to the above can be made for line bundles over $E_{T_f}^\lambda$.

The result required in section 3.5.1 is that for all such $\lambda$, the restriction (more accurately the pull-back under the above inclusion) of the line bundles $\mathcal{L}_\alpha'$ and $\mathcal{L}_\beta'$ to $E_{T_f}^\lambda$ (for any $a, b \in \mathbb{Z}$) are isomorphic. We demonstrate this using their factors of automorphy.

On $E_{T_f}^\lambda$ (the pull-back of) $\mathcal{L}_\alpha'$, $\mathcal{L}_\beta'$ have factors of automorphy given by restricting $F_\alpha$, $F_\beta$ to $\lambda \cdot z = a + b\tau$ and $\Lambda_f$ to $\Lambda_f^\lambda$. In the remainder of this appendix, we use $|_{E_{T_f}^\lambda}$ to denote these restrictions. Then isomorphism of $\mathcal{L}_\alpha'$ and $\mathcal{L}_\beta'$ on the locus is equivalent to the existence of a holomorphic function $H(z_f)$ such that

$$H(z_f + \nu_f + \mu_f \tau) F_\beta(z_f, \nu_f + \mu_f \tau) = F_\alpha(z_f, \nu_f + \mu_f \tau) H(z_f) \Big|_{E_{T_f}^\lambda}. \tag{313}$$

To demonstrate this, we first collect the following two results.

1. First consider the contribution from $\kappa_\alpha + \kappa_\alpha^C$ to the ratio $F_\alpha/F_\beta$ of factors of automorphy

$$e^{-2\pi i\left((\kappa_{\alpha\beta}+\kappa_{\alpha\beta}^C)(z_f,\mu_f)+(\kappa_{\alpha\beta}+\kappa_{\alpha\beta}^C)(\mu_f,z_f)+\tau(\kappa_{\alpha\beta}+\kappa_{\alpha\beta}^C)(\mu_f,\mu_f)\right)}, \tag{314}$$

where we have used the shorthand $\kappa_{\alpha\beta} = \kappa_\alpha - \kappa_\beta$. Note the relation on critical points of Morse flows

$$h|_\alpha - h|_\beta = \langle \zeta, [\Sigma_\lambda]\rangle (\lambda \cdot m), \tag{315}$$

where $\zeta$ is identified as an element of $H^2(X, \mathbb{R})$. From equation (20), one has that (314) equals

$$e^{-2\pi i\langle \mu_C, [\Sigma_\lambda]\rangle(\lambda\cdot z)} = e^{-2\pi i b\langle \mu_C, [\Sigma_\lambda]\rangle\tau} \tag{316}$$

on restriction to $E_{T_f}^\lambda$. We have used $\langle \mu_C, [\Sigma_\lambda]\rangle \in \mathbb{Z}$.

2. Now consider the contribution from the sum over $\Phi_\alpha$. By construction the $T^\lambda$-weight of $T_\alpha\Sigma_\lambda \subset T_\alpha X$ (and thus $T_\beta\Sigma_\lambda$) is 0. Since $T^\lambda$ fixes point-wise the curve $\Sigma_\lambda$, the (quantised) weights of $T^\lambda$ are constant over $\Sigma_\lambda$ by continuity, and in particular coincide at $\alpha$ and $\beta$. Thus there is a pairing of weights $\lambda'_\alpha \in \Phi_\alpha$ and $\lambda'_\beta \in \Phi_\beta$ such that

$$(\lambda'_\alpha - \lambda'_\beta) \cdot z = C_{\alpha\beta}(\lambda \cdot z). \tag{317}$$

The constants $\{C_{\alpha\beta}\}$ are integer, as the characters of the isotropy representations of $T_\alpha X$, $T_\beta X$ agree when $e^{2\pi i\lambda\cdot z} = 1$, in particular when $\lambda \cdot z \in \mathbb{Z}\backslash\{0\}$. Note that (317) also implies $\lambda'_\alpha \cdot \mu = \lambda'_\beta \cdot \mu$ for all $\nu_f + \mu_f\tau \in \Lambda_f^\lambda$.

The contribution from $\Phi_\alpha, \Phi_\beta$ in the ratio $F_\alpha/F_\beta$ is:

$$e^{-\pi i\left(\sum_{\lambda'_\alpha \in \Phi_\alpha}(\lambda'_\alpha\cdot z)(\lambda'_\alpha\cdot\mu)+\frac{\tau}{2}\sum_{\lambda'_\alpha \in \Phi_\alpha}(\lambda'_\alpha\cdot\mu)^2 - \sum_{\lambda'_\beta \in \Phi_\beta}(\lambda'_\beta\cdot z)(\lambda'_\beta\cdot\mu)-\frac{\tau}{2}\sum_{\lambda'_\beta \in \Phi_\beta}(\lambda'_\beta\cdot\mu)^2\right)}. \tag{318}$$

Using (317), on restriction to $E_{T_f}^\lambda$ this equals

$$e^{-\pi i\sum_{\lambda'}(\lambda'_\alpha-\lambda'_\beta)\cdot z(\lambda'\cdot\mu)} = e^{-\pi i b\sum_{\lambda'}C_{\alpha\beta}(\lambda'\cdot\mu)\tau}, \tag{319}$$

where the sum is over $\Phi_\alpha$ or $\Phi_\beta$, the answer is the same. The $a$-dependence drops out as the symplectic pairing of weights in $\Phi_\alpha$ (or $\Phi_\beta$) implies $\frac{1}{2}\sum_{\lambda'} C_{\alpha\beta}(\lambda' \cdot \mu) \in \mathbb{Z}$.

Using these results in conjunction, we have

$$\left.\frac{F_\alpha(z_f, \nu_f + \mu_f\tau)}{F_\beta(z_f, \nu_f + \mu_f\tau)}\right|_{E_{T_f}^\lambda} = e^{-2\pi i b\left(\langle \mu_C, [\Sigma_\lambda]\rangle+\frac{1}{2}\sum_{\lambda'}C_{\alpha\beta}(\lambda'\cdot\mu)\right)\tau}. \tag{320}$$

Then the function

$$H(z_f) = e^{2\pi i b\left(\langle z_C, [\Sigma_\lambda]\rangle+\frac{1}{2}\sum_{\lambda'}C_{\alpha\beta}\lambda'\cdot z\right)} \tag{321}$$

obeys (313), where the $\nu_f$-dependence drops out by using $\langle \nu_C, [\Sigma_\lambda]\rangle \in \mathbb{Z}$ for all $\nu_C$, and the symplectic pairing of the weights $\lambda'$ in $\Phi_\alpha$ or $\Phi_\beta$. We have therefore established the isomorphism of $\mathcal{L}'_\alpha$ and $\mathcal{L}'_\beta$ on $E_{T_f}^\lambda$.

# B   Elliptic Genus of $\mathbb{C}^*$ Chiral Multiplets

In this appendix we give the computation of the contribution to the wavefunction of the mirror duality interface, i.e. the mother function in section 6.3.1, from the $\mathbb{C}^*$-valued $(2,2)$ chiral multiplets $\Phi_\beta$ which appear in interface. Our strategy is to first compute it in the NS-NS sector. In this sector the contribution takes the form of an elliptic genus which coincides with the superconformal index, and therefore it is possible to use an operator counting argument to compute it. We then perform a spectral flow to R-R sector. We will stick to our example of supersymmetric QED, the generalisation to arbitrary abelian theories is straightforward.

In the NS-NS sector, $q = e^{2\pi i\tau}$ grades the operator counting by the left-moving Hamiltonian $H_L$, which by unitary bound arguments is equivalent to a grading by $J_3 + \frac{1}{4}(U(1)_V + U(1)_A)$. We claim that the contribution from the $\mathbb{C}^*$ matter is given in NS-NS sector by:

$$\prod_{\beta=1}^{N} q^{\frac{1}{4}} t \frac{\vartheta(q^{\frac{1}{2}}t^2)\vartheta(C_\beta D_\beta)}{\vartheta(C_\beta)\vartheta(D_\beta)} , \tag{322}$$

where we have identified:

- $C_\beta \equiv q^{\frac{1}{4}} t (s v_\beta^{-1})^{-\varepsilon_\beta}$ as the charge of $e^{-\varepsilon_\beta \phi_\beta}$, which creates translation modes of $\phi_\beta$.

- $D_\beta \equiv q^{\frac{1}{4}} t^{-1} (\tilde{s}_\beta/\tilde{s}_{\beta-1})^{\varepsilon_\beta}$ as the charge of $e^{\varepsilon_\beta \widetilde{\phi}_\beta}$, the T-dual, which creates winding modes of $\phi_\beta$.

To prove this, using the identity

$$-q^{\frac{1}{12}} \frac{\vartheta(C_\beta D_\beta)}{\vartheta(C_\beta)\vartheta(D_\beta)} = \frac{1}{(q;q)_\infty^2} \sum_{n\in Z} \frac{(C_\beta)^n}{1 - q^n D_\beta} , \tag{323}$$

we rewrite the (322) as

$$\prod_{\beta=1}^{N} \frac{(q^{\frac{1}{2}}t^2;q)_\infty (q^{\frac{1}{2}}t^{-2};q)_\infty}{(q;q)_\infty^2} \sum_{n\in\mathbb{Z}, m\in\mathbb{Z}^{\geq 0}} q^{mn}(C_\beta)^n (D_\beta)^m , \tag{324}$$

and justify the contribution from each component in turn.

The description for the duality interface is non-Lagrangian and involves both the $\mathcal{N} = (2,2)$ chiral $\Phi_\beta$ and its T-dual $\widetilde{\Phi}_\beta$. To perform the operator counting, we choose the duality frame of $\mathcal{T}$. One then counts the operators in $Q$-cohomology in $\Phi_\beta$, as well as the twist operators $e^{\varepsilon_\beta \widetilde{\phi}_\beta}$. Due to the superpotential in (211), only positive powers of the twist operator $e^{m\varepsilon_\beta \widetilde{\phi}_\beta}$ survive [70, 85], i.e. $m \geq 0$.

Note that $\phi_\beta$ is the bottom component of a $\mathcal{N} = (2,2)$ chiral $\Phi_\beta$ which transforms as a supermultiplet under vector and axial R-symmetries as:

$$\begin{aligned}
V: \quad & e^{-i\alpha F_V} \Phi_\beta(z,\bar{z},\theta^\pm,\bar{\theta}^\pm) e^{i\alpha F_V} = \Phi_\beta(z,\bar{z},e^{-i\alpha}\theta^\pm, e^{i\alpha}\bar{\theta}^\pm) + i\varepsilon_\beta \alpha , \\
A: \quad & e^{-i\alpha F_A} \Phi_\beta(z,\bar{z},\theta^\pm,\bar{\theta}^\pm) e^{i\alpha F_A} = \Phi_\beta(z,\bar{z},e^{\mp i\alpha}\theta^\pm, e^{\pm i\alpha}\bar{\theta}^\pm) .
\end{aligned} \tag{325}$$

The gauge-covariant operators which contribute in $Q$-cohomology are:

| Operator | Index |
|---|---|
| $e^{-\varepsilon_\beta \phi_\beta}$ | $C_\beta$ |
| $D_z^n \phi_\beta, n \geq 1$ | $q^n$ |
| $D_{\bar{z}}^n \phi_\beta, n \geq 1$ | $q^n$ |
| $e^{\varepsilon_\beta \widetilde{\phi}_\beta}$ | $D_\beta$ |
| $D_z^n \psi_-^{\phi_\beta}, n \geq 0$ | $q^{\frac{1}{2}+n} t^{-2}$ |
| $D_z^n \bar{\psi}_-^{\phi_\beta}, n \geq 0$ | $q^{\frac{1}{2}+n} t^2$ |

Here $D_z$ are gauge-covariant holomorphic derivatives on $E_\tau$, and $\psi_-^{\phi_\beta}, \bar\psi_-^{\phi_\beta}$ are fermions in $\Phi_\beta$. Notice that the gauge-covariant derivatives of the $\mathbb{C}^*$-valued scalars are

$$D_z \phi_\beta = \partial_z \phi_\beta + iA_z, \tag{326}$$

where $A_z$ acts in the appropriate representation, and are gauge neutral since the gauge transformation acts as a shift $\phi_\beta \to \phi_\beta + i\alpha(z,\bar z)$. Similarly $D_z \phi_\beta$ is neutral under flavour and $R$-symmetries. We now match the formulae (324):

- The fermions $D_z^n \psi_-^{\phi_\beta}$, and $D_z^n \bar\psi_-^{\phi_\beta}$ for $n \geq 0$ contribute $(q^{\frac{1}{2}} t^2; q)_\infty (q^{\frac{1}{2}} t^{-2}; q)_\infty$.

- The covariant derivatives $D_z^n \phi_\beta$ and $D_z^n \bar\phi_\beta$ for $n \geq 1$ contribute $(q;q)_\infty^{-2}$.

- We interpret the sum in (324) as the contribution of translation modes from $e^{-\varepsilon_\beta \phi_\beta}$ and winding modes from $e^{\varepsilon_\beta \widetilde\phi_\beta}$. Recall the BPS operators are arbitrary powers of $e^{-\varepsilon_\beta \phi_\beta}$, but only positive powers of $e^{\varepsilon_\beta \widetilde\phi_\beta}$. Inserting $\mathfrak{m}$ powers of the twist operator $e^{\varepsilon_\beta \widetilde\phi_\beta}$ means that any operator with charge $\mathfrak{n}$ under the translation symmetry (which is a gauge symmetry here) acquires a spin $\mathfrak{mn}$. Thus we have the contribution of composite operators

$$e^{-\mathfrak{n}\varepsilon_\beta \phi_\beta} e^{\mathfrak{m}\varepsilon_\beta \widetilde\phi_\beta} \quad \Rightarrow \quad q^{\mathfrak{mn}} (C_\beta)^{\mathfrak{n}} (D_\beta)^{\mathfrak{m}}. \tag{327}$$

A nice consistency check for this calculation is that the contribution (322) is symmetric under $t \leftrightarrow t^{-1}$, $C_\beta \leftrightarrow D_\beta$, although we chose to expand it asymmetrically in (324). One could alternatively expand as

$$\prod_{\beta=1}^N \frac{(q^{\frac{1}{2}} t^{-2}; q)_\infty (q^{\frac{1}{2}} t^2; q)_\infty}{(q;q)_\infty^2} \sum_{\mathfrak{n}\in\mathbb{Z}, \mathfrak{m}\in\mathbb{Z}^{\geq 0}} q^{\mathfrak{mn}} (D_\beta)^{\mathfrak{n}} (C_\beta)^{\mathfrak{m}}. \tag{328}$$

This exchanges translation and winding modes, and reflects the fact that physically one can choose to count operators in either the $\mathcal{T}$ or $\widetilde{\mathcal{T}}$ duality frame: in the $\widetilde{\mathcal{T}}$ frame $\widetilde\phi_\beta$ are translation modes and $\phi_\beta$ are winding modes.

In order to obtain the result in R-R sector, and to match onto the mother function, we perform a spectral flow. To do so, the operators must have $2\mathbb{Z}$ quantised left and right $R$-charges (concretely, in their character they must have even powers of $t$). Noting that $q^{\frac{1}{4}} t^{\pm 1}$ are the $U(1)_V$ and $U(1)_A$ fugacities respectively, we use gauge and flavour symmetries to redefine the $R$-charges so that

$$s \to s q^{\frac{1}{4}} t, \qquad \tilde s_\beta \to \tilde s_\beta (q^{\frac{1}{4}} t^{-1})^\beta. \tag{329}$$

We can then perform the spectral flow in the genus by substituting $t \to t q^{-1/4}$, redefining back the $R$-charges[17] by $s \to s t^{-1}$ and $\tilde s_\beta \to \tilde s_\beta t^\beta$ and then normalising by a monomial to ensure the result has the correct quasi-periodicities to reflect the contribution to mixed anomalies coming from the dynamical $\theta$-angles in (211). The contribution in R-R sector is thus:

$$\prod_{i=1}^N \frac{\vartheta(t^2)\vartheta((s v_\beta^{-1})^{-\varepsilon_\beta}(\tilde s_\beta/\tilde s_{\beta-1})^{\varepsilon_\beta})}{\vartheta(t(s v_\beta^{-1})^{-\varepsilon_\beta})\vartheta(t^{-1}(\tilde s_\beta/\tilde s_{\beta-1})^{\varepsilon_\beta})}. \tag{330}$$

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
