# Peer review of "d $\mathcal{N}=4$ Gauge Theories on an Elliptic Curve"

_SciPost Physics, doi:SciPost Phys. 13, 005 (2022)_

## Round 1 · Referee Report · Anonymous (Referee 1) · 2021-12-20

Strengths

1-Interesting new results on 3d N=4 (and 3d N=2) supersymmetric field theories on an elliptic curve, their boundary conditions, and their mathematical interpretation as elliptic cohomology classes.
2-Clearly written.
3-Very detailed.

Weaknesses

1-There is no introduction to the mathematical literature on elliptic cohomology for a physicists audience. For the non-expect in that part of maths (like myself), it is simply stated that elliptic cohomology classes is defined in terms of sections of lines bundles on a certain associated variety , but further "intuition" might have been appreciated. On the other hand, it is understandable, given the subject, that one cannot cover all the "basics".

Report

The authors study in much details the physics of 3d N=4 supersymmetric (softly broken to N=2 susy) NLSM, or rather their gauge theory UV completions. The paper include a review of all the relevant field theory results, including a number of things that clarify previous discussions.

The main motivation of the paper is to provide a SUSY QFT realisation of the mathematics of elliptic cohomology of some variety X (with a number of assumptions that simplify matter consideably). They do this by engineering X as a Higgs branch of a 3d N=4 theory, and by by showing how the space of vacua of the theory compactified on an elliptic curve can be identified with the elliptic cohomology variety of X. They further show that boundary conditions in this setup naturally engineer elliptic cohomology classes (sections of line bundles on the elliptic cohomology variety), and in particular that Neumann boundary condition can be identified with the so-called stable enveloppes.

The paper is clearly written and I would strongly recommend it for publication as it is.

Requested changes

Here are just a small list of likely typos:
1-In eq.(2.1) and similarly on page 8 and elsewhere, it seems the symbol "Z" is used to denote the maximal torus of a group (or the Cartan subalgebra of its Lie algebra)? This is confusing since this is not the center of G.
2-Two lines above (2.22), "m" should be "x".
3-p 25 and 26: "principle bundle" should be "principal bundle".
4-middle of p26: "can intersection"-->"can intersect".

---

## Round 1 · Referee Report · Anonymous (Referee 2) · 2022-5-4

Report

I highly recommend the paper: 3d $\mathcal{N}=4$ Gauge Theories on an Elliptic Curve by Mathew Bullimore, Daniel Zhang. for publication in your Journal.

The authors flesh out the physics underlying a construction of elliptic stable envelopes by Aganagic and Okounkov. Elliptic stable envelopes provide a basis in elliptic cohomology of Higgs and Coulomb branches of 3d N=4 theories. They play an important role in understanding the action of quantum symmetries of these gauge theories, and in establishing precise mathematical predictions of 3d mirror symmetry.

While it may not contain directly new results, and is moreover largely limited to the simplest case of abelian gauge theories, it does describe the relevant physics in clear and insightful ways which have not appeared elsewhere (not counting the paper by Nekrasov and Dedushenko that appeared simultaneously). It will become a standard reference in this important area at the intersection of mathematics and physics.

I do not see need for any modifications, the paper can be published as is.

---

## Editorial Decision

published